# DICT: UNCERTAINTY-CONSTRAINED TRUSTWORTHINESS FOR GRAPH LEARNING

## ABSTRACT

Graph Neural Networks (GNNs) face growing demands for trustworthiness, encompassing robustness, fairness, etc. However, these dimensions are often undermined by various perturbations, which induce distributional uncertainty and compromise the trustworthiness of GNNs. To address this, we propose DICT, a novel framework that models **di**stributional un**c**ertainty to achieve **t**rustworthy graph learning. Specifically, DICT formulates a unified optimization objective that captures perturbation-induced distributional shifts in graph topology, node features, and labels, and minimizes the worst-case risk over the uncertainty set. To make the primal infinite-dimensional problem tractable, we integrate strong duality and local Lipschitz continuity of the loss to reformulate the objective as a finite-dimensional min-max problem. We focus on robustness and fairness as primary instantiations of DICT because they are not only critical in real-world applications, but also provide transferable modeling principles for broader trustworthiness objectives. By formulating fairness in the form of an uncertainty set, DICT pioneers unified robustness and fairness within a single optimization framework. Extensive experiments across diverse benchmarks and GNN backbones demonstrate that DICT consistently improves both robustness and fairness, validating the effectiveness and adaptability of the DICT framework. We envision uncertainty constraints as a foundational principle for trustworthy graph learning and a step toward broader advancements in trustworthy AI.

## 1 INTRODUCTION

GNNs are effective for processing graph-structured data, and have been widely applied in domains such as social networks Fan et al. (2019), finance Motie & Raahemi (2024), and healthcare Paul et al. (2024). However, it is now widely acknowledged that task performance alone does not fully reflect the practicality and credibility of GNNs, highlighting the urgent need for trustworthy graph learning Grari et al. (2024); Hussain et al. (2022). Existing trustworthy graph learning methods address multiple aspects including but not limited to robustness, fairness, privacy, and interpretability Dai et al. (2024); Li & Wang (2025); Dai & Wang (2021a); Yuan et al. (2020).

Despite progress in their respective directions, existing approaches generally lack collaborative modeling capabilities, often resulting in conflicts when multiple trustworthiness objectives coexist in complex environments, and struggle to achieve holistic optimization. Taking robustness and fairness as an example, studies have shown that structural perturbations can not only degrade overall performance but also amplify prediction bias against certain sensitive groups, thereby aggravating unfairness Dai & Wang (2021a). This indicates that different trustworthiness objectives may conflict during optimization, highlighting the need for coordination and joint optimization within a unified framework. We suggest that structural perturbations Li & Wang (2025); Wu et al. (2019), group biases Wang et al. (2022); Yang et al. (2024), privacy attacks Olatunji et al. (2021); Xu et al. (2024), and inconsistent interpretations Yuan et al. (2020); Li et al. (2024) are all manifestation of uncertainty at the data level, which can lead to distribution drifts between the training distribution and the true distribution, making the model untrustworthy. In view of such, we ask:

> Can these trustworthy objectives be modeled into a distributional uncertainty framework?

To answer this question, we note that distributional uncertainty can be naturally modeled in an uncertainty set in a neighborhood around the empirical distribution. Particularly, distributionally robust optimization (DRO) Sagawa et al. (2019); Sadeghi et al. (2021); Wang et al. (2024) provides a mathematically rigorous and computationally feasible way to explicitly model and handle the uncertainty gap between the empirical and real distributions. By optimizing over neighborhoods that account for perturbations in graph structure, features, and labels, DRO provides a unified foundation for coordinating multiple trustworthiness objectives within a graph learning framework.

Based on this, we propose DICT, a well-founded framework for trustworthy graph learning, which constructs a Wasserstein uncertainty set to capture distributional shifts. While classical DRO relies on the i.i.d. assumption to construct empirical distributions, this assumption breaks down in graph-structured data, which are inherently non-i.i.d. due to dependencies between nodes and edges. This raises a fundamental challenge: **How can we define, analyze, and optimize DICT objectives in the context of graph learning?** To address this, we construct a Wasserstein uncertainty set by generating perturbed graphs of a single graph to form an empirical distribution, enabling DICT to model uncertainty without relying on the i.i.d. assumption. We leverage the strong duality assumption and the local Lipschitz continuity of the loss function to derive a tractable dual objective for the original infinite-dimensional DICT objective. For node classification tasks, we develop adaptive perturbation estimators by using first-order gradient linearization for labeled nodes and second-order Gaussian probing for unlabeled nodes. By solving the general DICT objective, which provides a blueprint that can optimize various trustworthiness objectives, including robustness, fairness, privacy, etc., by customizing loss functions, uncertainty sets, perturbations, and transportation costs.

In this paper, we focus on robustness and fairness as primary instantiations due to their transferable modeling principles. For robustness, we derive distributionally robust generalization bounds to prove the generalization performance of the model under the worst-case perturbations, providing theoretical and empirical guarantees on stability. For fairness, we make distribution shifts consistent with group fairness constraints by encoding sensitive attributes into the uncertainty set. We focus on these two objectives not only because they are among the most extensively studied and practically pressing goals in trustworthy learning, but also they offer transferable modeling principles applicable to broader trustworthiness dimensions, such as security and privacy. Therefore, by addressing both robustness and fairness objectives, our framework can be naturally extended to more general trustworthy learning tasks. In summary, our contributions are three-fold:

- **Framework:** We propose DICT, the first unified framework that models distributional uncertainty in graph topology, features, and labels for trustworthy graph learning, offering a general and extensible formulation for diverse trustworthiness objectives.

- **Methodology:** We derive a tractable dual formulation of the DICT objective via strong duality and local Lipschitz continuity, and develop efficient perturbation estimators for labeled and unlabeled nodes.

- **Evaluation:** We focus DICT on robustness and fairness, and experiments across benchmarks and backbones show it consistently improves performance and trustworthiness.

## 2 MODELING DICT FOR DISTRIBUTIONAL UNCERTAINTY

We consider the semi-supervised node classification setting on attributed graphs. Let $G = (\mathcal{V}, \mathcal{E})$ denote a weighted[1]graph, where $\mathcal{V} = \{v_1, \dots, v_N\}$ is the set of $N$ nodes, $\mathcal{E} \subseteq \mathcal{V} \times \mathcal{V}$ is the set of edges. For attributed graph, node features $X = \{x_1, \dots, x_N\}$ are provided, where $x_n$ corresponds to the $D$-dimensional feature of node $v_n$. $A \in \mathbb{R}^{N \times N}$ is the symmetric adjacency matrix of the graph $G$. We assume that each graph instance $(A, X, \mathbf{y})$ is drawn from an unknown joint distribution $\mathbb{P}_{\text{real}}$, where $\mathbf{y} = (y_1, \dots, y_N) \in [0, 1]^N$ denotes the latent ground-truth label vector for all $N$ nodes, while the observed labels are obtained via a binary mask $\mathbf{M_y} \in \{0, 1\}^N$ as $\widehat{\mathbf{y}} = \mathbf{y} \odot \mathbf{M_y}$, where $\odot$ denotes element-wise masking. Therefore, only the masked entries participate in supervision. Our goal is to approximate $\mathbb{P}_{\text{real}}$ based on a single observed graph $G$ and its partially observed label $\widehat{\mathbf{y}}$.

**Empirical Distribution Generator.** Define the attributed graph space $\mathcal{G} = \{(A, X) : A = A^\top\}$ and the label space $\mathcal{Y} = [0, 1]^N$. We consider the product space $\mathcal{Z} := \mathcal{G} \times \mathcal{Y}$, equipped with the

---

[1]Most fairness methods define edges via pairwise similarity scores rather than binary links. Modeling $A_{i,j} \in [0, 1]$ thus aligns with the data generation process and ensures the convexity required by DRO duality.

transportation cost $c(\cdot, \cdot)$ (to be specified in Equation (5)). Since this single sample offers limited support, we propose a $K$-fold stochastic perturbation scheme to enrich the empirical distribution. Specifically, we generate $K$ perturbed graphs $\{(A_k, X_k)\}_{k=1}^K$ and their corresponding noisy labels $\{\widehat{\mathbf{y}}_k\}_{k=1}^K$ by independently injecting noise into the graph structure, node features, and labels. While node-level interactions are inherently non-i.i.d., the set of perturbed graphs can be viewed as approximately i.i.d. at the graph level. This yields a mixed empirical distribution:

$$\mathbb{P}_{\mathrm{tr}} := \eta\, \delta_{(A, X, \widehat{\mathbf{y}})} + (1 - \eta) \cdot \tfrac{1}{K} \sum_{k=1}^K \delta_{(A_k, X_k, \widehat{\mathbf{y}}_k)},$$

where $\eta \in [0, 1]$ balances the contribution of the original graph and its perturbations, and $\delta_{(A, X, \widehat{\mathbf{y}})}$ denotes the Dirac measure centered at the instance $(A, X, \widehat{\mathbf{y}})$. This mixture distribution serves as the center of the Wasserstein uncertainty set in our DICT objective.

**Objective.** Given the observed labels $\{\widehat{\mathbf{y}}, \widehat{\mathbf{y}}_k\}_{k=1}^K$, the goal is to recover the corresponding ground-truth labels $\{\mathbf{y}, \mathbf{y}_k\}_{k=1}^K$. To this end, we aim to learn a predictive function $f(A, X; \theta)$ that infers the missing labels while satisfying the trustworthiness criteria. The learning objective reduces to:

$$\min_{\theta \in \Theta} \mathbb{E}_{(A, X, \mathbf{y}) \sim \mathbb{P}_{\mathrm{tr}}} \left[ \mathcal{L}\left( f\left( A, X; \theta \right), \mathbf{y} \right) \right], \tag{1}$$

where $\mathcal{L}$ is instantiated according to the downstream tasks. However, in trustworthy graph learning, distributional shift occurs due to discrepancies between training and testing data distributions. To model distributional uncertainty in graph learning, we consider the following optimization problem:

$$\min_{\theta \in \Theta} \sup_{\mathbb{P} \in \mathcal{P}} \mathbb{E}_{(A, X, \mathbf{y}) \sim \mathbb{P}} \left[ \mathcal{L}\left( f(A, X; \theta), \mathbf{y} \right) \right], \tag{2}$$

where we assume that the real graph distribution $\mathbb{P}_{\mathrm{real}}$ falls within an uncertainty set $\mathcal{P} := \{\mathbb{P} : \delta(\mathbb{P}, \mathbb{P}_{\mathrm{tr}}) \leq r\}$ of radius $r$ around the empirical distribution $\mathbb{P}_{\mathrm{tr}}$. To this end, our goal is to promote the worst-case performance in this uncertainty set. Depending on the choice of distributional metric $\delta(\cdot, \cdot)$, various uncertainty sets $\mathcal{P}$ can be constructed in practice, inducing distinct trustworthiness aspects. The common choice is the $\zeta$-divergence Csiszár et al. (2004), and prior work on DRO Chen et al. (2025) defines the uncertainty set via $\chi^2$-divergence. However, such uncertainty sets only contain distributions absolutely continuous w.r.t. $\mathbb{P}_{\mathrm{tr}}$. This enforces that the support of $\mathcal{P}$ is a subset of the original distribution's support, limiting its expressiveness and making it less suitable for modeling distribution shifts.

To overcome this limitation, we adopt the Wasserstein integral probability metric (Wasserstein IPM) Esfahani et al. (2015), which offers greater flexibility by allowing shifts in support and explicitly modeling transportation costs over structured domains such as graphs. This enables fine-grained control over perturbations in topology, features, and labels while maintaining continuity and theoretical tractability. The Wasserstein IPM between two graph distributions $\mathbb{P}$ and $\mathbb{Q}$ is defined as:

$$\delta_W(\mathbb{P}, \mathbb{Q}) := \inf_{\pi \in \Pi(\mathbb{P}, \mathbb{Q})} \mathbb{E}_\pi \left[ c\left( (A, X, \mathbf{y}), (\widehat{A}, \widehat{X}, \widehat{\mathbf{y}}) \right) \right],$$

where $(A, X, \mathbf{y}) \sim \mathbb{P}, (\widehat{A}, \widehat{X}, \widehat{\mathbf{y}}) \sim \mathbb{Q}$, $c$ denotes transportation cost and $\Pi(\mathbb{P}, \mathbb{Q})$ denotes the set of all joint distributions with marginals $\mathbb{P}$ and $\mathbb{Q}$. Formally, we consider constructing the uncertainty set as follows:

$$\mathcal{P}(\mathbb{P}_{\mathrm{tr}}, r) := \{\mathbb{P} \in \mathcal{B} : \delta_W(\mathbb{P}, \mathbb{P}_{\mathrm{tr}}) \leq r\}.$$

Here, $\mathcal{B}$ is the set of Borel probability measures on $\mathcal{Z}$. By integrating this uncertainty into Equation (2), we propose the following problem as the primal DICT objective:

$$\min_{\theta \in \Theta} \sup_{\mathbb{P} \in \mathcal{P}(\mathbb{P}_{\mathrm{tr}}, r)} \mathbb{E}_{(A, X, \mathbf{y}) \sim \mathbb{P}} \mathcal{L}\left( f(A, X; \theta), \mathbf{y} \right). \tag{3}$$

**Remark.** Directly optimizing over the uncertainty set $\mathcal{P}(\mathbb{P}_{\mathrm{tr}}, r)$ constitutes an infinite-dimensional optimization problem. Following prior work Blanchet et al. (2019); Gao & Kleywegt (2023), we leverage strong duality to reformulate the inner maximization as a minimization over the Lagrange dual variable $\lambda \geq 0$ in Section 3. In this view, Equation (3) provides not only a distributionally robust objective but also a unified foundation for trustworthy GNNs design. By customizing the uncertainty set $\mathcal{P}$, transportation cost $c$, and task loss $\mathcal{L}$, DICT can flexibly accommodate diverse trustworthiness dimensions under a general optimization framework. Due to space constraints, we focus on two technically central axes: **robustness** and **fairness**. As an extension of fairness, differential privacy (DP) Dwork et al. (2006); Kasiviswanathan et al. (2011) prevents leakage of sensitive information by injecting noise into data or gradients. In our setting, local differential privacy (LDP) Dai & Wang (2022) can be realized by perturbing binary sensitive attributes with probability $\rho = \frac{1}{\exp(\epsilon) + 1}$, ensuring $\epsilon$-LDP and enabling fair GNNs training with strong privacy guarantees.

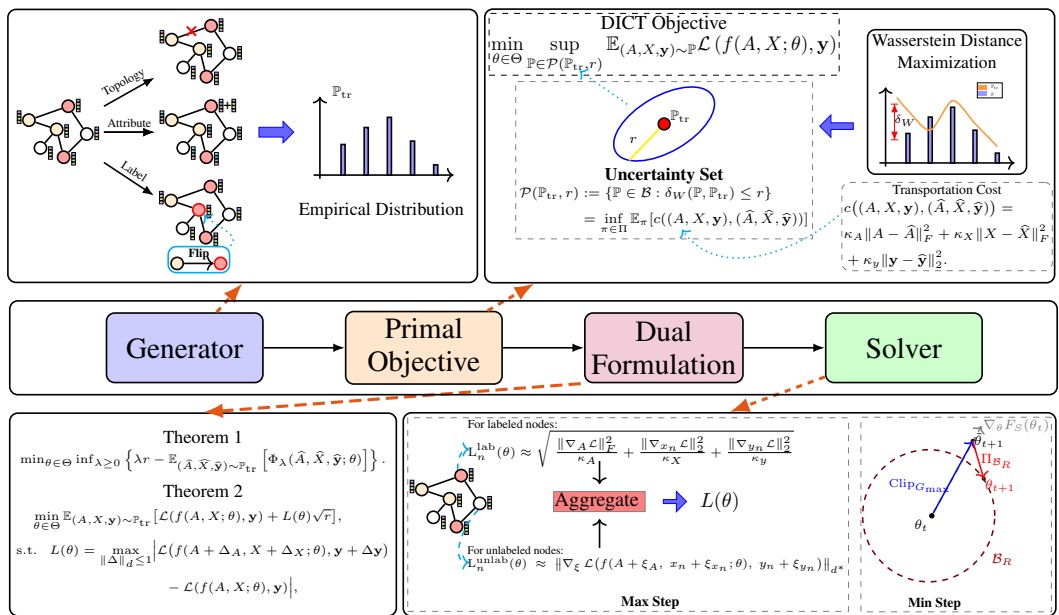

Figure 1: An overview of the DICT framework. We (1) perturb topology, features, and labels to form $\mathbb{P}_{\text{tr}}$; (2) define Wasserstein DICT objective; (3) apply duality to get Lipschitz-regularized loss; (4) estimate $L(\theta)$ and solve the min–max problem.

## 3 SOLVING DICT OBJECTIVE VIA LIPSCHITZ DUALITY

In this section, we detail the overall pipeline of the DICT framework. As described in Figure 1, our framework consists of four components: (1) a generator that constructs an empirical graph distribution by injecting perturbations, (2) a primal objective that defines a Wasserstein uncertainty set to model perturbation-induced distributional shifts, (3) a dual reformulation that transforms the infinite-dimensional worst-case risk into a tractable min-max optimization, and (4) a solver that efficiently optimizes the min-max problem by estimating the local Lipschitz constant.

To reformulate the primal objective in Equation (3) as a finite-dimensional problem, we derive the following dual form of the DICT objective:

**Theorem 1** (Strong Dual Formulation of the DICT Objective). *Consider the primal DICT objective in Equation (3), then under the strong duality conditions (see Appendix B, Lemma 1), it admits the following equivalent dual formulation:*

$$\min_{\theta \in \Theta} \inf_{\lambda \geq 0} \left\{ \lambda r - \mathbb{E}_{(\widehat{A}, \widehat{X}, \widehat{\mathbf{y}}) \sim \mathbb{P}_{\text{tr}}} \left[ \Phi_\lambda(\widehat{A}, \widehat{X}, \widehat{\mathbf{y}}; \theta) \right] \right\}, \tag{4}$$

*where $\lambda$ is the Lagrange dual variable, and the inner infimum is given by:* $\Phi_\lambda\left(\widehat{A}, \widehat{X}, \widehat{\mathbf{y}}; \theta\right) :=$ $\inf_{(A, X, \mathbf{y})} \{ \lambda\, c\big((A, X, \mathbf{y}), (\widehat{A}, \widehat{X}, \widehat{\mathbf{y}})\big) - \mathcal{L}\big(f(A, X; \theta), \mathbf{y}\big) \}$. *The cost function $c(\cdot, \cdot)$ defines a ground quadratic metric over the joint graph-label space $\mathcal{Z} = \mathcal{G} \times \mathcal{Y}$, specified as:*

$$c\big((A, X, \mathbf{y}), (\widehat{A}, \widehat{X}, \widehat{\mathbf{y}})\big) = \kappa_A \|A - \widehat{A}\|_F^2 + \kappa_X \|X - \widehat{X}\|_F^2 + \kappa_y \|\mathbf{y} - \widehat{\mathbf{y}}\|_2^2, \tag{5}$$

*where $\| \cdot \|_F$ denotes the Frobenius norm, $\| \cdot \|_2$ denotes the Euclidean norm, and the coefficients $(\kappa_A, \kappa_X, \kappa_y)$ reflect the relative confidence in the alignment between an instance and its perturbations in terms of structure, features, and labels.*

The proof is presented in Appendix C.1. Theorem 1 derives the dual formulation of the DICT objective, establishing a unified distributional framework for downstream optimization.

To derive an equivalent closed-form objective, we reparameterize samples in the uncertainty set as perturbations to observed instances. Leveraging local Lipschitz continuity of the loss function, we analyze the dual potential $\Phi_\lambda$ and establish an exact reformulation of the dual objective in Equation (4) as a min-max problem, as formalized below:

**Theorem 2** (Equivalent Lipschitz-Regularized Objective). *Under the assumptions of strong duality and local Lipschitz continuity of the loss function $\mathcal{L}$ (as established in GNN Lipschitz bounds Jia et al. (2023)), the DICT problem Equation (3) is equivalent to the following optimization problem:*

$$\min_{\theta \in \Theta} \ \mathbb{E}_{(A,X,\mathbf{y}) \sim \mathbb{P}_{tr}} \big[ \mathcal{L}(f(A,X;\theta), \mathbf{y}) + L(\theta)\sqrt{r} \big],$$

$$\text{s.t.} \quad L(\theta) = \max_{\|\Delta\|_d \leq 1} \Big| \mathcal{L}\big(f(A + \Delta_A, X + \Delta_X; \theta), \mathbf{y} + \Delta\mathbf{y}\big) - \mathcal{L}(f(A,X;\theta), \mathbf{y}) \Big|. \quad (6)$$

*Here, $L(\theta)$ denotes the local Lipschitz constant of the loss $\mathcal{L}$ with respect to perturbations $\Delta = (\Delta_A, \Delta_X, \Delta_y) \in \mathcal{Z}$ applied to $(A, X, \mathbf{y}) \in \mathcal{Z}$. The perturbation norm is $\|\cdot\|_d$ defined as $\|\Delta\|_d := \sqrt{\kappa_A \|\Delta_A\|_F^2 + \kappa_X \|\Delta_X\|_2^2 + \kappa_y \|\Delta_y\|_2^2}$.*

The detailed proof is provided in Appendix C.2. Here, we present the *sketch of proof.* The dual formulation of the DICT objective has already been established in Theorem 2. Assuming $\mathcal{L}$ is locally $L(\theta)$-Lipschitz with respect to a perturbation norm $\|\Delta\|_d$, we upper- and lower-bound the dual potential using Young's inequality and a scaled adversarial perturbation. This yields a tight closed-form expression $\Phi_\lambda = \mathcal{L} + \frac{L(\theta)^2}{4\lambda}$. Minimizing over $\lambda$ gives the optimal $\lambda^\star = \frac{L(\theta)}{2\sqrt{r}}$, leading to the final form of the objective in Theorem 2.

Notably, Theorem 2 reformulates the primal DICT objective as a regularized empirical loss, where the penalty $L(\theta)\sqrt{r}$ quantifies sensitivity to local perturbations. The hyperparameter $r$ modulates the trade-off between trustworthiness and accuracy by scaling the uncertainty set, while $L(\theta)$ captures the local Lipschitz continuity of the loss. This formulation turns an infinite-dimensional worst-case problem into a tractable surrogate with a single interpretable regularizer, solving a min-max game:

- **Maximization.** DICT implicitly identifies the worst-case perturbation direction over node features, graph topology, and labels. Rather than explicitly enumerating adversarial graphs, it encodes this direction via the Lipschitz penalty $L(\theta)\sqrt{r}$, which captures the maximal rate of increase in loss over a Wasserstein ball.

- **Minimization.** Given this robustified surrogate objective, the model parameters $\theta$ are optimized to minimize empirical loss while controlling sensitivity to perturbations. This promotes local smoothness in the model's predictions and fosters representation invariance under semantically meaningful distributional shifts.

**Remark.** Although conventional graph regularizations (e.g., Laplacian smoothing) encourage prediction smoothness, DICT provides a more adaptive and generalizable mechanism: (1) its regularization is dynamic, tied to the model's local Lipschitz constant, and hence focuses on directions of maximal sensitivity; and (2) by optimizing over Wasserstein neighborhoods, DICT augments training with adversarially perturbed virtual samples, improving robustness and generalization.

Directly computing the Lipschitz constant $L(\theta)$ in Equation (6) is generally intractable, as it involves a worst-case analysis of the loss sensitivity to multimodal perturbations. To make the optimization practical, we adopt a gradient-based linear approximation strategy inspired by adversarial training methods Goodfellow et al. (2015); Feng et al. (2019). To this end, we define the node specific Lipschitz constant $L_n(\theta)$ by restricting the perturbations in the definition of $L(\theta)$ to be supported only on node $n$, i.e. $\Delta \in \mathcal{S}_n := \{\Delta | \Delta \text{ is supported only on node n}\}$.

For each node $n$ in the labeled set $\mathcal{N}_o$, we approximate the node-specific Lipschitz constant $L_n(\theta)$ using the first-order Taylor expansion of the loss w.r.t. perturbation $\Delta_n$, i.e. $L_n(\theta) \approx \max_{\|\Delta_n\|_d} \langle \nabla_{\Delta_n} \mathcal{L}, \Delta_n \rangle$, which corresponds to the dual norm of the gradient $\|\nabla_{\Delta_n} \mathcal{L}\|_{d^*}$ under the perturbation norm $\|\cdot\|_d$ (see Appendix C.3 for formal definition and proof in Lemma 2). This yields:

$$\mathrm{L}_n^{lab}(\theta) = \sqrt{\frac{\|\nabla_A \mathcal{L}\|_F^2}{\kappa_A} + \frac{\|\nabla_{x_n} \mathcal{L}\|_2^2}{\kappa_X} + \frac{\|\nabla_{y_n} \mathcal{L}\|_2^2}{\kappa_y}},$$

Here, each gradient is computed with respect to one input channel while holding others fixed, and the weights $(\kappa_A, \kappa_X, \kappa_y)$ are inherited from the transportation cost.

For unlabeled nodes, such approximation is infeasible since the gradient will always be zero. This is because $\mathcal{L}(f(A, x_n; \theta), \widehat{y}_n)$ achieves the minimum value at $x_n$ (note that $\hat{y}_n := f(A, x_n; \theta)$ for unlabeled nodes). Realizing that the first-order gradient is always zero, we approximate the

local Lipschitz constant via a second-order Taylor approximation of $\mathcal{L}(f(A, x_n; \theta), \widehat{y}_n)$. That is $L_n(\theta) \approx \arg\max_{\|\Delta_n\|_d \leq 1} \frac{1}{2}\Delta_n^{\mathsf{T}} H_n \Delta_n$, where $H_n = \nabla^2_{\Delta_n \Delta_n} \mathcal{L}$ is the Hessian matrix. To estimate this quantity efficiently, we apply a single-sample approximation using a Gaussian probe $\xi \sim \mathcal{N}(0, I)$ in the space of node-local perturbations $\Delta_n$. The resulting surrogate is:

$$\mathrm{L}_n^{\text{unlab}}(\theta) = \left\| \nabla_\xi \mathcal{L}\big( f(A + \xi_A,\ x_n + \xi_{x_n}; \theta),\ y_n + \xi_{y_n} \big) \right\|_{d^*}.$$

The detailed procedure is provided in Appendix C.3. Finally, we aggregate node-level sensitivity scores $\mathrm{L}_n(\theta)$ into a global regularizer $L(\theta)$ using the $p$-norm smoothing Yang et al. (2020).

## 4  DICT FOR TRUSTWORTHY GRAPH LEARNING

In this section, we build on the previously established DICT objective and dual formulation to show how the framework addresses specific trustworthiness goals. Specifically, we show how DICT can be specialized to enhance robustness and fairness. By formulating fairness in the form of an uncertainty set, the objective unifies robustness and fairness within a single optimization framework. Detailed descriptions of the perturbation strategies are provided in Appendix D.1.

### 4.1  ACHIEVING ROBUSTNESS WITH DICT

GNNs are known to be vulnerable to adversarial perturbations on both graph topology and node features. Classical defense methods cast robustness as a sample-level min-max objective:

$$\min_{\theta \in \Theta} \max_{\Delta_A \in \mathcal{P}_A, \Delta_X \in \mathcal{P}_X} \mathcal{L}\big(f(A + \Delta_A, X + \Delta_X; \theta), \mathbf{y}\big),$$

where $\Delta_A$ and $\Delta_X$ denote constrained perturbations to the graph topology and node features, respectively, within predefined perturbation sets $\mathcal{P}_A$ and $\mathcal{P}_X$. We show that such adversarial training can be recovered as a special case of our DICT. Specifically, we specialize DICT for robustness by formulating the following objective:

$$\min_{\theta \in \Theta} \sup_{\mathbb{P} \in \mathcal{P}(\mathbb{P}_{\text{tr}}, r)} \mathbb{E}_{(A, X, \mathbf{y}) \sim \mathbb{P}} \left[ \mathcal{L}_{\text{Rb}} \left( f(A, X; \theta), \mathbf{y} \right) \right], \tag{7}$$

where the robust loss $\mathcal{L}_{\text{Rb}}$ incorporates multiple robustness-promoting terms: $\mathcal{L}_{\text{Rb}} = \mathcal{L}_{\text{train}} + \varphi \mathcal{L}_{\text{smooth}} + \beta \mathcal{L}_{\text{adv}}$. Here, $\mathcal{L}_{\text{train}}$ is the supervised loss on labeled nodes (e.g., cross-entropy); $\mathcal{L}_{\text{smooth}}$ enforces prediction consistency between connected nodes, instantiated via Laplacian regularization $\sum_{i,j=1}^{N} A_{ij} |\mathbf{y}_i - \mathbf{y}_j|$; and $\mathcal{L}_{\text{adv}}$ penalizes local instability through virtual adversarial noise, such as $d(f(A, \mathbf{x}_i; \theta), f(A, \widehat{\mathbf{x}}_i; \theta))$, where $d(\cdot, \cdot)$ is a divergence metric like KL, JS, or $\ell_2$.

**Proposition 1** (Robustness Guarantee of DICT). *Let* $\mathbb{P}_{\text{tr}} = \delta_{(A,X,y)}$ *be a Dirac measure over a single graph instance. Define the uncertainty set as:* $\mathcal{P}(\mathbb{P}_{\text{tr}}, r) = \left\{ \delta_{(A+\Delta_A, X+\Delta_X, y)} \mid \kappa_A \|\Delta_A\|_F^2 + \kappa_X \|\Delta_X\|_F^2 \leq r \right\}$. *Then the DICT objective reduces to:*

$$\min_{\theta \in \Theta} \max_{\Delta_A, \Delta_X \in \mathbb{L}_r} \mathcal{L}(f(A + \Delta_A, X + \Delta_X; \theta), y),$$

*where* $\mathbb{L}_r$ *denotes the perturbation ellipsoid* $\kappa_A \|\Delta_A\|_F^2 + \kappa_X \|\Delta_X\|_F^2 \leq r$. *Thus, classical adversarial training is recovered as a special case of DICT.*

We further establish the generalization bounds under the setting, showing that DICT not only enhances worst-case robustness but also improves generalization:

**Theorem 3** (Generalization Bound for DICT). *Let* $\mathcal{A}$ *be clipped projected-SGD with step sizes* $\alpha_t$, *budget* $S_\alpha = \sum_t \alpha_t$, *and clipping threshold* $G_{\max}$. *On the mixed training set* $S = \{(A, X, \widehat{\mathbf{y}}), (A_k, X_k, \widehat{\mathbf{y}}_k)\}_{k=1}^{K}$, *define* $\widehat{R}_{\text{rob}}(\theta) = \mathbb{E}_{\mathbb{P}_{\text{tr}}}[\mathcal{L}(f(A, X; \theta), \mathbf{y})] + L(\theta)\sqrt{r}$. *If* $0 \leq \mathcal{L} \leq B$, *then with probability at least* $1 - \delta$, *the output* $\theta_S = \mathcal{A}(S)$ *satisfies:*

$$R_{\text{rob}}(\theta_S) \leq \widehat{R}_{\text{rob}}(\theta) + \frac{4BG_{\max}S_\alpha}{K+1} + B\sqrt{\frac{\log(1/\delta)}{2(K+1)}}.$$

Assumptions and proof are in Appendix E. The bound combines the empirical DICT loss, a vanishing stability term (empirically validated in Appendix I.2), and a concentration term, showing that adversarial augmentation improves robustness and generalization. Particularly, complexity is absorbed into the Lipschitz-sensitive term $L(\theta)\sqrt{r}$.

### 4.2 ENFORCING FAIRNESS WITH DICT

Perturbations in graph data can amplify social biases and cause prediction discrepancies across sensitive groups, making fairness a key challenge in graph learning. We focus on the binary case and defer multi-group extensions to Appendix D.2. Let $\mathbf{s} \in \{0, 1\}^N$ be the sensitive attribute, where $s_n = 1$ denotes the protected group. Most existing methods incorporate fairness constraints (Dong et al., 2021; Kang et al., 2020; Li et al., 2021a; Dai & Wang, 2021a), typically formulated as:

$$\min_{\theta \in \Theta} \mathcal{L}(f(A, X; \theta), \mathbf{y}) + \gamma \mathcal{L}_{\text{SP}}(f(A, X; \theta), \mathbf{s}), \tag{8}$$

where $\mathcal{L}_{\text{SP}}$ is a fairness regularizer quantifying statistical parity violation between sensitive groups. It is typically instantiated as: $\mathcal{L}_{\text{SP}} = d(f(A, x_n, s_n = 1; \theta), f(A, x_n, s_n = 0; \theta))$, where $d(\cdot, \cdot)$ is the specified distance such as KL, JS, or $\ell_2$. Unlike regularization-based methods, DICT achieves fairness by restricting the uncertainty set to penalize perturbations along sensitive attributes, thereby unifying robustness and fairness within a distributional optimization framework. To capture fairness perturbations, we extend the Wasserstein transportation cost to include the sensitive attribute:

$$c\big((A, X, \mathbf{y}, \mathbf{s}), (\widehat{A}, \widehat{X}, \widehat{\mathbf{y}}, \widehat{\mathbf{s}})\big) = \kappa_A \|A - \widehat{A}\|_F^2 + \kappa_X \|X - \widehat{X}\|_F^2 + \kappa_s \|\mathbf{s} - \widehat{\mathbf{s}}\| + \kappa_y \|\mathbf{y} - \widehat{\mathbf{y}}\|_2^2, \tag{9}$$

where the coefficient $\kappa_s$ focuses on biases. We instantiate DICT to achieve fairness by:

$$\min_{\theta \in \Theta} \sup_{\mathbb{P} \in \mathcal{P}(\mathbb{P}_{\text{tr}}, r)} \mathbb{E}_{\mathbf{s} \sim \mathbb{P}_{\mathbf{s}}} \left[ \mathbb{E}_{(G, \mathbf{y}) \sim \mathbb{P}_{|\mathbf{s}}} \left[ \mathcal{L}_{\text{Fair}}(f(G; \theta), \mathbf{y}) \right] \right], \tag{10}$$

where $\mathcal{L}_{\text{Fair}}$ consists of supervised loss and fairness penalty: $\mathcal{L}_{\text{Fair}} = \mathcal{L}_{\text{train}} + \gamma \mathcal{L}_{\text{adv-fair}}$. Here, $\mathcal{L}_{\text{adv-fair}}$ enforces invariance to sensitive flips: $\mathcal{L}_{\text{adv-fair}} = \frac{1}{N} \sum_{i=1}^{N} d\left(f(A, \mathbf{x}_i, \mathbf{s}_i; \theta), f(A, \mathbf{x}_i, 1 - \mathbf{s}_i; \theta)\right)$.

**Proposition 2** (Fairness Guarantee of DICT). *Define the uncertainty set with the fairness-aware cost:* $\mathcal{P}(\mathbb{P}_{\text{tr}}, r) = \left\{ \delta_{(A, X, y, \mathbf{s} + \Delta_s)} \,\middle|\, \kappa_s \|\Delta_s\|^2 \leq r \right\}$. *Then the DICT objective reduces to:*

$$\min_{\theta \in \Theta} \max_{\Delta_s \in \mathbb{O}_r} \mathcal{L}(f(A, X, \mathbf{s} + \Delta_s; \theta), \mathbf{y}), \tag{11}$$

*which recovers the fairness-constrained objective as a special case when $r \geq \gamma^2$. Here, $\mathbb{O}_r$ denotes the perturbation ball $\kappa_s \|\Delta_s\|^2 \leq r$.*

By integrating sensitive-attribute awareness directly into the uncertainty set of Equation (5) rather than explicit constraints, the framework enforces fairness across subgroups by implicitly bounding disparities between sensitive groups.

## 5 EXPERIMENTS

In this section, we evaluate the proposed DICT by addressing the following research questions (RQs). **RQ1**: Can DICT consistently improve robustness and fairness across diverse datasets and GNN architectures? **RQ2**: How does DICT compare to state-of-the-art baselines designed specifically for robustness and fairness? **RQ3**: How do different modalities of perturbation (structure, features, labels, sensitive attributes) contribute to robustness and fairness under DICT, and how can we balance them via hyperparameter tuning?

### 5.1 EXPERIMENTAL SETTINGS

**Datasets.** We evaluate DICT on both fairness and robustness benchmarks. For fairness evaluation, we follow prior work Dong et al. (2022a); Dai & Wang (2021a) and adopt four attributed graph datasets of varying scales. Specifically, we include two **small-scale graphs**: German and NBA, and two **large-scale graphs**: Bail and Credit. To further assess robustness, we conduct experiments on three citation network benchmarks: Cora, Citeseer, and PubMed. Full dataset statistics are listed in Table 1.

Table 1: Statistics of datasets used in fairness and robustness experiments.

| Dataset | #Nodes | #Edges | #Features | #Classes | Sensitive |
|---|---|---|---|---|---|
| German | 1,000 | 22,242 | 27 | 2 | Gender |
| NBA | 403 | 16,570 | 39 | 2 | Nationality |
| Bail | 18,876 | 321,308 | 18 | 2 | Race |
| Credit | 30,000 | 1,436,858 | 13 | 2 | Age |
| Cora | 2,708 | 5,429 | 1,433 | 7 | – |
| Citeseer | 3,327 | 4,732 | 3,703 | 6 | – |
| Pubmed | 19,717 | 44,338 | 500 | 3 | – |

**GNN Backbones and Baselines.** We adopt four widely used graph neural networks (GNNs) as the backbone encoders: GCN Kipf & Welling (2017), GIN Xu et al. (2019b), GraphSAGE Hamilton et al. (2017) and GAT Veličković et al. (2018). To evaluate DICT, we compare it against representative baselines targeting either robustness or fairness. For robustness evaluation, we consider GCN-SVD Entezari et al. (2020), GCN-Jaccard Wu et al. (2019), and Pro-GNN Jin et al. (2020), all of which are implemented with a GCN backbone to ensure fair comparison. For fairness evaluation, we benchmark against three fairness-aware GNNs: NIFTY Agarwal et al. (2021), FairGNN Dai & Wang (2021a), and FairVGNN Wang et al. (2022).

**Evaluation Metrics.** We evaluate model performance in node classification task from two key perspectives: robustness and fairness. (1) Robustness Metric. We assess model behavior on both clean and perturbed graph instances. **Clean Acc** measures classification performance on the original, unmodified graph. **Attack Acc** is computed on graphs perturbed by synthetic noise, To quantify robustness, we compute the absolute performance degradation under distributional shift, defined as $\Delta\text{Acc} = |\text{AttackAcc} - \text{CleanAcc}|$. A lower $\Delta\text{Acc}$ indicates stronger robustness. (2) Fairness Metrics. we adopt two widely-used group fairness metrics ($\Delta_{\text{SP}}$) and ($\Delta_{\text{EO}}$) Dai & Wang (2021a); Chen et al. (2025). Lower values of both metrics indicate improved fairness.

Table 2: Comparison between GNNs with original networks (Vanilla) and trustworthy network (DICT). Metrics are categorized into **Robustness** and **Fairness**. ↑ higher is better; ↓ lower is better. Best results are in bold.

| | | GCN | | GraphSAGE | | GIN | | GAT | |
|---|---|---|---|---|---|---|---|---|---|
| | | Vanilla | DICT | Vanilla | DICT | Vanilla | DICT | Vanilla | DICT |
| German | Clean Acc(%) ↑ | 64.80 ± 3.19 | **70.91 ± 0.58** | 66.49 ± 0.3 | **71.25 ± 0.7** | 69.39 ± 0.8 | **70.87 ± 0.5** | 65.72 ± 2.27 | **70.22 ± 0.77** |
| | Attack Acc(%) ↑ | 60.95 ± 5.80 | **69.60 ± 0.11** | 62.87 ± 1.3 | **69.98 ± 0.6** | 63.37 ± 0.5 | **69.38 ± 1.2** | 62.35 ± 1.80 | **68.33 ± 0.25** |
| | ΔAcc(%) ↓ | 3.85 | **1.31** | 3.62 | **1.27** | 6.02 | **1.49** | 3.37 | **1.89** |
| | $\Delta_{\text{SP}}$(%) ↓ | 7.04 ± 0.43 | **1.08 ± 0.15** | 6.26 ± 1.2 | **1.14 ± 0.3** | 3.37 ± 1.0 | **0.6 ± 0.3** | 8.27 ± 0.31 | **1.17 ± 0.21** |
| | $\Delta_{\text{EO}}$(%) ↓ | 7.47 ± 0.85 | **0.63 ± 0.26** | 5.67 ± 0.8 | **0.53 ± 0.4** | 2.05 ± 1.1% | **1.30 ± 0.8** | 7.32 ± 0.25 | **0.82 ± 0.15** |
| Bail | Clean Acc(%) ↑ | 88.12 ± 0.7 | **88.62 ± 0.3** | 89.13 ± 0.3 | **94.69 ± 0.7** | 82.85 ± 0.8 | **85.89 ± 0.5** | 87.32 ± 0.7 | **89.37 ± 0.5** |
| | Attack Acc(%) ↑ | 67.65 ± 0.6 | **84.12 ± 0.1** | 70.23 ± 1.3 | **91.25 ± 0.6** | 75.81 ± 0.5 | **83.15 ± 1.2** | 68.25 ± 0.6 | **87.23 ± 0.6** |
| | ΔAcc(%) ↓ | 20.47 | **4.50** | 18.90 | **3.44** | 7.04 | **2.74** | 19.07 | **2.14** |
| | $\Delta_{\text{SP}}$(%) ↓ | 8.18 ± 1.2 | **1.83 ± 0.9** | 2.28 ± 1.2 | **0.79 ± 0.4** | 4.79 ± 1.0 | **2.24 ± 1.3** | 9.38 ± 1.2 | **1.63 ± 0.8** |
| | $\Delta_{\text{EO}}$(%) ↓ | 5.26 ± 1.3 | **1.39 ± 1.0** | 1.37 ± 0.8 | **0.91 ± 0.7** | 6.92 ± 1.1 | **1.25 ± 1.1** | 7.19 ± 1.3 | **1.29 ± 0.4** |
| NBA | Clean Acc(%) ↑ | 65.86 ± 0.6 | **71.48 ± 0.2** | 75.17 ± 0.3 | **75.90 ± 0.4** | 60.64 ± 0.8 | **67.46 ± 0.9** | 72.34 ± 0.8 | **73.42 ± 0.2** |
| | Attack Acc(%) ↑ | 39.75 ± 0.7 | **69.75 ± 0.2** | 60.24 ± 0.3 | **70.35 ± 0.3** | 45.78 ± 0.2 | **66.18 ± 1.3** | 66.40 ± 1.1 | **70.31 ± 0.7** |
| | ΔAcc(%) ↓ | 26.11 | **1.73** | 14.86 | **5.55** | 14.86 | **1.28** | 5.94 | **3.11** |
| | $\Delta_{\text{SP}}$(%) ↓ | 7.53 ± 0.2 | **3.21 ± 0.7** | 6.74 ± 1.2 | **1.42 ± 0.5** | 16.13 ± 1.1 | **3.30 ± 1.4** | 13.20 ± 1.3 | **2.21 ± 0.7** |
| | $\Delta_{\text{EO}}$(%) ↓ | 10.28 ± 0.3 | **2.72 ± 1.1** | 14.20 ± 0.8 | **2.67 ± 0.8** | 10.53 ± 0.3 | **2.72 ± 1.2** | 11.90 ± 2.0 | **1.72 ± 1.1** |

## 5.2 MAIN RESULTS

**Consistent Improvements Across Datasets and Architectures (RQ1).** We evaluate DICT against vanilla GNNs on three datasets (German, Bail, NBA) and four architectures (GCN, Graph-SAGE, GIN, GAT). Results are shown in Table 2. We highlight several key observations: (1) Consistent Robustness Improvements. For example, on NBA using GCN, $\Delta\text{Acc}$ decreases from $26.11\%$ (Vanilla) to $1.73\%$ (DICT), which demonstrates DICT's ability to stabilize predictions and highlights the effectiveness of the Wasserstein DICT formulation in controlling distributional shifts. (2) Consistent Fairness Improvements. On German with GCN, $\Delta_{\text{SP}}$ is reduced from $7.04\%$ to $1.08\%$, and $\Delta_{\text{EO}}$ from $7.47\%$ to $0.63\%$. This joint improvement supports the hypothesis that robustness and fairness can be optimized simultaneously under the distributional uncertainty framework. (3) Robustness and Fairness Do Not Conflict. Notably, clean accuracy is slightly improved (e.g., $+5.56\%$ on Bail with GraphSAGE), suggesting that our DICT objective does not trade accuracy for trustworthiness, but instead encourages better generalization by controlling local Lipschitz sensitivity. The joint improvement of robustness and fairness verifies that these objectives can be optimized simultaneously under the distribution uncertainty framework. (4) Adaptation across Architectures. The improvements of DICT hold across all four GNNs architecture (GCN, GraphSAGE, GIN, GAT), showing that it is architecture-agnostic and can be readily integrated with diverse GNN designs.

**Comparison with Other Models (RQ2).** We compare DICT with state-of-the-art baselines on both robustness and fairness benchmarks. Figure 2a and Figure 2b report the results in terms of $\Delta\text{Acc}$ (robustness) and $\Delta_{\text{EO}}$ (fairness), respectively. On all three citation networks (Cora, Citeseer, PubMed), DICT consistently achieves the lowest robustness gap $\Delta\text{Acc}$ among all baselines. For

robustness evaluation, we perform structure perturbations with a perturbation rate of 40% using random edge insertion and deletion. On fairness datasets (German, Bail, Credit), DICT outperforms prior methods in terms of group fairness. This demonstrates that DICT is more effective in mitigating disparities across sensitive groups under real-world graph distributions. Unlike prior methods that handle robustness and fairness separately, DICT unifies both objectives by modeling adversarial perturbations and group disparities as distributional shifts within a single framework.

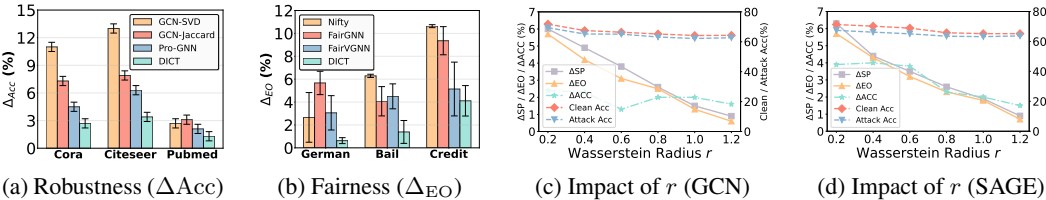

(a) Robustness ($\Delta$Acc)   (b) Fairness ($\Delta_{\text{EO}}$)   (c) Impact of $r$ (GCN)   (d) Impact of $r$ (SAGE)

Figure 2: Comparison against baselines (a–b) and sensitivity to Wasserstein radius $r$ (c–d).

**Hyper-parameter Analysis (RQ3).** We examine the effect of the Wasserstein radius $r$ and transport cost weights $\kappa = \{\kappa_A, \kappa_X, \kappa_y, \kappa_s\}$ on model performance using the German dataset. As shown in Figure 2 (c–d), increasing $r$ generally improves both robustness and fairness under both GCN and SAGE backbones, until a saturation point. When $r$ exceeds 1.0, the marginal improvements diminish, indicating that excessive regularization provides limited additional benefit and may even slow down training (see Section I.3). In the ablation study of transport cost weights shown in Figure 3, $\kappa_A$ and $\kappa_X$ are found to primarily influence robustness, while $\kappa_s$ has the strongest effect on fairness. When a modality-specific transportation cost weight (e.g., $\kappa_A$) is set to zero, the DICT framework imposes no penalty on perturbations in that modality. For instance, $\kappa_A = 0$ allows arbitrary structure perturbations within the Wasserstein ball, revealing the model's vulnerability to topological shifts. Notably, excessively large $\kappa_s$ can degrade fairness, suggesting that each hyper-parameter exhibits an effective range. These findings demonstrate that robustness and fairness do not improve indefinitely with larger values but can be jointly optimized by balancing $r$ and $\kappa$ within appropriate ranges.

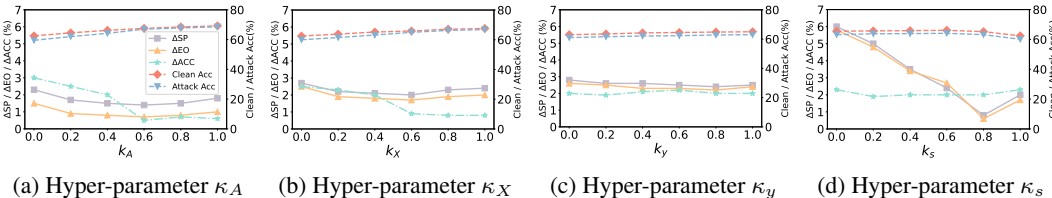

(a) Hyper-parameter $\kappa_A$   (b) Hyper-parameter $\kappa_X$   (c) Hyper-parameter $\kappa_y$   (d) Hyper-parameter $\kappa_s$

Figure 3: Impact of $\kappa$ on fairness and robustness.

## 6 CONCLUSION

In this paper, we introduce DICT, a framework for trustworthy graph learning, which constructs Wasserstein uncertainty sets over graphs to jointly model perturbation-induced uncertainty. Furthermore, we leverage strong duality to transform the infinite-dimensional supremum problem into a tractable Lipschitz-regularized empirical loss, design scalable first- and second-order estimators to estimate the penalty term, and optimize the resulting min-max objective function via truncated projected SGD. Across diverse datasets and GNN backbones, DICT outperforms prior methods by simultaneously enhancing robustness and fairness, thereby validating the strength of its distributionally robust design. More broadly, our theoretical insights and the DICT framework offer a well-defined objective, a feasible dual solution path, and a complete recipe for trustworthy graph learning. Looking ahead, our formulation can be extended to other trustworthiness dimensions such as safety and privacy by adjusting the loss, uncertainty set, perturbations and transportation cost. We hope DICT lay both theoretical and empirical foundations for future research in the trustworthy graph learning community.

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

# A    RELATED WORK

## A.1    ROBUSTNESS IN GRAPH NEURAL NETWORKS

GNNs, as extensions of neural networks to graph-structured data, are inherently vulnerable to adversarial attacks. Due to message passing and graph topology, perturbations to structures, node attributes, or labels can significantly degrade performance. Adversarial attacks are typically categorized as *white-box* or *black-box*, depending on whether the attacker has access to model parameters Xu et al. (2019a); Dai et al. (2018). Existing defenses mainly target structural and feature-level perturbations. For instance, GCN-JACCARD Wu et al. (2019) removes edges between nodes with low feature similarity, while attention-based defenses Tang et al. (2020); Zhang & Zitnik (2020) downweight adversarial nodes or edges during aggregation.Label noise, which includes label flipping, also poses serious challenges. Early approaches like NRGNN Dai et al. (2021) mitigate this by linking unlabeled nodes with pseudo-labeled neighbors, improving resilience to label corruption Dai & Wang (2021b); Li et al. (2021b).

However, most existing robustness guarantees are limited to discrete perturbation sets under bounded budgets, typically defined as $\|A - \hat{A}\| + \|X - \hat{X}\| \leq \Delta$ Xu et al. (2019a). In contrast, DICT leverages *Wasserstein-based distributional distance* $\delta_W(\mathbb{P}, \mathbb{Q})$ to model robustness against *continuous distributional shifts*, providing stronger and more generalizable guarantees.

## A.2    FAIRNESS IN GRAPH NEURAL NETWORKS

Fairness is a key component of trustworthy GNNs, especially in sensitive domains like recommendation and social analysis. Societal biases in training data can propagate and amplify through graph structures and message passing, leading to discriminatory predictions Dai & Wang (2021a); Wang & Leskovec (2020). To mitigate such issues, group fairness methods typically intervene at the feature, structure, or hybrid level. Feature-based approaches suppress sensitive information using adversarial learning Bose & Hamilton (2019), mask generators Wang et al. (2022), or fairness-aware data augmentation Yang et al. (2024). Structure-based methods modify the graph by dropping, reweighting Dong et al. (2022b); Spinelli et al. (2021), or adding edges Dong et al. (2022a); Ling et al. (2023), though the latter incurs $\mathcal{O}(n^2)$ cost. Hybrid approaches jointly adjust features and topology to promote fairness across modalities Dong et al. (2022a); Kose & Shen (2022).

Despite progress, balancing fairness with model accuracy and scalability remains challenging. Fairness constraints often incur performance trade-offs or optimization overhead, especially when sensitive attributes correlate with predictive features. Designing fair GNNs that remain accurate and robust under distribution shifts remains an open problem.

## A.3    DISTRIBUTIONALLY ROBUST OPTIMIZATION

Distributionally Robust Optimization (DRO) offers a principled framework for decision-making under uncertainty by minimizing the worst-case expected loss over an *uncertainty set* of distributions near the empirical data, typically defined via KL-divergence Hu & Hong (2013); Wu et al. (2023), Wasserstein distance Sinha et al. (2018), or MMD Staib & Jegelka (2019). DRO has further been employed to improve fairness in trustworthy learning, spanning adversarial debiasing with ROAD Grari et al. (2024), distributional robustness under noisy protected groups Wang et al. (2020), and sample-robust optimization techniques that enhance fairness generalization Ferry et al. (2023). Recent work has applied DRO to graph learning for improved robustness and fairness. For instance, Sadeghi et al. (2021) constructs Wasserstein uncertainty sets over node features to enhance GNNs stability, while DR-GNN Wang et al. (2024) and DRGO Zhao et al. (2025) incorporate DRO with smoothing and denoising to improve OOD generalization. In fairness, group DRO models Sagawa et al. (2019) minimize worst-case loss, and others extend DRO with fairness penalties Taskesen et al. (2020).

However, existing approaches do not consider combinations of multiple perturbation types. This prevents them from capturing the interactions among structural, feature, label perturbations, which often occur in real-world settings. In addition, current formulations are typically tailored to a single trustworthiness objective (e.g., robustness or fairness) and lack a unified mechanism for simultaneously addressing robustness, fairness, and privacy. These limitations motivate our DICT framework, which integrates multiple trustworthiness goals within a unified Wasserstein-DRO formulation.

# B  AUXILIARY LEMMAS

This section contains a collection of results that are needed in the proofs, most of which are classical theorems in DRO. We first present a strong duality result for DRO problems with Wasserstein distance in a very general setting.

Let $\Xi$ be a Polish (separable complete metric) space with metric $d$. Let $\mathcal{B}(\Xi)$ denote the set of Borel measures on $\Xi$.

**Definition 1** (Growth Rate). *Define the Growth rate $\kappa$ of $\Psi$ as:*

$$\kappa := \inf \left\{ \lambda \geq 0 : \int_{\Xi} \Psi(\lambda, \zeta) \nu(d\zeta) > -\infty \right\}.$$

*Particularly, if $\int_{\Xi} \Psi(\lambda, \zeta) \nu(d\zeta) = -\infty$ for all $\lambda \geq 0$, then $\kappa = \infty$.*

**Lemma 1** (Theorem 1 in Gao & Kleywegt (2023)). *Consider any $p \in [1, \infty)$, any $\nu \in \mathcal{P}(\Xi)$, any $\theta > 0$, and any $\Psi \in L^1(\nu)$ such that $\kappa < \infty$. Then:*

$$\sup_{\mu \in \mathcal{B}(\Xi)} \left\{ \mathbb{E}_{\mu}[\Psi(x, \xi)] : W_p(\mu, \nu) \leq \theta \right\}$$

$$= \min_{\lambda \geq 0} \left\{ \lambda \theta^p - \int_{\Xi} \inf_{\xi \in \Xi} \left[ \lambda d^p(\xi, \zeta) - \Psi(x, \xi) \right] \nu(d\zeta) \right\}.$$

*holds for any Polish space $(\Xi, d)$ and measurable function $\Psi$.*

Since the cross-entropy loss with softmax output is convex in the predicted distribution, the strong duality conditions required by Lemma 1 are satisfied in our setting. To quantify worst–case perturbations we require the dual of the weighted ellipsoidal norm $\|\Delta\|_d$ in Equation (13). Throughout, we write $\| \cdot \|_{d*}$ to denote this dual norm.

**Definition 2** (Dual Norm). *For a normed space $(\mathcal{V}, \| \cdot \|)$ with inner product $\langle \cdot, \cdot \rangle$, the dual norm $\| \cdot \|_* : \mathcal{V} \to \mathbb{R}_{\geq 0}$ is:*

$$\|g\|_* = \sup_{\|v\| \leq 1} \langle g, v \rangle, \qquad g \in \mathcal{V}.$$

Given positive weights $\kappa_A, \kappa_X, \kappa_y$, define for $\Delta = (\Delta_A, \Delta_X, \Delta_y)$

$$\|\Delta\|_d = \kappa_A \|\Delta_A\|_F^2 + \kappa_X \|\Delta_X\|_2^2 + \kappa_y \|\Delta_y\|_2^2.$$

To quantify worst–case perturbations, we write $\|\cdot\|_{d*}$ for the dual of the weighted measure $\|\cdot\|_d$ in Equation (13), defined in general by:

$$\|g\|_{d*} = \sup_{\|\Delta\|_d \leq 1} \langle g, \Delta \rangle.$$

**Lemma 2** (Dual Norm of $\| \cdot \|_d$). *For any $g = (g_A, g_X, g_y)$ in the same product space,*

$$\|g\|_{d*} = \sup_{\|\Delta\|_d \leq 1} \langle g, \Delta \rangle = \sqrt{\frac{\|g_A\|_F^2}{\kappa_A} + \frac{\|g_X\|_2^2}{\kappa_X} + \frac{\|g_y\|_2^2}{\kappa_y}}.$$

*Proof.* To convert the weighted ellipsoid constraint into a unit Euclidean ball, we apply a blockwise re-parameterisation. Define the linear mapping:

$$\mathbf{W} = \mathrm{diag}\left( \sqrt{\kappa_A}\mathbf{I}, \ \sqrt{\kappa_X}\mathbf{I}, \ \sqrt{\kappa_y}\mathbf{I} \right), \quad \text{and let} \quad z = \mathbf{W}\Delta.$$

Then $\Delta = \mathbf{W}^{-1}z$, and we compute

$$\|\Delta\|_d = \|\mathbf{W}^{-1}z\|_d = \|z_A\|_F^2 + \|z_X\|_2^2 + \|z_y\|_2^2 = \|z\|_2^2.$$

Hence, under this re-parameterisation, the feasible set $\{\|\Delta\|_d \leq 1\}$ becomes the unit ball $\{z : \|z\|_2 \leq 1\}$.

We now maximise the transformed objective:

$$\langle g, \Delta \rangle = \langle g, \mathbf{W}^{-1}z \rangle = \langle \mathbf{W}^{-\top}g, z \rangle,$$

over $\|z\|_2 \leq 1$, where $\mathbf{W}^{-\top} := (\mathbf{W}^{-1})^{\top}$. This is a standard constrained linear maximisation over the $\ell_2$ ball, which we solve via the Lagrangian:

$$\mathcal{L}(z, \lambda) = \langle \mathbf{W}^{-\top} g, z \rangle + \lambda(1 - \|z\|_2^2), \quad \lambda \geq 0.$$

Setting the gradient with respect to $z$ to zero yields the unique maximiser: $z^* = \frac{1}{2\lambda} \mathbf{W}^{-\top} g$. Imposing the active constraint $\|z^*\|_2 = 1$, we obtain:

$$1 = \|z^*\|_2^2 = \frac{1}{4\lambda^2} \|\mathbf{W}^{-\top} g\|_2^2 \quad \Rightarrow \quad 4\lambda^2 = \|\mathbf{W}^{-\top} g\|_2^2.$$

Finally, evaluating the optimal value gives:

$$\sup_{\|\Delta\|_d \leq 1} \langle g, \Delta \rangle = \langle \mathbf{W}^{-\top} g, z^* \rangle = \frac{1}{2\lambda} \|\mathbf{W}^{-\top} g\|_2^2 = \sqrt{\frac{\|g_A\|_F^2}{\kappa_A} + \frac{\|g_X\|_2^2}{\kappa_X} + \frac{\|g_y\|_2^2}{\kappa_y}},$$

as claimed. $\qquad\square$

## C  PROOF FOR MAIN RESULTS

In this section, we formally establish the DICT framework for trustworthy graph learning under uncertainty. We first introduce the dual formulation of the DICT objective, and by customizing the uncertainty set $\mathcal{P}$, transportation cost $c$, and task loss $\mathcal{L}$, it can flexibly accommodate diverse trustworthiness dimensions under a DICT optimization framework. We then solve the resulting min–max problem by leveraging local Lipschitz continuity, yielding an equivalent regularized objective. Finally, we develop efficient first- and second-order approximation methods to estimate worst-case perturbation effects for both labeled and unlabeled nodes.

### C.1  PROOF FOR THE DUAL FORMULATION

In this section, we prove the dual formulation (Theorem 1) of DICT (Equation (3)).

**Theorem 1** (Strong Dual Formulation of the DICT Objective) Consider the primal DICT objective in Equation (3), then under the strong duality conditions (see Lemma 1 in Appendix B), it admits the following equivalent dual formulation:

$$\min_{\theta \in \Theta} \inf_{\lambda \geq 0} \left\{ \lambda r - \mathbb{E}_{(\widehat{A}, \widehat{X}, \widehat{\mathbf{y}}) \sim \mathbb{P}_{\text{tr}}} \left[ \Phi_\lambda(\widehat{A}, \widehat{X}, \widehat{\mathbf{y}}; \theta) \right] \right\},$$

where the inner infimum is given by:

$$\mathcal{P}(\mathbb{P}_{\text{tr}}, r) = \left\{ \delta_{(A+\Delta_A, X+\Delta_X, y)} \,\middle|\, \kappa_A \|\Delta_A\|_F^2 + \kappa_X \|\Delta_X\|_F^2 \leq r \right\}$$

The cost function $c(\cdot, \cdot)$ defines a ground metric over the joint graph-label space $\mathcal{Z} = \mathcal{G} \times \mathcal{Y}$, specified as:

$$c\big((A, X, \mathbf{y}), (\widehat{A}, \widehat{X}, \widehat{\mathbf{y}})\big) = \kappa_A \|A - \widehat{A}\|_F^2 + \kappa_X \|X - \widehat{X}\|_F^2 + \kappa_y \|\mathbf{y} - \widehat{\mathbf{y}}\|_2^2,$$

where $\|\cdot\|_F$ denotes the Frobenius norm, $\|\cdot\|_2$ denotes the Euclidean norm, and the coefficients $(\kappa_A, \kappa_X, \kappa_y)$ reflect the relative confidence in the alignment between an instance and its perturbations in terms of structure, features, and labels.

*Proof.* We begin with the general DRO formulation over a Wasserstein ball:

$$\min_{\theta \in \Theta} \sup_{\mathbb{P} \in \mathcal{P}(\mathbb{P}_{\text{tr}}, r)} \mathbb{E}_{(A, X, \mathbf{y}) \sim \mathbb{P}} \left[ \mathcal{L}(f(A, X; \theta), \mathbf{y}) \right].$$

Introducing a Lagrange multiplier $\lambda \geq 0$ for the constraint $\delta_W(\mathbb{P}, \mathbb{P}_{\text{tr}}) \leq r$, the relaxed Lagrangian is:

$$\Lambda(\mathbb{P}, \lambda) = \mathbb{E}_{\mathbb{P}}[\mathcal{L}] - \lambda \left( \delta_W(\mathbb{P}, \mathbb{P}_{\text{tr}}) - r \right).$$

Applying strong duality (Lemma 1) yields:

$$\sup_{\mathbb{P} \in \mathcal{P}(\mathbb{P}_{\text{tr}}, r)} \mathbb{E}_{\mathbb{P}}[\mathcal{L}] = \inf_{\lambda \geq 0} \sup_{\mathbb{P}} \Lambda(\mathbb{P}, \lambda).$$

Using the Kantorovich duality of Wasserstein DRO Gao & Kleywegt (2023), the inner supremum can be rewritten as:

$$\lambda r - \mathbb{E}_{(\widehat{A}, \widehat{X}, \widehat{\mathbf{y}}) \sim \mathbb{P}_{\mathrm{tr}}} \left[ \inf_{(A, X, \mathbf{y})} \left\{ \lambda c - \mathcal{L}(f(A, X; \theta), \mathbf{y}) \right\} \right].$$

Substituting this back into the outer objective gives the dual formulation in Equation (4), where the inner minimization is denoted as $\Phi_\lambda(\cdot)$.

In our instantiation, the loss aggregates prediction errors only over observed nodes $\mathcal{N}_o$. $\qquad \square$

Then we present a more general optimization objective that explicitly accounts for fairness.

**Proposition 3.** *Let the general DRO objective be defined as:*

$$\min_{\theta \in \Theta} \sup_{\mathbb{P} \in \mathcal{P}(\mathbb{P}_{\mathrm{tr}}, r)} \mathbb{E}_{(A, X, \mathbf{y}, \mathbf{s}) \sim \mathbb{P}} \left[ \mathcal{L}(f(A, X; \theta), \mathbf{y}) \right],$$

*with the Wasserstein transportation cost:*

$$c((A, X, \mathbf{y}, \mathbf{s}), (\widehat{A}, \widehat{X}, \hat{\mathbf{y}}, \hat{\mathbf{s}})) = \kappa_A \|A - \widehat{A}\|_F^2 + \kappa_X \|X - \widehat{X}\|_F^2 + \kappa_s \|\mathbf{s} - \hat{\mathbf{s}}\|_2^2 + \kappa_y \|\mathbf{y} - \hat{\mathbf{y}}\|_2^2.$$

*Then:*

1. *Setting $\kappa_s = 0$, the DRO objective reduces to the robustness-aware formulation in Equation (7).*

2. *Setting $\kappa_s > 0$, the DRO objective reduces to the fairness-aware formulation in Equation (10).*

### C.2 PROOF FOR THE LIPSCHITZ-REGULARIZED OBJECTIVE

In this section, we derive the equivalent objective of DICT optimization in Theorem 2.

**Theorem 2** (Equivalent Lipschitz-Regularized Objective) Under the assumptions of strong duality and local Lipschitz continuity of the loss function $\mathcal{L}$, the DICT problem Equation (3) is equivalent to the following optimization problem:

$$\min_{\theta \in \Theta} \mathbb{E}_{(A, X, \mathbf{y}) \sim \mathbb{P}_{\mathrm{tr}}} \left[ \mathcal{L}(f(A, X; \theta), \mathbf{y}) + L(\theta) \sqrt{r} \right],$$

$$\text{s.t.} \quad L(\theta) = \max_{\|\Delta\|_d \leq 1} \left| \mathcal{L}(f(A + \Delta_A, X + \Delta_X; \theta), \mathbf{y} + \Delta_{\mathbf{y}}) - \mathcal{L}(f(A, X; \theta), \mathbf{y}) \right|,$$

where $L(\theta)$ denotes the local Lipschitz constant of the loss $\mathcal{L}$ with respect to perturbations $\Delta = (\Delta_A, \Delta_X, \Delta_y) \in \mathcal{Z}$. The perturbation norm is $\|\cdot\|_d$ defined as $\|\Delta\|_d := \kappa_A \|\Delta_A\|_F^2 + \kappa_X \|\Delta_X\|_2^2 + \kappa_y \|\Delta_y\|_2^2$.

*Proof.* The original DRO objective under a Wasserstein uncertainty set can be equivalently reformulated, via Kantorovich duality, as:

$$\min_{\theta \in \Theta} \inf_{\lambda \geq 0} \left\{ \lambda r + \mathbb{E}_{(\widehat{A}, \widehat{X}, \widehat{\mathbf{y}}) \sim \mathbb{P}_{\mathrm{tr}}} \left[ \Phi_\lambda(\widehat{A}, \widehat{X}, \widehat{\mathbf{y}}; \theta) \right] \right\}, \tag{12}$$

where

$$\Phi_\lambda(\widehat{A}, \widehat{X}, \widehat{\mathbf{y}}; \theta) = \sup_{(A, X, \mathbf{y})} \left[ \mathcal{L}(f(A, X; \theta), \mathbf{y}) - \lambda c((A, X, \mathbf{y}), (\widehat{A}, \widehat{X}, \widehat{\mathbf{y}})) \right].$$

We parameterize perturbations around each training sample as:

$$A = \widehat{A} + \Delta_A, \quad X = \widehat{X} + \Delta_X, \quad \mathbf{y} = \widehat{\mathbf{y}} + \Delta_y.$$

The perturbation cost becomes:

$$d = \|\Delta\|_d = \sqrt{\kappa_A \|\Delta_A\|_F^2 + \kappa_X \|\Delta_X\|_F^2 + \kappa_y \|\Delta_y\|_2^2}. \tag{13}$$

So the dual potential simplifies to:

$$\Phi_\lambda = \sup_\Delta \left[ \mathcal{L}\big(f(\widehat{A} + \Delta_A, \widehat{X} + \Delta_X; \theta), \widehat{\mathbf{y}} + \Delta_y\big) - \lambda c \right].$$

Therefore, $(\Delta_A, \Delta_X, \Delta_y)$ is the direction that leads to the largest change on the model prediction of $X$.

Assuming $\mathcal{L}$ is locally $L(\theta)$-Lipschitz under the perturbation metric $c(\cdot, \cdot)$, we obtain:

$$\mathcal{L}(f(A, X; \theta), \mathbf{y}) \leq \mathcal{L}(f(\widehat{A}, \widehat{X}; \theta), \widehat{\mathbf{y}}) + L(\theta)\sqrt{c},$$

for $\forall (\widehat{A}, \widehat{X}, \widehat{\mathbf{y}}) \sim \mathbb{P}_{\mathrm{tr}}$. Therefore,

$$\Phi_\lambda \leq \sup_{d \geq 0} \left[ \mathcal{L}(f(\widehat{A}, \widehat{X}; \theta), \widehat{\mathbf{y}}) + L(\theta)\sqrt{c} - \lambda c \right].$$

Applying Young's inequality: $L(\theta)\sqrt{c} - \lambda c \leq \frac{L(\theta)^2}{4\lambda}$, we get:

$$\Phi_\lambda(\widehat{A}, \widehat{X}; \theta) \leq \mathcal{L}(f(\widehat{A}, \widehat{X}; \theta), \widehat{\mathbf{y}}) + \frac{L(\theta)^2}{4\lambda}.$$

Note that although $L(\theta)$ is Lipschitz w.r.t. Euclidean perturbations (i.e., $\ell_2$ norm), our metric $c$ accumulates *squared* perturbations, so the effective perturbation norm is $\sqrt{c}$. This explains the appearance of $L(\theta)\sqrt{c}$ rather than $L(\theta)c$ in the upper bound.

In another direction, to establish equivalence, we must prove the reverse inequality. Recall that $L(\theta)$ is defined as the infimum of all constants $L > 0$ satisfying $|\mathcal{L}(f(A, X; \theta), \mathbf{y}) - \mathcal{L}(f(\widehat{A}, \widehat{X}; \theta), \widehat{\mathbf{y}})| \leq L\sqrt{c}$ for every perturbation with cost $d$. Hence, for any $\varepsilon > 0$ there exists a unit perturbation $\Delta_\varepsilon$ with $\|\Delta_\varepsilon\|_d \leq 1$ such that:

$$|\mathcal{L}(f(\widehat{A} + \Delta_{\varepsilon,A}, \widehat{X} + \Delta_{\varepsilon,X}; \theta), \widehat{\mathbf{y}} + \Delta_{\varepsilon,y}) - \mathcal{L}(f(\widehat{A}, \widehat{X}; \theta), \widehat{\mathbf{y}})| \geq L(\theta) - \varepsilon.$$

Let $t \geq 0$ and scale this perturbation by $t$: $\widetilde{\Delta} = t\Delta_\varepsilon$ so that $d = t^2$. Then:

$$\Phi_\lambda(\widehat{A}, \widehat{X}; \theta) \geq \sup_{t \geq 0} \left[ \mathcal{L}(f(\widehat{A}, \widehat{X}; \theta), \widehat{\mathbf{y}}) + (L(\theta) - \varepsilon)t - \lambda t^2 \right].$$

The quadratic in $t$ is maximised at $t^\star = \frac{L(\theta) - \varepsilon}{2\lambda}$, yielding:

$$\Phi_\lambda(\widehat{A}, \widehat{X}; \theta) \geq \mathcal{L}(f(\widehat{A}, \widehat{X}; \theta), \widehat{\mathbf{y}}) + \frac{(L(\theta) - \varepsilon)^2}{4\lambda}.$$

Because $\varepsilon > 0$ is arbitrary, letting $\varepsilon \to 0$ gives:

$$\Phi_\lambda(\widehat{A}, \widehat{X}; \theta) \geq \mathcal{L}(f(\widehat{A}, \widehat{X}; \theta), \widehat{\mathbf{y}}) + \frac{L(\theta)^2}{4\lambda}.$$

Combining this lower bound with the upper bound proved earlier, we have:

$$\Phi_\lambda(\widehat{A}, \widehat{X}; \theta) = \mathcal{L}(f(\widehat{A}, \widehat{X}; \theta), \widehat{\mathbf{y}}) + \frac{L(\theta)^2}{4\lambda},$$

so the inequality chain is now tight in both directions.

Substituting back into Eq. Equation (12), we obtain:

$$\min_{\theta \in \Theta, \lambda \geq 0} \left\{ \mathbb{E}_{(\widehat{A}, \widehat{X}, \widehat{\mathbf{y}}) \sim \mathbb{P}_{\mathrm{tr}}} \mathcal{L}(f(\widehat{A}, \widehat{X}; \theta), \widehat{\mathbf{y}}) + \lambda r + \frac{L(\theta)^2}{4\lambda} \right\}.$$

The closed-form minimizer of $\lambda$ is:

$$\lambda^\star(\theta) = \frac{L(\theta)}{2\sqrt{r}},$$

leading to the final solution:

$$\min_{\theta \in \Theta} \left\{ \mathbb{E}_{\mathbb{P}_{\mathrm{tr}}} \left[ \mathcal{L}(f(A, X; \theta), \mathbf{y}) + L(\theta)\sqrt{r} \right] \right\},$$

$$\text{s.t.} \quad L(\theta) = \max_{\|\Delta\|_d \leq 1} \left| \mathcal{L}\big(f(A + \Delta_A, X + \Delta_X; \theta), \mathbf{y} + \Delta_y\big) - \mathcal{L}(f(A, X; \theta), \mathbf{y}) \right|.$$

$\square$

## C.3 PROOF FOR APPROXIMATION

For an unlabeled node $n \notin \mathcal{N}_o$, the first–order gradient of the loss vanishes at the prediction point, i.e. $\nabla_{x_n} \mathcal{L}(f(A, x_n; \theta), \hat{y}_n) = 0$, with $\hat{y}_n := f(A, x_n; \theta)$. Consequently, the local worst–case increment is governed by the second–order term under the joint perturbation metric:

$$\|\Delta\|_d := \kappa_A \|\Delta_A\|_F^2 + \kappa_X \|\Delta_X\|_2^2 + \kappa_y \|\Delta_y\|_2^2.$$

**Lemma 3** (Second–Order Upper Bound for Unlabeled Nodes). *Let the perturbation vector associated with node $n$ be:*

$$\Delta_n := (\Delta_A, \ \Delta_{x_n}, \ \Delta_{y_n}) \in \mathbb{R}^{N+D+1}.$$

*Define the block Hessian:*

$$H_n := \nabla^2_{\Delta_n \Delta_n} \mathcal{L}(f(A, x_n; \theta), y_n)\Big|_{\Delta=0}.$$

*Then,*

$$L_n(\theta) := \sup_{\substack{\|\Delta\|_d \leq 1 \\ \Delta_{\bar{n}}=0}} \left| \mathcal{L}\big(f(A + \Delta_A, x_n + \Delta_{x_n}; \theta), y_n + \Delta_{y_n}\big) - \mathcal{L}\big(f(A, x_n; \theta), y_n\big) \right| = \frac{1}{2} \lambda_{\max}(\widetilde{H}_n),$$

*where the rescaled Hessian is:*

$$\widetilde{H}_n := \mathbf{W}^{-\top} H_n \mathbf{W}^{-1},$$

*where $\mathbf{W} := \mathrm{diag}\left( \frac{1}{\sqrt{2}} \sqrt{\kappa_A} \mathbf{I}, \ \frac{1}{\sqrt{2}} \sqrt{\kappa_X} \mathbf{I}, \ \frac{1}{\sqrt{2}} \sqrt{\kappa_y} \mathbf{I} \right).$*

*Proof.* The constraint $\|\Delta\|_d \leq 1$ under $\Delta_{\bar{n}} = 0$ reduces to an ellipsoidal norm constraint on $\Delta_n$. Define the reparameterization $z := \mathbf{W}\Delta_n$, so that $\|\Delta_n\|_d \leq 1 \iff \|z\|_2 \leq 1$. Using the second–order Taylor expansion:

$$\mathcal{L}(\Delta_n) = \mathcal{L}(0) + \tfrac{1}{2}\Delta_n^\top H_n \Delta_n + o(\|\Delta_n\|^2) = \mathcal{L}(0) + \tfrac{1}{2} z^\top \widetilde{H}_n z.$$

Maximizing this quadratic form over $\|z\|_2 \leq 1$ yields $\frac{1}{2}\lambda_{\max}(\widetilde{H}_n)$. $\square$

**Lemma 4** (Efficient Spectral Estimate via Power Iteration). *To approximate the spectral norm without forming $\widetilde{H}_n$ explicitly, we apply power iteration. Let $z^{(0)} \sim \mathcal{N}(0, I)$, and iterate:*

$$v^{(t)} := \widetilde{H}_n z^{(t)}, \qquad z^{(t+1)} := \frac{v^{(t)}}{\|v^{(t)}\|_2}, \quad t = 0, \ldots, T-1.$$

*Then,*

$$L_n(\theta) \approx \tfrac{1}{2}(z^{(T)})^\top \widetilde{H}_n z^{(T)}.$$

*Each iteration requires one Hessian–vector product (e.g., via "torch.autograd.functional.hvp").*

When full power iteration is computationally expensive, we adopt a stochastic surrogate by sampling a single Gaussian probe in the perturbation space.

We sample a random probe $\xi := (\Delta_A, \Delta_{x_n}, \Delta_{y_n})$ from the standard normal distribution under the metric-induced parameterization:

$$\xi \sim \mathcal{N}(0, I), \quad \Delta_n = \mathbf{W}^{-1} \xi,$$

where $\mathbf{W}$ is the block–diagonal scaling operator defined in Lemma 3, and each component of $\Delta_n$ is understood to have support restricted to node $n$.

We then approximate the worst-case loss increment via the directional derivative in the sampled direction:

$$\mathrm{L}_n^{\mathrm{unlab}}(\theta) = \left\| \nabla_\xi \mathcal{L}\big(f(A + \xi_A, x_n + \xi_{x_n}; \theta), \ y_n + \xi_{y_n}\big) \right\|_{d^*},$$

where the dual norm $\|\cdot\|_{d^*}$ is defined in Lemma 2. This estimator requires only a single backward pass and is highly efficient in practice.

# D    PERTURBATION MODELS AND EXTENSION

In this section, we present perturbation models for simulating perturbations in graph-structured data. Additionally, we extend our fairness evaluation to handle multi-category sensitive attributes, broadening the applicability of our DICT framework to more realistic scenarios.

## D.1    PERTURBATION MODELS

We adopt a graph-centric perturbation scheme to simulate distribution shifts in GNNs inputs. Specifically, we inject noise into the topology $A$, node features $X$, node labels $\mathbf{y}$, and sensitive attributes $\mathbf{s}$ through the following mechanisms:

**Topology Noise.**    We perturb the adjacency matrix $A$ by randomly dropping and re-adding edges with probability $\epsilon_e$, while approximately preserving node degrees:

$$\widehat{A}_{ij} = \begin{cases} 0 & \text{if } A_{ij} = 1 \text{ and } \text{Random}(0,1) \leq \epsilon_e \\ 1 & \text{if } A_{ij} = 0 \text{ and } |\widehat{\Delta}_i| \approx |\Delta_i| \\ A_{ij} & \text{otherwise} \end{cases}$$

The overall perturbation intensity is quantified by the edge-change ratio $\epsilon_e = \|A - \widehat{A}\|_1 / \|A\|_1$.

**Attribute Noise.**    For each feature dimension $j$, we inject Gaussian noise proportional to the empirical standard deviation $\sigma_j$:

$$\widehat{X}_{:,j} = X_{:,j} + \epsilon_x \cdot \mathcal{N}(0, \sigma_j^2),$$

where $\epsilon_x$ controls the noise magnitude.

**Sensitive Attribute Noise.**    We flip a fraction $\gamma_{\max} \in [0,1]$ of sensitive attribute values, either uniformly or in a group-balanced fashion:

$$\widehat{s}_i = \begin{cases} 1 - s_i & \text{if } i \in \mathcal{I}_{\text{flip}} \\ s_i & \text{otherwise} \end{cases}$$

Here, $\mathcal{I}_{\text{flip}}$ is a random index set with cardinality $\lfloor \gamma_{\max} \cdot N \rfloor$, optionally stratified across sensitive groups.

**Label Noise.**    With probability $\epsilon_l$, we randomly flip each node's label to a different class in the label space $\mathcal{Y}$:

$$\widehat{y}_i = \begin{cases} y_l \neq y_i & \text{if } \text{Random}(0,1) \leq \epsilon_l \\ y_i & \text{otherwise} \end{cases}$$

This framework allows systematic injection of noise across multiple modalities, facilitating the evaluation of robustness and fairness under controlled perturbation regimes.

## D.2    EXTENSION TO MULTICLASS AND MULTICATEGORY SENSITIVE ATTRIBUTES

Here, we show how our proposed DICT framework can be extended to handle multiclass and multi-category sensitive attribute problems. According to Locatello et al. (2019), let $y \in \{y_1, \ldots, y_c\}$ and $s \in \{s_1, \ldots, s_k\}$ represent the multi-class label and multi-category sensitive attribute, where $c$ is the number of classes and $k$ is the number of sensitive attribute categories, the evaluation metric can be extended to:

$$\Delta_{\text{SP}} = \frac{1}{k} \sum_{i=1}^{k} \max_{y_j} |P(\hat{y} = y_j) - P(\hat{y} = y_j \mid a = a_i)|.$$

Similarly, for equalized odds (EO), we compute $\Delta_{\text{EO}}$ across all sensitive group pairs and condition on ground-truth labels.

To encode group discrepancies into the DICT objective, we extend the transportation cost $c(\cdot, \cdot)$ in Eq. Equation (5) to account for categorical shifts in $\mathbf{s}$:

$$c((A, X, \mathbf{y}, \mathbf{s}), (\widehat{A}, \widehat{X}, \widehat{\mathbf{y}}, \widehat{\mathbf{s}})) = \kappa_A |A - \widehat{A}|_F^2 + \kappa_X |X - \widehat{X}|_F^2 + \kappa_y |\mathbf{y} - \widehat{\mathbf{y}}|_2^2 + \kappa_s \cdot \mathbf{1}[\mathbf{s} \neq \widehat{\mathbf{s}}], \tag{14}$$

where $\mathbf{1}[\cdot]$ is the indicator function. This instantiation treats any change in sensitive group membership as a unit perturbation, preserving categorical semantics.

Under this setting, we adopt a modified adversarial fairness loss that promotes prediction invariance under all group perturbations:

$$\mathcal{L}_{\text{adv-fair}} = \frac{\sum_{i \neq j} \sum_{n:s_n=i} d(f(A, x_n, s_n = i; \theta), f(A, x_n, s_n = j; \theta))}{Nk(k-1)}, \tag{15}$$

where $d(\cdot, \cdot)$ is a divergence metric (e.g., $\ell_2$, KL). This encourages group-invariant predictions.

We then instantiate the fairness-aware DICT objective as:

$$\min_{\theta \in \Theta} \sup_{\mathbb{P} \in \mathcal{P}(\mathbb{P}_{\text{tr}}, r)} \mathbb{E}_{(A, X, \mathbf{y}, \mathbf{s}) \sim \mathbb{P}} \left[ \mathcal{L}_{\text{train}}(f(A, X; \theta), \mathbf{y}) + \gamma \mathcal{L}_{\text{adv-fair}} \right], \tag{16}$$

which is compatible with the Lipschitz-regularized formulations in Theorem 2. The local sensitivity $L_n(\theta)$ is estimated analogously, with $\nabla_{s_n} \mathcal{L}$ computed via one-hot encoding gradients and incorporated into the total regularization.

# E    GENERALIZATION BOUNDS FOR DICT

In this section, we provide a theoretical analysis to demonstrate the robustness of DICT to distribution shift. Recall that $\mathbb{P}_{\text{real}}$ represents the underlying joint distribution, and $\mathbb{P}_{\text{tr}}$ denotes the mixed empirical distribution. Let $K$ denote the number of perturbed graphs in $\mathbb{P}_{\text{tr}}$.

## E.1    PRELIMINARIES

**Definition 3** (Empirical and Population Risks). *Given a loss function $\mathcal{L} : \mathcal{Y} \times \mathcal{Y} \to \mathbb{R}_+$, the expected loss of $\theta \in \Theta$ is defined as:*

$$R(\theta) := \mathbb{E}_{(A, X, \mathbf{y}) \sim \mathbb{P}_{\text{real}}} \big[ \mathcal{L}(f(A, X; \theta), \mathbf{y}) \big],$$

*and the empirical loss w.r.t. empirical distribution $\mathbb{P}_{\text{tr}}$ is defined as:*

$$\widehat{R}_S(\theta) := \mathbb{E}_{(A, X, \mathbf{y}) \sim \mathbb{P}_{\text{tr}}} [\mathcal{L}(f(A, X; \theta), \mathbf{y})].$$

**Assumption 1** (Bounded Per-Example Loss). *There exists a finite constant $B > 0$ such that, for every parameter $\theta \in \Theta$, graph $(A, X)$ and label $y$, the supervised loss obeys:*

$$0 \leq \mathcal{L}\big(f(A, X; \theta), \mathbf{y}\big) \leq B.$$

*In DICT we guarantee this by (i) projecting the network weights onto an $\ell_2$-ball of radius $R$ and (ii) clipping (or label-smoothing) the output logits to $[-R, R]$, which yields $B = -\log\big(\frac{1}{1+e^R}\big)$ for cross-entropy losses.*

This bounded-loss setting enables a clean McDiarmid concentration step; if clipping is undesirable, the same stability proof can be recast under sub-Gaussian tails via a Bernstein inequality, preserving the $\mathcal{O}\big((K + 1)^{-1/2}\big)$ rate at the expense of slightly larger constants. Our goal is to bound the *robust population risk* $R(\theta)$ of the parameter $\mathcal{A}(S)$ returned by the training algorithm $\mathcal{A}$ through the perturbation-space Lipschitz constant $L(\theta)$ and the Wasserstein radius $r$. To avoid any explicit dependence on weight norms, we adopt an **algorithmic-stability route** in the spirit of Hardt et al. (2016b).

**Empirical Objective.** For a mixed sample $S = \left\{ (A, X, \widehat{\mathbf{y}}), (A_k, X_k, \widehat{\mathbf{y}}_k) \right\}_{k=1}^{K}$ define:

$$F_S(\theta) := \widehat{R}_S(\theta) + L(\theta)\sqrt{r}. \tag{17}$$

Then $\mathcal{A}$ performs $T$ projected-SGD steps with *gradient clipping* radius $G_{\max}$:

$$g_t = \mathrm{clip}_{G_{\max}}\big(\nabla_\theta F_S(\theta_t)\big),$$
$$U(\theta_t) := \theta_{t+1} = \Pi_{\mathcal{B}_R}(\theta_t - \alpha_t g_t),$$

where $\Pi_{\mathcal{B}_R}$ projects onto an $\ell_2$-ball of radius $R^2$, and $U : \Theta \to \Theta$ is the update rule.

**Definition 4** (Uniform Stability Bousquet & Elisseeff (2002)). $\mathcal{A}$ is $\beta$-uniformly stable *if for all data sets $S, S'$ such that $S$ and $S'$ differ in at most one example, we have:*

$$\sup_{(A,X,y)} \left| \mathcal{L}\big(f(A, X; \mathcal{A}(S)), y\big) - \mathcal{L}\big(f(A, X; \mathcal{A}(S')), y\big) \right| \leq \beta.$$

**Lemma 5** (Generalization in Expectation). *Let $\mathcal{A}$ be $\beta$-uniformal stable. Then,*

$$|\mathbb{E}_S[\widehat{R}_S(\mathcal{A}(S)) - R(\mathcal{A}(S))| \leq \beta.$$

*Proof.* Denote by:

$$S = ((A, X, \mathbf{y}), (A_1, X_1, \mathbf{y}_1), \ldots, (A_K, X_K, \mathbf{y}_K))$$

and

$$S' = ((A', X', \mathbf{y}'), (A_1', X_1', \mathbf{y}_1'), \ldots, (A_K', X_K', \mathbf{y}_K'))$$

two independent random graph samples and let:

$$S^{(k)} = ((A, X, \mathbf{y}), (A_1, X_1, \mathbf{y}_1), \ldots, (A_k', X_k', \mathbf{y}_k'), \ldots, (A_K, X_K, \mathbf{y}_K))$$

be the sample that is identical to $S$ except in the $i$'th example where we replace $(A_k, X_k, \mathbf{y}_k)$ with $(A_k', X_k', \mathbf{y}_k')$. With this notation, we get that:

$$\mathbb{E}_S\big[\widehat{R}_S(\mathcal{A}(S))\big]$$

$$= \mathbb{E}_S\left[ \eta\mathcal{L}\big(f(A, X; \mathcal{A}(S)), \mathbf{y}\big) + \frac{1-\eta}{K} \sum_{k=1}^{K} \mathcal{L}\big(f(A_k, X_k; \mathcal{A}(S)), \mathbf{y}_k\big) \right]$$

$$= \mathbb{E}_S\mathbb{E}_{S'}\left[ \eta\mathcal{L}\big(f(A', X'; \mathcal{A}(S^{(k)})), \mathbf{y}'\big) + \frac{1-\eta}{K} \sum_{k=1}^{K} \mathcal{L}\big(f(A_k', X_k'; \mathcal{A}(S^{(k)})), \mathbf{y}_k'\big) \right] \tag{18}$$

$$= \mathbb{E}_S\mathbb{E}_{S'}\left[ \eta\mathcal{L}\big(f(A', X'; \mathcal{A}(S)), \mathbf{y}'\big) + \frac{1-\eta}{K} \sum_{k=1}^{K} \mathcal{L}\big(f(A_k', X_k'; \mathcal{A}(S)), \mathbf{y}_k'\big) \right] + \delta$$

$$= \mathbb{E}_S\big[R(\mathcal{A}(S))\big] + \delta.$$

Furthermore, taking the supremum over any two data sets $S, S'$ differing in only one sample, we can bound the difference as:

$$|\delta| \leq \sup_{S, S', A, X, \mathbf{y}} \mathcal{L}(f(A, X; \mathcal{A}(S)), \mathbf{y}) - \mathcal{L}(f(A, X; \mathcal{A}(S')), \mathbf{y}),$$

by our assumption on the uniform stability of $\mathcal{A}$. The claim follows. $\qquad\square$

Lemma 5 demonstrates that uniform stability is a sufficient condition for bounding the generalization gap. We now turn our attention to the structural characteristics of iterative update rules that determine the degree of stability attained during training.

To this end, we introduce two key properties of the update operator: expansiveness, which captures the sensitivity of the update to perturbations in its input, and boundedness, which constrains the movement of the iterate at each step. These properties enable us to analyze how deviations in the input—e.g., due to changes in the training set—propagate through the optimization trajectory.

---

[2] Clipping and projection are already standard in robust/fair GNNs training to prevent exploding gradients and weight drift; we merely exploit them for the analysis.

**Definition 5.** *An update rule $U$ is $\eta$-expansive if*

$$\sup_{\theta, \theta' \in \Theta} \frac{\|U(\theta) - U(\theta')\|}{\|\theta - \theta'\|} \leq \eta.$$

**Definition 6.** *An update rule $U$ is $\sigma$-bounded if*

$$\sup_{\theta \in \Theta} \|\theta - U(\theta)\| \leq \sigma.$$

Note that projection onto $\mathcal{B}_R$ is 1-*expansive*, while gradient clipping of radius $G_{\max}$ makes each step $\sigma_t$-bounded with $\sigma_t = \alpha_t G_{\max}$. Leveraging these two properties, we derive the following lemma characterizing how a sequence of model updates diverges under perturbations to the training set.

**Lemma 6** (Growth Recursion). *Fix an arbitrary sequence of updates $U_1, \ldots, U_T$ and another sequence $U'_1, \ldots, U'_T$. Let $\theta_0 = \theta'_0$ be a starting point in $\Theta$ and define $\delta_t = \|\theta'_t - \theta_t\|$ where $\theta_t, \theta'_t$ are defined recursively through:*

$$\theta_{t+1} = U(\theta_t), \qquad \theta'_{t+1} = U'(\theta'_t).$$

*Then, we have the recurrence relation:*

$$\delta_0 = 0$$

$$\delta_{t+1} \leq \begin{cases} \eta \delta_t & U_t = U'_t \text{ is } \eta\text{-expansive} \\ \min(\eta, 1)\delta_t + 2\sigma_t & U_t \text{ and } U'_t \text{ are } \sigma\text{-bounded} \\ & U_t \text{ is } \eta \text{ expansive} \end{cases}$$

*Proof.* The initial bound on $\delta_t$ follows immediately from the assumption that $U_t = U'_t$ together with the definition of the expansiveness property. To derive the second bound, we invoke Definition 6, which states that if $G_t$ and $G'_t$ are $\sigma$-bounded, then the triangle inequality yields:

$$\begin{aligned} \delta_{t+1} &= \|U(\theta_t) - U'(\theta'_t)\| \\ &\leq \|U(\theta_t) - \theta_t + \theta'_t - U'(\theta'_t)\| + \|\theta_t - \theta'_t\| \\ &\leq \delta_t + \|U(\theta_t) - \theta_t\| + \|U(\theta'_t) - \theta'_t\| \\ &\leq \delta_t + 2\sigma, \end{aligned}$$

which gives half of the second bound. We can alternatively bound $\delta_{t+1}$ as:

$$\begin{aligned} \delta_{t+1} &= \|U_t(\theta_t) - U'_t(\theta'_t)\| \\ &= \|U_t(\theta_t) - U_t(\theta'_t) + U_t(\theta'_t) - U'_t(\theta'_t)\| \\ &\leq \|U_t(\theta_t) - U_t(\theta'_t)\| + \|U_t(\theta'_t) - U'_t(\theta'_t)\| \\ &\leq \|U_t(\theta_t) - U_t(\theta'_t)\| + \|\theta'_t - U_t(\theta'_t)\| + \|\theta'_t - U'_t(\theta'_t)\| \\ &\leq \eta \delta_t + 2\sigma \,. \end{aligned}$$

$\square$

In order to prove that the stochastic gradient method is stable, we will analyze the output of the algorithm on two graph sets that differ in precisely one location. Recalling that $\theta_t$ is obtained from $\theta_{t-1}$ via a gradient update, our goal is to bound $\delta_t = \|\theta_t - \theta'_t\|$ recursively and in expectation as a function of $\delta_{t-1}$. There are two cases to consider. In the first case, $U$ selects the index of an example at step $t$ on which is identical in $S$ and $S'$. The mini-batch index sampled at step $t$ is the same in $S$ and $S'$. The second case to consider is when $U$ selects the one example to update in which $S$ and $S'$ differ. Note that this happens only with probability $\frac{1}{K+1}$ if examples are selected randomly. In this case, we simply bound the increase in $\delta_t$ by the norm of the two gradients $\nabla_\theta F_G(\theta_{t-1})$ and $\nabla_\theta F_{G'}(\theta'_{t-1})$. The sum of the norms is bounded by $2\alpha_t L$ and we obtain $\delta_t \leq \delta_{t-1} + 2\alpha_t L$, where we assume that $\mathcal{L}$ is $L$-Lipschitz w.r.t. $\theta$. Combining the two cases, we can then solve a simple recurrence relation to obtain a bound on $\delta_T$. Here, we need to use an intriguing stability property of stochastic gradient method. Specifically, the first time step $t_0$ at which SGM even encounters the example in which $S$ and $S'$ differ is a random variable in $\{1, \ldots, K\}$ which tends to be relatively large. Specifically, for any $m \in \{1, \ldots, K\}$, the probability that $t_0 \leq m$ is upper bounded by $\frac{m}{n}$. This allows us to argue that SGM has a long "burn-in period" where $\delta_t$ does not grow at all.

**Lemma 7** (Stability of DICT Optimiser). *Assume the per-step loss is bounded:* $0 \leq \mathcal{L} \leq B$. *Let* $\{\alpha_t\}_{t=0}^{T-1}$ *be the learning-rate schedule and set* $S_\alpha := \sum_{t=0}^{T-1} \alpha_t$. *Then the projected–SGD routine in Eq. Equation (17) is* $\beta$-*uniformly stable with:*

$$\beta \; \leq \; \frac{2\,B\,G_{\max}\,S_\alpha}{K+1}$$

*where $K$ is the number of perturbation graphs in the mixed sample.*

*Proof.* Let $S$ and $S'$ differ at one index. Denote $\delta_t = \|\theta_t - \theta_t'\|_2$ as in Lemma 6. At iteration $t$ the sampler picks a graph uniformly from $\{0, \ldots, K\}$.

*(i) Same index sampled.* With probability $1 - \frac{1}{K+1}$ we have $U_t = U_t'$, which is 1-expansive; hence $\delta_{t+1} \leq \delta_t$ (first branch of Lemma 6).

*(ii) Differing index sampled.* With probability $\frac{1}{K+1}$ the two updates differ but are both 1-expansive and $\sigma_t$-bounded, so $\delta_{t+1} \leq \delta_t + 2\sigma_t$ (second branch of Lemma 6).

Taking expectation over the sampling randomness and unrolling the recursion yields:

$$\mathbb{E}[\delta_T] \; \leq \; \sum_{t=0}^{T-1} \frac{1}{K+1}\, 2\sigma_t \; = \; \frac{2\,G_{\max}\,S_\alpha}{K+1}.$$

Finally, bounded loss width $B$ converts parameter drift into loss difference, giving the stated $\beta$. $\quad\square$

### E.2 PROOF OF THEOREM 3

**Theorem 3** (Robust Generalisation for DICT) With probability at least $1 - \delta$ over the mixed sample $S$,

$$R(\theta_S) \; \leq \; \widehat{R}_S(\theta_S) + \frac{4\,B\,G_{\max}\,S_\alpha}{K+1} + B\sqrt{\frac{\log(1/\delta)}{2(K+1)}}.$$

*Proof.* Denote by $\theta_S = \mathcal{A}(S)$ the parameter obtained after $T$ clipped, projected SGD steps with learning-rate schedule $\{\alpha_t\}_{t=0}^{T-1}$ and let $S_\alpha := \sum_{t=0}^{T-1} \alpha_t$. Lemma 7 already tells us that this optimiser is $\beta$-uniformly stable with

$$\beta = \frac{2\,B\,G_{\max}\,S_\alpha}{K+1}. \tag{19}$$

Due to Bousquet–Elisseeff's stability theory (Lemma 5), the difference between the empirical risk $\widehat{R}_S$ and the population risk $R$ is bounded in expectation by the same $\beta$. Because replacing one graph–label pair in the mixed sample affects $\widehat{R}_S(\theta)$ by at most $B/(K+1)$, McDiarmid's inequality implies that, with probability at least $1 - \delta$,

$$R(\theta_S) \leq \widehat{R}_S(\theta_S) + \beta + B\sqrt{\frac{\log(1/\delta)}{2(K+1)}}. \tag{20}$$

Finally, DICT's Wasserstein duality ensures $R_{\mathrm{rob}}(\theta) = R(\theta) + L(\theta)\sqrt{r} \leq \widehat{R}_S(\theta) + L(\theta)\sqrt{r} = \widehat{\mathcal{R}}_{\mathrm{rob}}(\theta)$, so subtracting $\widehat{R}_{\mathrm{rob}}(\theta_S)$ from both sides of Equation (20) and substituting $\beta$ from Equation (19) yields:

$$R_{\mathrm{rob}}(\theta_S) \; \leq \; \widehat{R}_{\mathrm{rob}}(\theta_S) \; + \; \frac{4\,B\,G_{\max}\,S_\alpha}{K+1} \; + \; B\sqrt{\frac{\log(1/\delta)}{2(K+1)}},$$

which is exactly the claimed bound. $\quad\square$

The stability gap scales as $\mathcal{O}\big(G_{\max}S_\alpha/(K+1)\big)$: adding more perturbations ($K\uparrow$) *simultaneously* tightens the statistical guarantee, while overly large learning-rate mass $S_\alpha$ or clipping threshold $G_{\max}$ widens the bound. Crucially, no Lipschitz continuity in the *parameters* is assumed—the analysis relies only on gradient clipping and the bounded loss range already enforced in DICT's training loop.

## F    FAIRNESS GUARANTEE OF DICT

**Demographic Parity and Equal Opportunity as Special Cases.**    Our formulation can recover standard group fairness notions by appropriate definitions of $\mathcal{L}_{\text{adv-fair}}$. Setting the divergence $d(\cdot, \cdot)$ to measure differences in marginal predictions across groups corresponds to enforcing Demographic Parity (DP):

$$\mathcal{L}_{\text{DP}} = \frac{1}{N} \sum_{i=1}^{N} |f(A, \mathbf{x}_i, s_i{=}1; \theta) - f(A, \mathbf{x}_i, s_i{=}0; \theta)| .$$

Likewise, restricting the comparison to positively labeled nodes recovers Equal Opportunity (EO):

$$\mathcal{L}_{\text{EO}} = \frac{\sum_{i:y_i=1} |f(A, \mathbf{x}_i, s_i{=}1; \theta) - f(A, \mathbf{x}_i, s_i{=}0; \theta)|}{|\{i : y_i = 1\}|} .$$

Both cases fit naturally into our DRO formulation by tuning the support of $\mathbb{P}_{|\mathbf{s}}$ and the definition of divergence $d$, allowing principled fairness constraints to be enforced within the same robust optimization framework.

### F.1    PROOF FOR PROPOSITION 2

In this section, we prove the result in Proposition 2.

**Proposition 2 (Fairness Guarantee of DICT).**    Let $\mathbb{P}_{\text{tr}} = \delta_{(A,X,y,s)}$ and define the uncertainty set with fairness-aware metric:

$$\mathcal{P}(\mathbb{P}_{\text{tr}}, r) = \left\{ \delta_{(A,X,y,\mathbf{s}+\Delta_s)} \,\middle|\, \kappa_s \|\Delta\mathbf{s}\|^2 \le r \right\} .$$

Then the DICT objective reduces to:

$$\min_{\theta \in \Theta} \max_{\Delta_s \in \mathcal{B}(r)} \mathcal{L}(f(A, X, \mathbf{s} + \Delta_s; \theta), \mathbf{y}),$$

which recovers fairness-constrained ERM as a special case when $r \ge \beta^2$.

*Proof.*  By the assumption that $f$ is $L(\theta)$-Lipschitz with respect to the sensitive attribute $\mathbf{a}$ and using the definition of statistical parity loss, we have:

$$|f(\cdot, s = 1; \theta) - f(\cdot, s = 0; \theta)| \le L(\theta)\|1 - 0\| = L(\theta),$$

for each node. Therefore, for any sample $(A, X, a)$,

$$\mathcal{L}_{\text{SP}}(f(A, X; \theta), \mathbf{s}) \le L(\theta),$$

and averaging over the empirical dataset, we obtain:

$$\beta \mathbb{E}_{\mathbb{P}_{\text{tr}}}[\mathcal{L}_{\text{SP}}(f(A, X; \theta), \mathbf{s})] \le \beta L(\theta),$$

where $\beta$ reflects how often such sensitive pairwise comparisons appear in the empirical dataset (e.g., proportional to group imbalance and total valid comparisons).

Meanwhile, from the DRO derivation, we know that solving Equation (6) leads to a regularization term $L(\theta)\sqrt{r}$. Thus, if $r \ge \beta^2$, we conclude:

$$\mathcal{L}_{\text{SP}}(f(A, X; \theta), \mathbf{s}) \le \beta L(\theta) \le L(\theta)\sqrt{r}.$$

Hence, solving DRO under fairness-aware Wasserstein distance implicitly upper bounds the statistical parity loss by a controllable regularization term, guaranteeing a degree of group fairness.    □

## G    FAIRNESS EVALUATION METRICS

This section addresses model biases arising from GNNs message-passing mechanisms, where neighborhood aggregation propagates structural correlations to systematically cluster node representations by sensitive attributes, leading to predictions that become statistically dependent on sensitive attributes. We first present the definitions of fairness, which is to ensure groups of people with different protected sensitive attributes receive comparable treatments statistically.

**Definition 7** (Statistical Parity Dwork et al. (2012)). *Statistical parity requires the predictions to be independent of the sensitive attribute s, i.e., $\hat{y} \perp s$. It could be formally written as:*

$$P(\hat{y}|s = 0) = P(\hat{y}|s = 1). \tag{21}$$

Statistical parity is the first fairness definition and has been widely adopted. However, it often cripples the utility of the model. Hence, we introduce Equalized Opportunity to alleviate the issue:

**Definition 8** (Equal Opportunity Hardt et al. (2016a)). *Equal opportunity requires that the probability of an instance in a positive class being assigned to a positive outcome should be equal for both subgroup members. The property of equal opportunity is defined as:*

$$P(\hat{y} = 1|y = 1, s = 0) = P(\hat{y} = 1|y = 1, s = 1). \tag{22}$$

The equal opportunity expects the classifier to give equal true positive rates across the subgroups. According to Beutel et al. (2017); Louizos et al. (2016), we apply the following metrics to quantitatively evaluate statistical parity and equal opportunity:

$$\Delta_{\text{SP}} = |P(\hat{y} = 1|s = 0) - P(\hat{y} = 1|s = 1)|, \tag{23}$$
$$\Delta_{\text{EO}} = |P(\hat{y} = 1|y = 1, s = 0) - P(\hat{y} = 1|y = 1, s = 1)|, \tag{24}$$

where the probabilities are evaluated on the test set.

## H EXPERIMENTAL DETAILS

In this section, we provide comprehensive details regarding our experimental setup. We begin by describing the hardware and software environments in which all experiments were conducted. We then introduce the datasets used in our study, including their construction, features, and sensitive attributes. Finally, we outline implementation details to ensure full reproducibility and facilitate future comparisons.

### H.1 EXPERIMENTAL ENVIRONMENTS

All experiments are conducted on a server equipped with 8 NVIDIA RTX A6000 GPUs and an AMD EPYC 7763 64-Core Processor. The system runs Ubuntu 22.04.3 LTS with Linux kernel version 6.5.0-18-generic. We employ PyTorch 2.7.1+cu126 with CUDA 12.6 support, along with PyTorch Geometric 2.6.1 for graph processing. Key dependencies include NumPy 1.26.4, SciPy 1.16.0, and Pandas 1.5.3. All required Python packages are specified in the accompanying `requirements.txt` file to ensure reproducibility.

### H.2 DATASET DETAILS

**NBA.** Derived from a Kaggle dataset, the NBA dataset includes approximately 400 professional basketball players, with features such as individual performance statistics from the 2016–2017 season, nationality, age, and salary. To construct the graph, we retrieve social relationships among players based on their Twitter interactions using the official Twitter API. Nationality is binarized into U.S. versus non-U.S. players and used as the sensitive attribute. The classification task is to predict whether a player's salary exceeds the median salary threshold.

**German.** The German Credit dataset is widely used for assessing the probability of loan default. In this graph formulation, each client is modeled as a node, and edges are established based on similarities in credit-related attributes. The dataset encompasses a range of financial and demographic features, including credit history, loan duration, credit amount, employment status, personal status, and age. The classification task is to determine whether a client poses a high or low credit risk, with gender considered as the sensitive attribute.

**Bail.** The Recidivism dataset consists of data on individuals granted bail from 1990 to 2009, capturing both demographic information and prior criminal history. Nodes correspond to defendants released on bail, while edges are constructed by linking individuals who share common attributes such as criminal background or demographic traits. The prediction task aims to classify whether a released defendant is likely to reoffend with a violent or non-violent crime. Race is treated as the sensitive attribute in this setting.

**Cora, CiteSeer and Pubmed.** They are three widely used citation networks commonly used in research related to machine learning, particularly in the field of semi-supervised and supervised node classification in citation networks. It consists of academic papers, their authors, and citation relationships between papers. In these academic citation datasets, nodes usually represent academic papers or documents. Each node corresponds to a single paper. Edges in these datasets typically represent citation relationships between papers. If paper A cites paper B, there will be a directed edge from A to B. These edges represent the flow of academic influence and are often used for tasks like document classification or recommendation. In each of the datasets, articles are categorized into 7, 6, and 3 classes, respectively, depending on their article types. The node features are represented as bag-of-words representations.

### H.3 Implementation Details

We conduct hyperparameter selection via search over a predefined range of values. Specifically, the Wasserstein radius $r$ is chosen from $\{0.5, 1.0, 1.5, 2\}$. The transportation cost parameters are set as $\kappa_A \in \{0.3, 0.5, 0.7, 0.9\}$, $\kappa_X \in \{0.3, 0.5, 0.7, 0.9\}$, $\kappa_s \in \{0.4, 0.6, 0.8\}$, and $\kappa_y \in \{0.1, 0.2, 0.3, 0.4\}$. We adopt the Adam optimizer with learning rate $l \in \{0.001, 0.003, 0.01\}$, hidden dimensions $\in \{32, 64, 128, 256\}$, dropout $\in \{0.3, 0.5, 0.7\}$, weight decay $\in \{0, 10^{-5}\}$, and number of layers $\in \{2, 3, 4\}$. The best configuration is selected based on validation performance. To ensure statistical robustness, we repeat each experiment 5 times with different random seeds and train all models for 300 epochs per run.

## I Additional Results

### I.1 Complexity Analysis

We compare the computational complexity of our proposed DICT framework with representative adversarial-based methods, FairVGNN Wang et al. (2022) and SFG Chen et al. (2025), in terms of forward and backward propagation costs, memory requirements, and parameter counts. Let $m$ denote the number of GNN layers, $K$ the number of nodes, $E$ the number of edges, and $F$ the hidden dimension. In addition, $\rho$ is a small constant (typically $\rho = 1$) representing the number of additional forward/backward evaluations required by DICT to estimate the Lipschitz regularizer. Lower-order terms such as $O(E + F^2)$ introduced by the perturbation generator are omitted from the table because they are dominated by the message-passing complexity.

Table 3: Time and space complexity comparison among FairVGNN, SFG, and DICT.

| Model | Forward Time | Forward Space | Backward Time | Backward Space | Param. Count |
|---|---|---|---|---|---|
| FairVGNN | $mEF + mKF^2$ | $E + mF^2 + mKF$ | $mEF + mKF^2$ | $E + mF^2 + mKF$ | $mF^2$ |
| SFG | $mEF + mKF^2 + mF^3$ | $E + mF^2 + mKF + mF^2$ | $mEF + mKF^2$ | $E + mF^2 + mKF$ | $mF^2$ |
| DICT (ours) | $(1+\rho)(mEF + mKF^2)$ | $E + mF^2 + mKF$ | $(1+\rho)(mEF + mKF^2)$ | $E + mF^2 + mKF$ | $mF^2$ |

**Discussion.** FairVGNN incurs the standard GNN complexity of $O(mEF + mKF^2)$ in both forward and backward passes, with memory cost $O(E + mF^2 + mKF)$. SFG introduces a singular value decomposition (SVD) on each weight matrix to enforce a tight Lipschitz bound, resulting in an additional $O(mF^3)$ cost in the forward pass and an extra $mF^2$ space overhead, which quickly dominates when $F$ grows. In contrast, DICT only requires a constant-factor overhead due to the Lipschitz regularizer, yielding a total complexity of $O\big((1+\rho)(mEF + mKF^2)\big)$ while maintaining the same memory requirement as FairVGNN. DICT achieves a favorable trade-off by preserving the lightweight complexity of FairVGNN up to a constant factor, while avoiding the cubic cost of SFG, thereby offering better scalability to large-scale graphs.

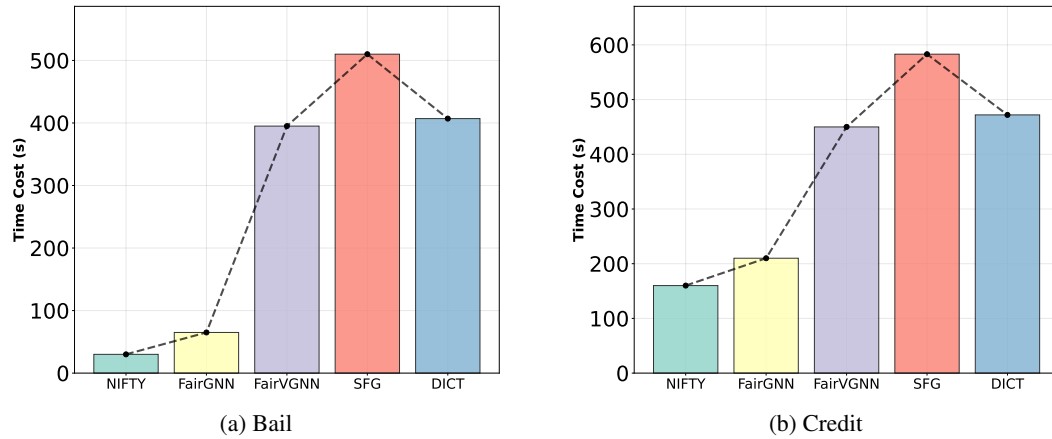

Figure 4: Training time cost on Bail and Credit with GCN backbone (in seconds)

**Training time Evaluation.** As shown in Figure 4, we further evaluate the training time cost of different methods on the Bail and Credit datasets using the GCN backbone. We compare representative fair graph learning approaches: NIFTY Agarwal et al. (2021), FairGNN Dai & Wang (2021a), FairVGNN Wang et al. (2022), the recent Lipschitz-constrained method SFG Chen et al. (2025), and our DICT. SFG constitutes the baseline for assessing the efficiency of Lipschitz-based fair GNNs. For a fair comparison, we standardize key hyperparameters across all methods by setting the hidden dimension to 256 and the number of GNN layers to 2. All models are trained under identical hardware and optimization conditions. The results show that DICT achieves nearly the same time cost as FairVGNN. Although introducing the Lipschitz regularizer inevitably adds a small overhead, DICT simultaneously achieves robustness and fairness within a unified framework—something FairVGNN cannot offer. Moreover, DICT is significantly faster than SFG, whose per-layer SVD-based Lipschitz projection leads to a substantial computational cost. These results demonstrate that DICT offers a superior efficiency–robustness–fairness trade-off compared with existing fair graph learning approaches.

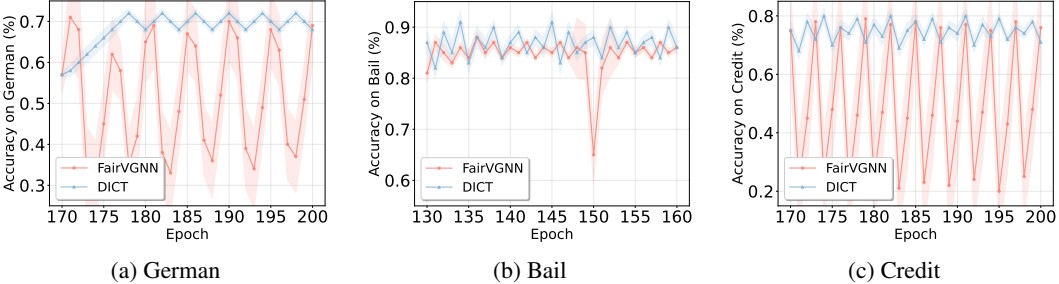

Figure 5: Training accuracy (%) curves on the German, Bail, and Credit datasets. Shaded regions indicate the rolling standard deviation (window size = 5), reflecting the model's stability across training epochs.

### I.2 STABILITY ACROSS EPOCHS

Figure 5 compares the training accuracy curves of DICT and FairVGNN Wang et al. (2022) on the German, Bail, and Credit datasets. DICT exhibits more stable convergence, as indicated by the smaller variance across epochs. The shaded regions, which represent rolling standard deviation with a window size of 5, show that FairVGNN suffers from frequent performance drops and high oscillation, particularly on the Credit dataset. This behavior suggests that FairVGNN is more sensitive to perturbations or optimization noise. In contrast, DICT maintains consistently smoother and higher training accuracy, reflecting its robustness and stable learning dynamics even under non-

convex training regimes. These results further validate the benefits of our Lipschitz-regularized DRO objective in enhancing training stability.

## I.3 Effect of Wasserstein Radius on Training Cost

Figure 6 illustrates how the training time of DICT scales with different Wasserstein radii $r$ across the German, Bail, and Credit datasets. As $r$ increases, the training cost rises gradually, reflecting the additional computational burden introduced by a larger uncertainty set. This is expected since larger $r$ values correspond to wider perturbation regions and thus more challenging optimization landscapes. Notably, the growth is smooth and does not exhibit exponential spikes, suggesting that DICT remains computationally tractable even when increasing the robustness budget. This scalability confirms the practicality of our formulation under varying robustness-strength settings.

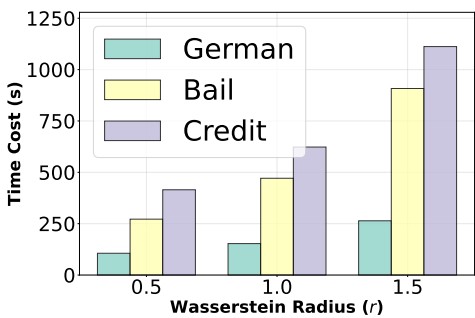

Figure 6: Training time (s) under varying Wasserstein radius $r$ on German, Bail, and Credit datasets.

## I.4 Results of Robustness Evaluation

In this section, we provide the clean accuracy, attack accuracy, and $\Delta$ACC results for all three citation networks (Cora, Citeseer, and PubMed). These results correspond to the main experiments reported in Section 5.2, and are presented here for completeness and transparency.

Table 4 reports the detailed performance of GCN-Jaccard Wu et al. (2019), GCN-SVDEntezari et al. (2020), Pro-GNN Jin et al. (2020), and our proposed DICT framework across the three datasets. As shown, DICT consistently achieves the highest clean accuracy, the best robustness under attacks.

Table 4: Comparison of robustness across three citation networks.

| Datasets | Metrics | GCN-Jaccard | GCN-SVD | Pro-GNN | DICT |
|---|---|---|---|---|---|
| Cora | Clean Acc(%) ↑ | 82.15 | 80.67 | 82.78 | **83.5** |
| | Attack Acc(%) ↑ | 74.84 | 69.54 | 78.35 | **80.78** |
| | $\Delta$Acc(%) ↓ | 7.31 | 11.13 | 4.52 | **2.72** |
| Citeseer | Clean Acc(%) ↑ | 72.10 | 70.65 | 73.28 | **74.3** |
| | Attack Acc(%) ↑ | 64.18 | 57.42 | 66.97 | **70.89** |
| | $\Delta$Acc(%) ↓ | 7.92 | 13.22 | 6.31 | **3.41** |
| Pubmed | Clean Acc(%) ↑ | 87.06 | 83.44 | **87.26** | 86.93 |
| | Attack Acc(%) ↑ | 83.95 | 80.72 | 85.13 | **85.61** |
| | $\Delta$Acc(%) ↓ | 3.11 | 2.72 | 2.13 | **1.32** |

## I.5 Achieving Privacy with DICT

As an extension of DICT, differential privacy (DP) Dwork et al. (2006); Kasiviswanathan et al. (2011) prevents leakage of sensitive information by injecting calibrated noise into the data or optimization process. In our setting, we focus on *local* differential privacy (LDP), where each user's sensitive attribute is privatized before being sent to the learner. Following Dai & Wang (2022), we

adopt the randomized response mechanism for binary sensitive attributes $s_i \in \{0, 1\}$: each $s_i$ is flipped independently with probability $\rho = \frac{1}{\exp(\epsilon)+1}$, yielding a privatized attribute $\tilde{s}_i$ that satisfies $\epsilon$-LDP (See Section 2 (Remark)). Concretely, we consider privacy budgets $\epsilon \in \{0.5, 1.0, 1.5, 2.0\}$, corresponding to flip probabilities $\rho \approx \{0.38, 0.27, 0.18, 0.12\}$. Importantly, the learner (DICT or any baseline) only observes $\tilde{s}$ and never accesses the raw sensitive attributes $s$, so the training procedure enjoys formal local privacy guarantees by design. For each $\epsilon$, we generate the sensitive attributes via randomized response and train DICT using exactly the same fairness-DRO objective as in the non-private case, simply replacing $s$ by $\tilde{s}$ in the fairness-related terms (e.g., group-wise constraints and sensitivity-based costs). For comparison, we also train NT-FairGNN Dai & Wang (2022), FairGCN Dai & Wang (2021a), and NTFC Lamy et al. (2019) under the same LDP protocol, following the implementation details in NT-FairGNN Dai & Wang (2022). We evaluate all methods in terms of group fairness ($\Delta_{\mathrm{SP}}$ and $\Delta_{\mathrm{EO}}$) and classification accuracy on the test set.

As shown in Figure 7, DICT consistently achieves the best trade-off across all privacy budgets. Under strong privacy (small $\epsilon$), DICT attains the lowest $\Delta_{\mathrm{SP}}$ and $\Delta_{\mathrm{EO}}$ among all methods, while also maintaining the highest accuracy. As $\epsilon$ increases (weaker privacy, smaller flip probability), all methods improve slightly in both fairness and accuracy, but DICT remains strictly better than NT-FairGNN, FairGCN, and NTFC on all three metrics. These results empirically confirm that the proposed DICT framework can be concretely instantiated as a privacy-preserving GNN, by plugging an LDP mechanism into the sensitive-attribute perturbation space, using the uncertainty-constrained DICT framework.

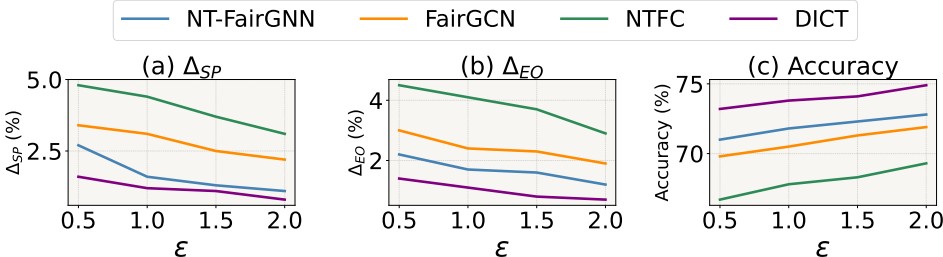

Figure 7: Performance (%) of DICT on NBA with private sensitive attributes under different privacy budgets.

## I.6 EXTENSION TO TRANSFORMER-BASED GNNS

To further evaluate whether DICT remains effective when applied to architectures beyond classical message-passing GNNs, we apply a Transformer-based graph encoder. Our DICT framework is inherently model-agnostic: it only requires the encoder $f_\theta$ to be a differentiable graph function, while robustness and fairness are enforced through a Wasserstein uncertainty set defined over graph structure, node features, and sensitive attributes. Transformer-based GNNs naturally satisfy this requirement—their attention layers and feed-forward blocks are fully differentiable, enabling gradient-based perturbation and Lipschitz estimation. We compare this Transformer-based DICT with two Transformer-based GNNs, NAGphormer Chen et al. (2023) and FairGT Luo et al. (2024), both of which employ global attention mechanisms to capture long-range structural dependencies.

We conduct experiments on three widely used datasets: German, Bail, and NBA. To ensure a fair comparison, we perform structure perturbations with the rate of 30% using random edge insertion and deletion. Meanwhile, we keep all experimental settings identical to FairGT Luo et al. (2024). As shown in Table 5, DICT consistently improves both clean and adversarial accuracy across the German, Bail, and NBA datasets. In addition, DICT achieves significantly stronger robustness (lower $\Delta$Acc) and superior fairness (lower $\Delta_{\mathrm{SP}}$ and $\Delta_{\mathrm{EO}}$). These results validate that DICT provides an architecture-independent, plug-and-play trustworthy learning framework that generalizes effectively from message-passing GNNs to advanced Transformer-based graph models.

Table 5: Comparison between NAGphormer, FairGT, and DICT on robustness and fairness metrics. ↑ means higher is better; ↓ means lower is better. Best results are in bold.

| Datasets | Metrics | NAGphormer | FairGT | DICT |
|---|---|---|---|---|
| German | Clean Acc(%) ↑ | 75.43±0.42 | 76.07±0.19 | **76.91±0.27** |
|  | Attack Acc(%) ↑ | 71.32±0.51 | 68.22±0.25 | **75.83±0.21** |
|  | $\Delta$Acc(%) ↓ | 4.11 | 7.85 | **1.08** |
|  | $\Delta_{SP}$(%) ↓ | 8.24±0.05 | 0.25±0.29 | **0.19±0.22** |
|  | $\Delta_{EO}$(%) ↓ | 6.86±0.03 | 0.17±0.04 | **0.11±0.51** |
| Bail | Clean Acc(%) ↑ | 93.39±0.32 | **95.48±0.36** | 94.62±0.23 |
|  | Attack Acc(%) ↑ | 86.12±0.64 | 86.33±0.33 | **92.89±0.33** |
|  | $\Delta$Acc(%) ↓ | 7.27 | 9.15 | **1.73** |
|  | $\Delta_{SP}$(%) ↓ | 7.13±0.29 | 0.55±0.34 | **0.33±0.29** |
|  | $\Delta_{EO}$(%) ↓ | 5.79 | 0.43 | **0.27** |
| NBA | Clean Acc(%) ↑ | 72.17±0.34 | 74.78±0.31 | **75.48±0.11** |
|  | Attack Acc(%) ↑ | 67.21±0.72 | 66.37±0.23 | **73.32±0.64** |
|  | $\Delta$Acc(%) ↓ | 5.05 | 8.41 | **2.16** |
|  | $\Delta_{SP}$(%) ↓ | 16.24±0.67 | 0.35±0.23 | **0.21±0.46** |
|  | $\Delta_{EO}$(%) ↓ | 12.32±0.35 | 0.26±0.42 | **0.17±0.36** |

## J    ALGORITHM

In this section, we present the optimization procedures for training the DICT framework under different trustworthiness objectives. We first detail the robustness-oriented training loop, which estimates local Lipschitz sensitivity scores and minimizes the regularized worst-case risk (see Algorithm 1). We then extend the procedure to incorporate fairness constraints by including sensitivity to the sensitive attribute (see Algorithm 2). Both variants follow a unified min–max formulation and leverage gradient-based approximations for efficient optimization under distributional perturbations. By adjusting the perturbation space and the transportation cost, DICT flexibly accommodates different trustworthiness dimensions, while maintaining a consistent training pipeline. This modular design enables the integration of multiple trustworthiness goals without altering the overall optimization structure.

---

**Algorithm 1** DICT Optimization Loop (Robustness-Only)

---

**Require:** Graph $G = (A, X)$, partial labels $\hat{\mathbf{y}}$, learning rate $\eta$, clipping radius $G_{\max}$, trade-off $r$, regularizer weights $\kappa_A, \kappa_X, \kappa_y$, $p$-norm index $p$, training iterations $T$

1: Initialize model parameters $\theta_0$
2: **for** each training step $t = 1, 2, \ldots, T$ **do**
3:     Sample perturbed graphs $\{(A_k, X_k, \hat{y}_k)\}_{k=1}^K$
4:     Construct empirical distribution $\mathbb{P}_{\mathrm{tr}}$ from mixed samples
5:     Initialize global Lipschitz score $L(\theta_t) \leftarrow 0$
6:     **for** each node $n$, where $n \in \mathcal{V}_l$ (labeled) or $n \in \mathcal{V}_u$ (unlabeled) **do**
7:       **if** $n$ is labeled **then**
8:         Compute gradients and estimate local score:
$$\mathrm{L}_n^{\mathrm{lab}}(\theta_t) \approx \sqrt{\frac{\|\nabla_{A_n}\mathcal{L}\|_F^2}{\kappa_A} + \frac{\|\nabla_{x_n}\mathcal{L}\|_2^2}{\kappa_X} + \frac{\|\nabla_{y_n}\mathcal{L}\|_2^2}{\kappa_y}}$$
9:       **else**
10:         Sample Gaussian noise $\xi \sim \mathcal{N}(0, I)$
11:         Compute probe gradient:
$$\mathrm{L}_n^{\mathrm{unlab}}(\theta_t) \approx \|\nabla_\xi \mathcal{L}(f(A + \xi_A, x_n + \xi_{x_n}; \theta_t), y_n + \xi_{y_n})\|_{d^*}$$
12:       **end if**
13:       Accumulate: $L(\theta_t) \leftarrow L(\theta_t) + L_n(\theta_t)^p$
14:     **end for**
15:     Aggregate global score:

$$L(\theta) = \left( \frac{\sum_{n \in \mathcal{N}_o}(\mathrm{L}_n^{\mathrm{lab}}(\theta))^p + \sum_{n \notin \mathcal{N}_o}(\mathrm{L}_n^{\mathrm{unlabb}}(\theta))^p}{N} \right)^{\frac{1}{p}}$$

16:     Compute empirical loss:

$$\widehat{R}(\theta_t) \leftarrow \mathbb{E}_{(A,X,\hat{y}) \sim \mathbb{P}_{\mathrm{tr}}}[\mathcal{L}(f(A, X; \theta_t), \hat{y})]$$

17:     Total objective:
$$F(\theta_t) \leftarrow \widehat{R}(\theta_t) + L(\theta_t)\sqrt{r}$$

18:     Compute gradient $g_t \leftarrow \nabla_\theta F(\theta_t)$
19:     Clip gradient: $g_t \leftarrow \mathrm{clip}_{G_{\max}}(g_t)$
20:     Update: $\theta_{t+1} \leftarrow \Pi_\Theta(\theta_t - \eta \cdot g_t)$
21: **end for**

---

---

**Algorithm 2** DICT Optimization Loop (Fairness-Only)

---

**Require:** Graph $G = (A, X)$, partial labels $\hat{\mathbf{y}}$, partially observed sensitive attributes $\hat{\mathbf{s}}$, learning rate $\eta$, clipping radius $G_{\max}$, trade-off $r$, regularizer weights $\kappa_A, \kappa_X, \kappa_y, \kappa_s$, $p$-norm index $p$, training iterations $T$

1: Initialize model parameters $\theta_0$
2: **for** each training step $t = 1, 2, \ldots, T$ **do**
3:     Sample fairness-aware perturbed graphs $\{(A_k, X_k, \hat{y}_k, \hat{s}_k)\}_{k=1}^K$ with transportation cost on **s**

4:     Construct empirical distribution $\mathbb{P}_{\text{tr}}$ from mixed samples
5:     Initialize fairness-aware Lipschitz score $L(\theta_t) \leftarrow 0$
6:     **for** each node $n$, where $n \in \mathcal{V}_l$ (labeled) or $n \in \mathcal{V}_u$ (unlabeled) **do**
7:       **if** $n$ is labeled **then**
8:         Compute gradients and estimate local score:
9:           $\mathrm{L}_n^{\text{lab}}(\theta_t) \approx \sqrt{\frac{\|\nabla_{A_n}\mathcal{L}\|_F^2}{\kappa_A} + \frac{\|\nabla_{x_n}\mathcal{L}\|_2^2}{\kappa_X} + \frac{\|\nabla_{y_n}\mathcal{L}\|_2^2}{\kappa_y} + \frac{\|\nabla_{s_n}\mathcal{L}\|_2^2}{\kappa_s}}$
10:       **else**
11:         Sample Gaussian noise $\xi \sim \mathcal{N}(0, I)$
12:         Compute fairness-aware probe gradient:
13:           $\mathrm{L}_n^{\text{unlab}}(\theta_t) \approx \|\nabla_\xi \mathcal{L}(f(A + \xi_A, x_n + \xi_{x_n}; \theta_t), y_n + \xi_{y_n})\|_{d^*}$
14:       **end if**
15:       Accumulate: $L(\theta_t) \leftarrow L(\theta_t) + L_n(\theta_t)^p$
16:     **end for**
17:     Aggregate fairness-aware score:
18:
$$L(\theta) = \left( \frac{\sum_{n \in \mathcal{N}_o} (\mathrm{L}_n^{\text{lab}}(\theta))^p + \sum_{n \notin \mathcal{N}_o} (\mathrm{L}_n^{\text{unlabb}}(\theta))^p}{N} \right)^{\frac{1}{p}}$$

19:     Compute empirical loss:
20:
$$\widehat{R}(\theta_t) \leftarrow \mathbb{E}_{(A, X, \hat{y}) \sim \mathbb{P}_{\text{tr}}} \left[ \mathcal{L}(f(A, X; \theta_t), \hat{y}) \right]$$

21:     Total fairness-aware objective:
22:
$$F(\theta_t) \leftarrow \widehat{R}(\theta_t) + L(\theta_t)\sqrt{r}$$

23:     Compute gradient $g_t \leftarrow \nabla_\theta F(\theta_t)$
24:     Clip gradient: $g_t \leftarrow \text{clip}_{G_{\max}}(g_t)$
25:     Update: $\theta_{t+1} \leftarrow \Pi_\Theta(\theta_t - \eta \cdot g_t)$
26: **end for**

---

# K   Use of Large Language Models (LLMs)

We used LLMs, specifically GPT-4o, as general-purpose assistive tools during the preparation of this paper. Their role was limited to retrieving and organizing related work through guided querying.

LLMs were not used for ideation, theoretical development, algorithm design, or generation of experimental results. All scientific contributions, including problem formulation, methodology, implementation, and analysis, are fully authored and verified by the human authors.

