# OpenReview forum: "DICT: Uncertainty-Constrained Trustworthiness for Graph Learning"
_ICLR.cc/2026/Conference — ICLR 2026 Conference Desk Rejected Submission_

### Official Review · Reviewer_aGWN · 2025-10-28

**Soundness:** 3
**Presentation:** 3
**Contribution:** 3
**Rating:** 6
**Confidence:** 3

**Summary:**

This paper introduces DICT, a unified framework for trustworthy graph learning that models distributional uncertainty via Wasserstein distributionally robust optimization (DRO). The authors address the challenge of ensuring robustness and fairness in graph neural networks (GNNs) . DICT formulates a unified optimization objective that captures perturbation-induced distributional shifts in graph topology, node
features, and labels, and minimizes the worst-case risk over the uncertainty set. The framework is instantiated for both robustness and fairness, with theoretical guarantees and extensive experiments demonstrating consistent improvements across multiple benchmarks.

**Strengths:**

Strengths:

1.The proposed method and framework are highly novel and appealing.

2.The experimental results demonstrate that the proposed method achieves excellent performance.

3.The paper provides a very clear derivation of the experiments.

**Weaknesses:**

Suggestions:

1.The experimental results would be more convincing if the raw accuracy values for the Cora, Citeseer, and PubMed datasets were provided, rather than only the $ \bigtriangleup acc$

2. The impact of hyperparameters on the experimental results is significant, particularly for $K_s$.

3. The influence of this work could be substantially broadenedif the method could be extended to more recent GNN research, such as Transformer-based GNNs.

**Questions:**

Please refer to the weakness

---

> ### Author Response · Authors · 2025-11-19
> **Response Part 1**
>
> Dear Reviwer aGWN,
>
> We express our gratitude for your thorough review and insightful comments on our paper. We are glad that you find our work **highly novel, very clear derivation** and the **excellent empirical performance**. We address the concerns you raised as follows:
>
> ---
> > **W1:** The experimental results would be more convincing if the raw accuracy values for the Cora, Citeseer, and PubMed datasets were provided, rather than only the $\Delta\mathrm{Acc}$.
> >
>
> Thanks for your valuable suggestion, and we apologize for missing the details! In the revised version, we have added the full raw accuracy results for the Cora, Citeseer, and PubMed datasets in Appendix I.4 (Results of Robustness Evaluation), where the newly added content is highlighted in red.
>
> As shown in the table below, DICT achieves the highest clean accuracy and the strongest robustness (i.e., smaller $\Delta\mathrm{Acc}$) under perturbations across all three benchmark datasets, demonstrating consistent improvements over GCN-Jaccard[1], GCN-SVD[2], and Pro-GNN[3]. We believe these details further strengthen the empirical evidence and make the experimental results more convincing.
>
> | **Datasets** | **Metrics** | **GCN-Jaccard** | **GCN-SVD** | **Pro-GNN** | **DICT** |
> | --- | --- | --- | --- | --- | --- |
> | **Cora** | Clean Acc(%) ↑ | 82.15 | 80.67 | 82.78 | **83.5** |
> |  | Attack Acc(%) ↑ | 74.84 | 69.54 | 78.35 | **80.78** |
> |  | $\Delta\mathrm{Acc}$(%) ↓ | 7.31 | 11.13 | 4.52 | **2.72** |
> | **Citeseer** | Clean Acc(%) ↑ | 72.10 | 70.65 | 73.28 | **74.3** |
> |  | Attack Acc(%) ↑ | 64.18 | 57.42 | 66.97 | **70.89** |
> |  | $\Delta\mathrm{Acc}$(%) ↓ | 7.92 | 13.22 | 6.31 | **3.41** |
> | **Pubmed** | Clean Acc(%) ↑ | 87.06 | 83.44 | **87.26** | 86.93 |
> |  | Attack Acc(%) ↑ | 83.95 | 80.72 | 85.13 | **85.61** |
> |  | $\Delta\mathrm{Acc}$(%) ↓ | 3.11 | 2.72 | 2.13 | **1.32** |
>
> ---
> > **W2:** The impact of hyperparameters on the experimental results is significant, particularly for $\kappa_s$
> >
>
> We thank the reviewer for pointing this out! We agree that the hyperparameters—particularly the transport cost weights $\kappa_A, \kappa_X, \kappa_y, \kappa_s$ —have a notable impact on the results. This behavior is **expected and inherent** to distributionally robust optimization under Wasserstein uncertainty sets.
>
> In DICT, each hyperparameter directly controls the penalty for perturbations along one direction:
>
> - $\kappa_A$ controls the magnitude of topology perturbations.
> - $\kappa_X$ controls the magnitude of feature perturbations.
> - $\kappa_y$ controls the magnitude of of label perturbations.
> - $\kappa_s$ controls the perturbation strength assigned to the sensitive attribute, and thus plays a critical role in fairness, as it directly determines the model’s tolerance to sensitive-attribute variations.
>
> Consequently, adjusting $\kappa_s$ penalizes unfair group-dependent shifts, which naturally leads to significant variation in fairness metrics ( $\Delta_\mathrm{SP}$, $\Delta_\mathrm{EO}$). This reflects **a core design property rather than instability.** Although all four hyperparameters jointly influence the robustness–fairness trade-off, we observe **continuous and predictable trends (Figure 3)**, and the optimal values are selected through validation.
>
> ---
> TO CONTINUE

---

> ### Author Response · Authors · 2025-11-19
> **Response Part 2**
>
> > **W3:** The influence of this work could be substantially broadened if the method could be extended to more recent GNN research, such as Transformer-based GNNs.
> >
>
> Thank you for the insightful comment! DICT requires only a mild regularity assumption on the backbone encoder—specifically, that the graph encoder is **locally Lipschitz** so that (i) the Wasserstein-robust objective admits a tractable dual form and (ii) the Lipschitz-based perturbation sensitivity used in DICT can be reliably estimated (Sec. 3, line 253-278). This assumption is satisfied by a broad class of models (see our detailed discussion in the response to Reviewer 8w2e, W7), including Transformer-based GNNs. Their computation consists of linear projections, dot-product attention with softmax, residual connections, and smooth nonlinearities, all of which are **locally Lipschitz** almost everywhere. As a result, DICT can be directly applied to Transformer-based GNNs without modifying the framework.
>
> In the revised version, we have **added the experiment using a Transformer-based graph encoder**, and compared with two representative Transformer-based GNNs, NAGphormer[4] and FairGT[5]. The newly added content is highlighted in red in **Appendix I.6** (Extension to Transformer-based GNNs). DICT consistently improves both clean and adversarial accuracy across the German, Bail, and NBA datasets. In addition, DICT achieves **significantly stronger robustness** (lower $\Delta\mathrm{Acc}$) and **superior fairness** (lower $\Delta_\mathrm{SP}$ and $\Delta_\mathrm{EO}$). These results validate that DICT provides a plug-and-play trustworthy learning framework  that generalizes effectively from message-passing GNNs to advanced Transformer-based graph models.
>
> | Datasets | Metrics | NAGphormer | FairGT | DICT |
> | --- | --- | --- | --- | --- |
> | German | Clean Acc(%) ↑ | 75.43±0.42 | 76.07±0.19 | **76.91±0.27** |
> |  | Attack Acc(%) ↑ | 71.32±0.51 | 68.22±0.25 | **75.83±0.21** |
> |  | $\Delta\mathrm{Acc}$(%) ↓ | 4.11 | 7.85 | **1.08** |
> |  | $\Delta_\mathrm{SP}$(%) ↓    | 8.24±0.05 | 0.25±0.29 | **0.19±0.22** |
> |  | $\Delta_\mathrm{EO}$(%) ↓ | 6.86±0.03 | 0.17±0.04 | **0.11±0.51** |
> | Bail | Clean Acc(%) ↑ | 93.39±0.32 | **95.48±0.36** | 94.62±0.23 |
> |  | Attack Acc(%) ↑ | 86.12±0.64 | 86.33±0.33 | **92.89±0.33** |
> |  |  $\Delta\mathrm{Acc}$(%) ↓ | 7.27 | 9.15 | **1.73** |
> |  | $\Delta_\mathrm{SP}$(%) ↓    | 7.13±0.29 | 0.55±0.34 | **0.33±0.29** |
> |  | $\Delta_\mathrm{EO}$(%) ↓ | 5.79 | 0.43 | **0.27** |
> | NBA | Clean Acc(%) ↑ | 72.17±0.34 | 74.78±0.31 | **75.48±0.11** |
> |  | Attack Acc(%) ↑ | 67.21±0.72 | 66.37±0.23 | **73.32±0.64** |
> |  |  $\Delta\mathrm{Acc}$(%) ↓ | 5.05 | 8.41 | **2.16** |
> |  | $\Delta_\mathrm{SP}$(%) ↓    | 16.24±0.67 | 0.35±0.23 | **0.21±0.46** |
> |  | $\Delta_\mathrm{EO}$(%) ↓ | 12.32±0.35 | 0.26±0.42 | **0.17±0.36** |
>
> ---
> Thank you, Reviewer aGWN, for the thoughtful and constructive feedback you have provided. We sincerely hope we have addressed your concerns and raised your impression of our work. We are happy to clarify any other questions if required.
>
> ---
> ## References
>
> [1] Wu H, Wang C, Tyshetskiy Y, et al. Adversarial Examples for Graph Data: Deep Insights into Attack and Defense[J].
>
> [2] Entezari N, Al-Sayouri S A, Darvishzadeh A, et al. All you need is low (rank) defending against adversarial attacks on graphs[C]//Proceedings of the 13th international conference on web search and data mining. 2020: 169-177.
>
> [3] Jin W, Ma Y, Liu X, et al. Graph structure learning for robust graph neural networks[C]//Proceedings of the 26th ACM SIGKDD international conference on knowledge discovery & data mining. 2020: 66-74.
>
> [4] Chen J, Gao K, Li G, et al. NAGphormer: A Tokenized Graph Transformer for Node Classification in Large Graphs[C]//The Eleventh International Conference on Learning Representations.
>
> [5] Luo R, Huang H, Yu S, et al. FairGT: a fairness-aware graph transformer[C]//Proceedings of the Thirty-Third International Joint Conference on Artificial Intelligence. 2024: 449-457.

---

> ### Author Response · Authors · 2025-11-26
>
> Dear Reviewer aGWN,
>
> I hope you are doing well. As the discussion period is nearing its end, I wanted to ensure that we have addressed all your concerns satisfactorily. If there are any additional points or feedback you would like us to consider, please let us know. Your insights are invaluable to us, and we are eager to address any remaining issues to further improve our work.
>
> Thank you for your time and effort in reviewing our paper.

---

### Official Review · Reviewer_8w2e · 2025-10-31

**Soundness:** 2
**Presentation:** 1
**Contribution:** 2
**Rating:** 2
**Confidence:** 4

**Summary:**

This paper proposes DICT, a distributionally robust optimization (DRO) framework for trustworthy graph learning. By constructing a Wasserstein uncertainty set over graph structure, features, labels, and sensitive attributes, DICT aims to jointly improve robustness and fairness in graph neural networks (GNNs). The authors transform the infinite-dimensional primal optimization problem into a tractable min-max formulation by leveraging strong duality and local Lipschitz continuity. Empirical evaluations on several fairness and robustness benchmarks demonstrate DICT’s effectiveness across multiple datasets and GNN architectures.

**Strengths:**

The framework is built upon a well-established distributionally robust optimization principle, extended to handle graph-structured data with non-i.i.d. dependencies. By introducing a unified objective that jointly models robustness and fairness via Wasserstein uncertainty sets, DICT provides a principled foundation for trustworthy GNNs. The dual formulation using Lipschitz regularization offers a computationally feasible approximation of the intractable primal problem. The experimental design is thorough, showing consistent improvements across architectures (GCN, GIN, GraphSAGE) and datasets. The use of gradient-based and Hessian-based approximations for perturbation sensitivity is reasonable and practically effective.

**Weaknesses:**

The distinction between label space and node set is unclear.
On page 2, lines 53–55, the label space is defined as Y=[0,1]^N, which suggests it has the same dimension as the node set. This may lead to confusion about whether each node has a binary label or if labels are assigned to all nodes regardless of supervision. Clarification of this formulation is necessary to avoid misunderstanding.

The assumption of i.i.d. perturbed graphs is problematic.
In line 114, the paper suggests that the set of perturbed graphs can be treated as approximately i.i.d. However, since all perturbed graphs are generated from the same base graph, they retain strong structural dependencies. The justification for this approximation is insufficient and should be clarified in relation to the theoretical guarantees provided later.

The meaning of the Dirac measure formulation is not explained clearly.
In line 116, the paper introduces the Dirac measure δ(A,X,y ̂), but does not provide adequate explanation of its role or implications in modeling graph distributions. This hinders the reader’s understanding of the empirical distribution used in the uncertainty set.

The label definitions in lines 119–120 are vague.
The relation between the observed labels y ̂,y ̂_kand the true labels y,y_kis not clearly explained. It remains ambiguous how these perturbed labels are generated, and whether they are considered noisy estimates or adversarial modifications.

The research problem and design goals are not clearly presented.
The introduction lacks a clear articulation of the core research problem and motivation. While the need for trustworthy graph learning is introduced, the specific challenge addressed by DICT is not sufficiently distinguished from existing methods. A concise formulation of the problem and the intended contributions would improve clarity.

Fairness and robustness are not formally defined.
Although the paper discusses robustness and fairness throughout, formal definitions for these concepts are missing. For instance, fairness metrics such as statistical parity or equalized odds are used in experiments but not rigorously defined in the main text. This affects the reproducibility and interpretability of the results.

Claims and Evidence
The empirical claims are supported by quantitative evaluations across datasets and backbones, and the use of ∆Acc, ∆SP, and ∆EO is appropriate. However, theoretical claims such as the validity of duality and Lipschitz approximations depend on assumptions (e.g., approximate i.i.d., local smoothness) that are not always adequately discussed or validated.

Relation to Broader Scientific Literature
The paper is well-referenced but could better situate itself in relation to closely related work. For instance, the ROAD method (Grari et al., 2024), which also uses robust optimization for fairness in GNNs, is cited but not discussed in depth. A clearer comparison with other DRO-based GNN approaches would strengthen the literature positioning.

Essential References Not Discussed
Some recent work in robust and fair graph learning, such as those applying kernel-based DRO or information-theoretic fairness constraints, could be included to broaden the context. Moreover, extensions to privacy or interpretability via similar uncertainty modeling are briefly mentioned but not adequately linked to existing studies. Authors are suggested to discuss them in related work and evaluations.

**Questions:**

refer to the weakness part

---

> ### Author Response · Authors · 2025-11-19
> **Response Part 1**
>
> Dear Reviewer 8w2e,
>
> We sincerely appreciate your thoughtful comments. We have carefully considered each of your questions and provide detailed responses below.
>
> ---
> > **W1**: The distinction between label space and node set is unclear. On page 2, lines 53–55, the label space is defined as $\mathcal Y=[0,1]^N$, which suggests it has the same dimension as the node set. This may lead to confusion about whether each node has a binary label or if labels are assigned to all nodes regardless of supervision. Clarification of this formulation is necessary to avoid misunderstanding.
> >
> Thank you for pointing this out, and we apologize for the ambiguity caused by our shorthand notation. In our formulation, $\mathbf y\in\mathcal Y=[0,1]^N$ denotes the **latent ground-truth label vector** for all $N$ nodes, while the **observed labels** are obtained via a binary mask $\mathbf M\_{\mathbf y}\in\\{0,1\\}^N$ as $\widehat{\mathbf y}=\mathbf y\odot \mathbf M_{\mathbf y}$, where $\odot$ denotes element-wise masking. Thus $\mathbf y$ and $\widehat{\mathbf y}$ share the same label space $\mathcal{Y}$, but only the masked entries participate in supervision.
>
> We have clarified this explicitly in the revised version (lines 100–102).
>
> ---
> > **W2**: The assumption of i.i.d. perturbed graphs is problematic. In line 114, the paper suggests that the set of perturbed graphs can be treated as approximately i.i.d. However, since all perturbed graphs are generated from the same base graph, they retain strong structural dependencies. The justification for this approximation is insufficient and should be clarified in relation to the theoretical guarantees provided later.
> >
>
> Thanks for raising this important point, we confirm that the perturbed graphs are **i.i.d. by construction**. Across all perturbation types described in Appendix D.1, the randomness in each perturbed sample is introduced **solely from i.i.d. noise variables**, while the base graph $G=(A,X,\widehat{\mathbf y})$ is fixed and therefore does not affect independence or identical distribution.
>
> This is precisely the same situation as the classical fact that adding a constant drift term to i.i.d. noise preserves i.i.d.—e.g., variables of the form $Z_k=\varepsilon_k+c$ remain i.i.d. because the constant shift $c$ does not alter the independence or distribution of the underlying noise.
>
> In addition, our theoretical analysis does **not** require independence at the node or edge level; Theorem 3 relies only on independence **across graph-level perturbation draws**, which our perturbation scheme fully satisfies.
>
> ---
> > **W3**: The meaning of the Dirac measure formulation is not explained clearly. In line 116, the paper introduces the Dirac measure δ(A,X,y ̂), but does not provide adequate explanation of its role or implications in modeling graph distributions. This hinders the reader’s understanding of the empirical distribution used in the uncertainty set.
> >
> Thanks for pointing this out. In our notation, the Dirac measure $\delta_{(A,X,\widehat{\mathbf y})}$ is the probability measure **concentrated on the single graph instance $(A,X,\widehat{\mathbf y})$**, characterized by $\int\varphi(A',X',{\mathbf y}')\mathrm d \delta_{(A,X,\widehat{\mathbf y})}$ for any bounded measurable test function $\varphi$ (e.g. a loss function $\mathcal L(f(A,X;\theta),\widehat{\mathbf y})$ in Eq. 1).
>
> This is a standard construction in measure theory and machine learning for representing individual samples as probability measures. We adopt this classical device in the graph-learning setting to embed each graph instance—including its perturbed realizations obtained by independently injecting noise into the graph structure, node features, and labels—into a common probability space and to form the empirical distribution  $\mathbb{P}\_{\mathrm{tr}} : = \eta \delta_{(A,X,\widehat{\mathbf{y}})} + (1 - \eta)\cdot \tfrac{1}{K} \textstyle\sum_{k=1}^K \delta_{(A_k,X_k, \widehat{\mathbf{y}}_k)}$, which is a convex combination of Dirac measures and serves as the **center of our Wasserstein uncertainty set**.
>
> Using Dirac measures in this way **extends the support of the empirical distribution to all perturbed graph realizations**, enabling the uncertainty set to more faithfully capture variability in topology, features, and labels, and ensuring that the Wasserstein distance and its dual formulation remain well-defined on this graph space.
>
> ---
> TO CONTINUE

---

> ### Author Response · Authors · 2025-11-19
> **Response Part 2**
>
> > **W4**: The label definitions in lines 119-120 are vague. The relation between the observed labels $\widehat{\mathbf y},\widehat{\mathbf y}_k$ and the true labels $\mathbf y,\mathbf y_k$ is not clearly explained. It remains ambiguous how these perturbed labels are generated, and whether they are considered noisy estimates or adversarial modifications.
> >
>
> Thank you for the helpful comment. We would like to clarify that $\mathbf y$ and $\mathbf y_k$ denote the **latent ground-truth labels** of the original and perturbed graphs, while $\widehat{\mathbf y}$ and $\widehat{\mathbf y}\_k$ are the **observed labels** used for training. The observed labels are related by a binary mask $\mathbf M\_{\mathbf y}\in\\{0,1\\}^N$ as $\widehat{\mathbf y}=\mathbf y\odot \mathbf M_{\mathbf y}$, and similarly for $\widehat{\mathbf y}_k$.
>
> With these definitions, we confirm that the perturbed labels in DICT are neither noisy estimates nor sample-level adversarial modifications. Instead, they arise as distributionally adversarial deviations induced by the maximization step in Theorem 2, which identifies the worst-case distributional direction inside the Wasserstein ball. As noted in our manuscript (“DICT implicitly identifies the worst-case perturbation direction… including labels” in lines 239-240), this adversariality operates **at the distribution level**, not by injecting random noise or performing instance-wise adversarial attacks. The deviation is bounded and controlled by the label cost term in the Wasserstein metric, corresponding to the Lipschitz-based worst-case sensitivity direction modeled in our framework.
>
> ---
> > **W5**: The research problem and design goals are not clearly presented. The introduction lacks a clear articulation of the core research problem and motivation. While the need for trustworthy graph learning is introduced, the specific challenge addressed by DICT is not sufficiently distinguished from existing methods. A concise formulation of the problem and the intended contributions would improve clarity.
> >
>
> We thank the reviewer for the helpful comment and apologize for any confusion caused. We would like to clarify that **the introduction presents the research problem, motivation, and the design goals of DICT**, although some parts may not have been sufficiently emphasized due to space constraints. We summarize and further highlight the corresponding sections here for clarity.
>
> ### Research Problem (Lines 41–47) and Design goals (Line 53)
>
> Existing approaches generally lack collaborative modeling capabilities, often resulting in conflicts when multiple trustworthiness objectives coexist in complex environments, and struggle to achieve holistic optimization.
>
> **DICT addresses the problem of jointly modelling these objectives through a unified distributional-uncertainty formulation.**
>
> TO CONTINUE

---

> ### Author Response · Authors · 2025-11-19
> **Response Part 3**
>
> ### Differences from existing methods and the challenges solved by DICT
>
> To clarify the **formulation**, we outline how each core equation in DICT directly addresses the challenges identified in trustworthy graph learning.
>
> 1. $\min_{\theta \in \Theta} \mathbb{E}\_{(A,X, {\mathbf{y}}) \sim \mathbb{P}\_{\mathrm{tr}}} \left[ \mathcal{L}\left(f\left(A,X; \theta\right), {\mathbf{y}}\right) \right]\quad (\text{Eq.1, Line 122} )\Rightarrow\min_{\theta\in\Theta}\sup_{\mathbb P\in\mathcal P}\mathbb E_{(A,X,\mathbf{y})\sim\mathbb P}\left[\mathcal L\left(f(A,X;\theta),\mathbf{y}\right)\right]\quad(\text{Eq.2, Line 126})$
>
> We introduce an uncertainty set that captures distributional shifts arising from discrepancies between the observed training graph and potential testing graphs, thereby providing the preliminary form of our optimization objective. By customizing the uncertainty set $\mathcal P$, the transportation cost $c$, and the task loss $\mathcal L$, DICT flexibly accommodates diverse trustworthiness dimensions within a unified optimization framework.
> For example, **robustness** is instantiated via
> $$\min_{\theta \in \Theta} \sup_{\mathbb{P} \in \mathcal{P}(\mathbb{P}\_{\mathrm{tr}}, r)}\mathbb{E}\_{(A, X, \mathbf{y}) \sim \mathbb{P}}\left[\mathcal{L}_{\mathrm{Rb}}\left(f(A, X; \theta), \mathbf{y} \right)\right]\quad(\text{Eq.7, Line 297})$$
> while **fairness** is achieved through
>
> $$\min_{\theta \in \Theta} \sup_{\mathbb{P} \in \mathcal{P}(\mathbb{P}\_{\mathrm{tr}}, r)}\mathbb{E}\_{\mathbf{s} \sim \mathbb{P}\_{\mathbf{s}}} \left[\mathbb{E}\_{(G, \mathbf{y}) \sim \mathbb{P}\_{|\mathbf{s}}}\left[\mathcal{L}_{\text{Fair}}(f(G; \theta), \mathbf{y})\right]\right] \quad(\text{Eq.10, Line 342})$$
> demonstrating how different trustworthiness objectives can be accommodated under the same DRO formulation and thereby addressing **Challenge 1**.
>
> **Challenge1: Can trustworthy objectives be modeled into a distributional uncertainty framework?**
>
> Existing trustworthiness objectives are often incompatible, and optimizing one can inadvertently worsen another due to uncontrolled distributional shifts. For example, structural perturbations not only degrade prediction accuracy but also amplify group-level bias [1]. These observations motivate our **core design goal**: to unify robustness, fairness, and other trustworthiness objectives within a single DRO framework, rather than addressing them through task-specific schemes as prior works do.
>
> 2. $\min_{\theta\in\Theta}\sup_{\mathbb P\in\mathcal P}\mathbb E_{(A,X,\mathbf{y})\sim\mathbb P}\left[\mathcal L\left(f(A,X;\theta),\mathbf{y}\right)\right]\quad(\text{Eq.2, Line 126})\Rightarrow\min_{\theta\in\Theta}\sup_{\mathbb P\in\mathcal P(\mathbb P_{\mathrm{tr}},r)}\mathbb E_{(A,X,\mathbf{y})\sim\mathbb P}\mathcal L\left(f(A,X;\theta),\mathbf{y}\right)\quad(\text{Eq.3, Line 150})$
>
> To address the fact that a single graph does not provide i.i.d. samples, DICT first **generates $K$ i.i.d. perturbed graphs $\\{(A_k,X_k,\widehat{\mathbf y}\_k)\\}_{k}$, by applying controlled perturbations to the original $(A,X,\widehat{\mathbf y})$. Based on these perturbed instances, DICT constructs the empirical perturbed distribution** $\mathbb{P}\_{\mathrm{tr}} := \eta\, \delta_{(A,X,\widehat{\mathbf{y}})} + (1 - \eta)\cdot \tfrac{1}{K} \textstyle\sum_{k=1}^K \delta_{(A_k,X_k, \widehat{\mathbf{y}}_k)}$, which then **serves as the center of the Wasserstein uncertainty set**. By generating perturbations of the entire graph and using them to form a principled empirical center, DICT resolves **Challenge 2**.
>
> **Challenge2: Can we define a uncertainty set for graph data given one non-i.i.d. graph?**
>
> Classical DRO relies on the i.i.d. assumption to construct empirical distributions. However, this assumption breaks down in graph-structured data, where nodes and edges are inherently dependent, **making it fundamentally unclear how to define the center of the uncertainty set when starting from a single graph.**
>
> TO CONTINUE

---

> ### Author Response · Authors · 2025-11-19
> **Response Part 4**
>
> 3. $\min_{\theta\in\Theta}\sup_{\mathbb P\in\mathcal P(\mathbb P_{\mathrm{tr}},r)}\mathbb E_{(A,X,\mathbf{y})\sim\mathbb P}\mathcal L\left(f(A,X;\theta),\mathbf{y}\right)\quad(\text{Eq.3, Line 150})\Rightarrow
> \min_{\theta \in \Theta} \inf_{\lambda \geq 0} \left\\{
> \lambda r -
> \mathbb{E}\_{(\widehat{A},\widehat{X},\widehat{\mathbf{y}}) \sim \mathbb{P}\_{\mathrm{tr}}}
> \left[
> \Phi_\lambda(\widehat{A},\widehat{X}, \widehat{\mathbf{y}};\theta)
> \right]
> \right\\}\quad(\text{Eq.4, Line 200})$$
> \Rightarrow\min_{\theta\in\Theta}
> \mathbb{E}\_{(A,X,\mathbf y)\sim\mathbb{P}_{\mathrm{tr}}} \big[
> \mathcal{L}(f(A,X; \theta), \mathbf{y}) + L(\theta) \sqrt{r}
> \big]\quad(\text{Eq.6, Line 220})$
>
> The min–max DICT objective is an infinite-dimensional optimization problem, as the inner supremum ranges over all probability distributions within the Wasserstein ball. DICT applies strong duality and local Lipschitz continuity to obtain a tractable dual surrogate that reduces the infinite-dimensional DRO formulation to a finite-dimensional expression amenable to gradient-based optimization, thereby resolving **Challenge 3**.
>
> **Challenge3: How to derive a computationally tractable formulation with theoretical guarantees?**
>
> Even after constructing a suitable uncertainty set, the resulting DRO objective remains an **infinite-dimensional optimization problem**, since the inner maximization is taken over all probability distributions within the Wasserstein ball. Such a formulation is intractable to optimize directly for graph neural networks.
>
> Please see **Appendix A** for a more detailed distinction between DICT and existing approaches.
>
> ---
> TO CONTINUE

---

> ### Author Response · Authors · 2025-11-19
> **Response Part 5**
>
> > **W6**: Fairness and robustness are not formally defined. Although the paper discusses robustness and fairness throughout, formal definitions for these concepts are missing. For instance, fairness metrics such as statistical parity or equalized odds are used in experiments but not rigorously defined in the main text. This affects the reproducibility and interpretability of the results.
> >
>
> We thank the reviewer for pointing out the need for clearer formal definitions. We would like to clarify that **both robustness and fairness are already defined in our submission**, though some definitions appear in the appendix due to page limits.
>
> **Definitions for robustness:**
>
> 1. In Section 4.1, lines 291–292, we define the **standard sample-wise adversarial robustness formulation**.
> $$\min_{\theta \in \Theta} \max_{\Delta_A \in \mathcal{P}_A, \Delta_X \in \mathcal{P}_X}
> \mathcal{L}\big(f(A + \Delta_A, X + \Delta_X; \theta), \mathbf{y}\big)$$
>
> 2. In Section 4.1, lines 296–299, we formally **define** robustness in DICT using Wasserstein distributionally robust optimization, which minimizes the worst-case expected loss over all distributions within a Wasserstein uncertainty set.
>  $$\min_{\theta \in \Theta} \sup_{\mathbb{P} \in \mathcal{P}(\mathbb{P}\_{\mathrm{tr}}, r)}\mathbb{E}\_{(A, X, \mathbf{y}) \sim \mathbb{P}}\left[\mathcal{L}_{\mathrm{Rb}}\left(f(A, X; \theta), \mathbf{y} \right)\right] \quad(\text{Eq.7})$$
>
> 3. In Section 4.1, Proposition 1 (lines 304–311) establishes that classical adversarial robustness is a special case of DICT.
> $$\min_{\theta \in \Theta}
> \max_{\Delta_A, \Delta_X \in \mathbb{L}_r}
> \mathcal{L}(f(A + \Delta_A, X + \Delta_X; \theta), y)$$
>
> 4. In Section 5.1 (Evaluation Metrics), lines 387–391, we describe how **robustness is empirically quantified**. We evaluate model behavior on both clean and perturbed graph instances. *Clean Acc* measures classification accuracy on the original graph, while *Attack Acc* is computed on synthetic perturbations. We measure **robustness** via  $\Delta\mathrm{Acc} =| \mathrm{Attack Acc} - \mathrm{Clean Acc}|$, where a smaller value indicates **better robustness** to distributional shift.
>
> **Definitions for fairness:**
>
> 1. In Section 4.2, lines 330–333, we define the standard **fairness-regularization formulation**, where fairness is enforced by adding a statistical-parity penalty $\mathcal{L}\_{SP}$  that quantifies prediction discrepancies across sensitive groups.
>  $$\min_{\theta \in \Theta}\mathcal{L}(f(A, X; \theta), \mathbf{y})+ \gamma \mathcal{L}_{\text{SP}}(f(A, X; \theta), \mathbf{s}) \quad(\text{Eq.8})$$
>
> 2. In Section 4.2, lines 341–345, we formally **define fairness** in DICT through the Wasserstein transportation cost, which augments the uncertainty set with a sensitive-attribute term   $\kappa_s\|s-\hat{s}\|$ .
>  $$\min_{\theta \in \Theta} \sup_{\mathbb{P} \in \mathcal{P}(\mathbb{P}\_{\mathrm{tr}}, r)}\mathbb{E}\_{\mathbf{s} \sim \mathbb{P}\_{\mathbf{s}}}\left[\mathbb{E}\_{(G, \mathbf{y}) \sim \mathbb{P}\_{| \mathbf{s}}}\left[\mathcal{L}_{\text{Fair}}(f(G; \theta),\mathbf{y})\right]\right]\quad(\text{Eq.10})$$
>
> 3. In Section 4.2, Proposition 2 (lines 347–353) establishes that classical fairness constraints are **special cases** of our Wasserstein-DRO **formulation**.
> $$\min_{\theta \in \Theta}
> \max_{\Delta_s \in \mathbb{O}_r}
> \mathcal{L}(f(A, X, \mathbf{s} + \Delta_s; \theta), \mathbf{y})\quad(\text{Eq.11})$$
>
> 4. In Section 5.1 (Evaluation Metrics), lines 391–393, we describe how **fairness is empirically quantified**. We report **Statistical Parity** (SP) and **Equal Opportunity** (EO)—two widely adopted group-fairness metrics—measuring prediction gaps across sensitive groups. Lower  $\Delta_\mathrm{SP}$/  $\Delta_\mathrm{EO}$values indicate **better fairness**, consistent with our theoretical formulation.
> 5. In Appendix G, lines 1350–1355, we formally define **Statistical Parity (SP)** .
> $$P(\hat{y}|s=0) = P(\hat{y}|s=1) \quad(\text{Eq.21})$$
>
> 6. In Appendix G, lines 1356–1360, we provide the formal definition of **Equal Opportunity (EO)** .
> $$P(\hat{y}=1|y=1, s=0) = P(\widehat{y}=1|y=1, s=1)\quad(\text{Eq.22})$$
>
> 7. In Appendix G, lines 1361–1366, we further specify the **quantitative evaluation metrics** used for fairness.
> $$\Delta_{\mathrm{SP}} = |P(\hat{y}=1|s=0) - P(\widehat{y}=1|s=1)|\quad(\text{Eq.23})$$
> $$\Delta_{\mathrm{EO}} = |P(\widehat{y}=1|y=1, s=0) - P(\widehat{y}=1|y=1, s=1)|\quad(\text{Eq.24})$$
>
> ---
> TO CONTINUE

---

> ### Author Response · Authors · 2025-11-19
> **Response Part 6**
>
> > **W7**: Claims and Evidence The empirical claims are supported by quantitative evaluations across datasets and backbones, and the use of** $\Delta{\mathrm{Acc}}$**,** $\Delta_{\mathrm{SP}}$**, and** $\Delta_{\mathrm{EO}}$ **is appropriate. However, theoretical claims such as the validity of duality and Lipschitz approximations depend on assumptions (e.g., approximate i.i.d., local smoothness) that are not always adequately discussed or validated.
> >
>
> Thanks for the insightful comment. We would like to emphasize that the theoretical development in DICT is **self-contained and complete**, and all required assumptions are **standard and easily satisfied** in modern graph learning.
>
> As detailed in our response to **W2**, the **i.i.d. assumption** holds because all perturbations are generated from **independently sampled noise variables**, while the base graph is fixed and therefore does not affect independence or identical distribution.
>
> The **local Lipschitz continuity assumption** is a mild and widely adopted condition. For example, Assumption 4.4 in [2] uses the same requirement. Moreover, modern GNNs such as GCN, GraphSAGE, and GAT, equipped with bounded activations and normalization layers, satisfy **local Lipschitz continuity almost everywhere** (see Theorem 3.1 in [3] and Theorems 4.2–4.4 in [4]), making this assumption natural in practice.
>
> In practice, the max-step in DICT provides an estimate of the local Lipschitz constant, and the overall computational cost of this procedure is reported in Appendix I.1(Complexity Analysis). These findings demonstrate that DICT can be applied in real-world deployment without substantial computational burden.
>
> Under these standard assumptions, the strong duality between the primal Wasserstein DRO objective and its Lagrangian form follows directly from classical convex–concave arguments (Theorem 4.2 in [5]). Therefore, no additional empirical validation is required.
>
> For clarity, we restate the supporting results as follows:
>
> > Lemma 1 (Theorem 3.1 in [3]) For an $m$-layer GNN with ReLU activations (whose Lipschitz constant is 1) and nonnegative weight matrices, the cumulative Lipschitz constant $\mathrm{Lip}(f)$ of the GNN encoder is bounded as
> $\mathrm{Lip}(f) \le
> \left\|
> (w_0^{(m)} + h_{\max}(\lambda) w_1^{(m)})
> \cdots
> (w_0^{(1)} + h_{\max}(\lambda) w_1^{(1)})
> \odot w_g
> \right\|$,
> where $\|\cdot\|$ is the spectral norm and $\odot$ denotes element-wise broadcasting.
> >
>
> This result establishes that standard GNN architectures with bounded activations are locally Lipschitz almost everywhere.
>
> > Lemma 2 (Theorem 4.2 in [4]) For a GCNConv layer defined as $\mathrm{GCNConv}(X) = \hat{A} X W$, the Lipschitz constant of the layer is bounded by
> $\mathrm{Lip}(\mathrm{GCNConv})
> \le
> \max_{i\in[N]}
> \left\|
> \left[
> |\hat{A}\_{ii}| \, \| W_{:,k} \|
> \right]\_{k=1}^{F'}
> \right\|,$
> *where $\hat{A}_{ii}$* is the $(i,i)$-entry of the normalized adjacency matrix and $W_{:,k}$ is the $k$-th column of the weight matrix.
> >
>
> This result confirms that GCN layers admit a finite Lipschitz bound and are therefore locally Lipschitz almost everywhere.
>
> > Lemma 3 (Theorem 4.3 in [4]) For a SAGEConv layer defined as
> $Z = X W_1 + V \sigma(X W_2 + 1 \otimes b^\top) W_3,$
> > its Lipschitz constant is bounded by
> > $\mathrm{Lip}(\mathrm{SAGEConv})
> > \le
> > \max_{i\in[N]}
> > \left\|
> > \left[
> > \| (W_1)\_{:,k} \| +
> > V_{ii}  \| (W_2 W_3)\_{:,k} \|
> > \right]\_{k=1}^{F'}
> > \right\|,$
> > where $V_{ii}$ is the $(i,i)$-entry of the mean aggregation matrix $V$, $(W_1)\_{:,k}$ is the $k$-th column of $W_1$, and $(W_2 W_3)\_{:,k}$ is the $k$-th column of $W_2 W_3$.
> >
>
> This result shows that SAGEConv layers also admit finite Lipschitz bounds and are locally Lipschitz almost everywhere.
>
> > Lemma 4 (Theorem 4.4 in [4])
> > Consider a single-head GATConv layer defined as
> > $Z = S X W,$
> > where $W \in \mathbb{R}^{F \times F'}$ is the trainable weight matrix and
> > $S_{ij} =
> > \frac{\exp(\sigma(a[W x_i \,\|\, W x_j]))}
> > {\sum_{k \in \mathcal{N}i} \exp(\sigma(a[W x_i \,\|\, W x_k]))}$
> > *are the attention coefficients.
> > Let $M_{\mathrm{Lip}}(\mathrm{GATConv})$* be the matrix with entries
> > $M_{\mathrm{Lip}}(\mathrm{GATConv})\_{ik}=(S_{ii} X_{i,:} W_{:,k}-S_{ii} \sum_{j \in \mathcal{N}\_i} S_{ij} X_{j,:} W_{:,k})\\|v\\|+S_{ii} \| W_{:,k} \|,$
> > where $X_{i,:}$ is the $i$-th row of $X$, $W_{:,k}$ is the $k$-th column of $W$,
> > and $\sigma$ is an activation function such as LeakyReLU.
> > Then for any $\epsilon > 0$, there exists a punctured neighborhood $\mathcal{U}\_0(X)$ such that
> > $\mathrm{Lip}\_{X,\mathcal{U}\_0(X)}(\mathrm{GATConv})\le
> > \| M_{\mathrm{Lip}}(\mathrm{GATConv}) \|_{\infty,2} + \epsilon.$
> >
>
> This establishes that GATConv layers admit finite local Lipschitz bounds almost everywhere.
>
> ---
> TO CONTINUE

---

> ### Author Response · Authors · 2025-11-19
> **Response Part 7**
>
> > **W8:** Relation to Broader Scientific Literature The paper is well-referenced but could better situate itself in relation to closely related work. For instance, the ROAD method (Grari et al., 2024), which also uses robust optimization for fairness in GNNs, is cited but not discussed in depth. A clearer comparison with other DRO-based GNN approaches would strengthen the literature positioning.
> >
>
> We thank the reviewer for the helpful suggestion. We agree that a clearer positioning of DICT relative to closely related DRO-based approaches can strengthen the presentation of the paper. We provide the requested clarification below and have added a dedicated paragraph in the revised version. (**Lines 743–755 in Appendix A.3**)
>
> 1. On the ROAD method
>
> We thank the reviewer for highlighting the ROAD [6], which indeed adopts distributionally robust optimization for fairness. However, ROAD is **not graph-based**—it is designed for tabular classification and does not involve graph structures, message passing. Its uncertainty modeling is therefore fundamentally different from the graph-specific distributional shifts considered in DICT. We have clarified this distinction in the revised text. (**Lines 743–745**)
>
> 2.  Comparison with existing DRO-based GNN approaches.
>
> Regarding the reviewer’s comment that *“a clearer comparison with other DRO-based GNN approaches would strengthen the literature positioning,”* we note that our original submission **already discussed DRO-based GNN approaches** in **Appendix A.3 (Lines 745–750)**. Specifically, our **Related Work** summarizes prior studies that incorporate distributionally robust optimization into graph learning, including **“Distributionally Robust Semi-Supervised Learning over Graphs, 2021”[7]**, **DR-GNN[8]**, **DRGO[9]**, **group DRO[10]**,**DR-FLR[11]** , which apply DRO to improve robustness or fairness.
>
> To further strengthen the literature positioning, we have added a summary of the limitations of existing DRO-based graph learning methods in Appendix A.3 (Lines 751–755). Specifically, existing approaches do not consider combinations of **multiple perturbation types**. This prevents them from capturing the interactions among structural, feature, label, and sensitive-attribute perturbations. In addition, current formulations are typically tailored to a single trustworthiness objective (e.g., robustness or fairness) and **lack a unified mechanism for simultaneously addressing robustness, fairness, and privacy**. These limitations motivate our DICT framework, which integrates multiple trustworthiness goals within a unified Wasserstein-DRO formulation.
>
> ---
> >**W9**: Essential References Not Discussed Some recent work in robust and fair graph learning, such as those applying kernel-based DRO or information-theoretic fairness constraints, could be included to broaden the context. Moreover, extensions to privacy or interpretability via similar uncertainty modeling are briefly mentioned but not adequately linked to existing studies. Authors are suggested to discuss them in related work and evaluations.
> >
>
> Thank the reviewer for this helpful suggestion to broaden the literature context. We would like to clarify that Appendix A  provides a comprehensive discussion of the essential references relevant to this work (**as also noted by you in W8 that *“the paper is well-referenced”***), including robustness methods (Appendix A.1, Lines 704-720), fairness methods(Appendix A.2, Lines 723-736), and DRO-based approaches (Appendix A.2, Lines 739-755). If you believe that additional specific papers should be incorporated—such as the kernel-based DRO or information-theoretic fairness works mentioned—we would be grateful if the reviewer could point to the exact references, and we will be happy to include and discuss them in the revised version. Regarding privacy and interpretability methods, we note that these are indeed important directions within trustworthy graph learning. However, due to space limitations and in order to keep the technical focus clear, our work concentrates primarily on robustness and fairness. Nevertheless, our formulation naturally extends to privacy as well. As detailed in our response to **Reviewer dD69 (Part 3)**, DICT can incorporate privacy-related perturbations through the same uncertainty-modeling mechanism. In addition, we have explicitly added the privacy discussion in the revised manuscript (Figure 7 in Appendix I.7). We leave the implementation of interpretability extensions to future work.
>
> ---
> We sincerely hope we have addressed your concerns and raised your impression of our work. We are happy to clarify any other questions if required.
>
> ---
>
> TO CONTINUE

---

> ### Author Response · Authors · 2025-11-19
> **Response Part 8**
>
> ## References
> [1] Dai E, Wang S. Say no to the discrimination: Learning fair graph neural networks with limited sensitive attribute information[C]//Proceedings of the 14th ACM international conference on web search and data mining. 2021: 680-688.
>
> [2] Yuan H, Sun Q, Shi J, et al. How Much Can Transfer? BRIDGE: Bounded Multi-Domain Graph Foundation Model with Generalization Guarantees[C]//Forty-second International Conference on Machine Learning.
>
> [3] Chen Q, Wu Z, Su X, et al. Stable Fair Graph Representation Learning with Lipschitz Constraint[C]//Forty-second International Conference on Machine Learning.
>
> [4] Jia Y, Zou D, Wang H, et al. Enhancing node-level adversarial defenses by Lipschitz regularization of graph neural networks[C]//Proceedings of the 29th ACM SIGKDD conference on knowledge discovery and data mining. 2023: 951-963.
>
> [5] Mohajerin Esfahani P, Kuhn D. Data-driven distributionally robust optimization using the Wasserstein metric: Performance guarantees and tractable reformulations[J]. Mathematical Programming, 2018, 171(1): 115-166.
>
> [6]Grari V, Laugel T, Hashimoto T, et al. On the fairness road: Robust optimization for adversarial debiasing[J]. arXiv preprint arXiv:2310.18413, 2023.
>
> [7] Sadeghi A, Ma M, Li B, et al. Distributionally robust semi-supervised learning over graphs[J]. arXiv preprint arXiv:2110.10582, 2021.
>
> [8] Wang B, Chen J, Li C, et al. Distributionally robust graph-based recommendation system[C]//Proceedings of the ACM web conference 2024. 2024: 3777-3788.
>
> [9] Zhao C, Yang E, Liang Y, et al. Distributionally robust graph out-of-distribution recommendation via diffusion model[C]//Proceedings of the ACM on Web Conference 2025. 2025: 2018-2031.
>
> [10] Sagawa S, Koh P W, Hashimoto T B, et al. Distributionally robust neural networks for group shifts: On the importance of regularization for worst-case generalization[J]. arXiv preprint arXiv:1911.08731, 2019.
>
> [11] Taskesen B, Nguyen V A, Kuhn D, et al. A distributionally robust approach to fair classification[J]. arXiv preprint arXiv:2007.09530, 2020.

---

> ### Author Response · Authors · 2025-11-26
>
> Dear Reviewer 8w2e,
>
> I hope you are doing well. As the discussion period is nearing its end, I wanted to ensure that we have addressed all your concerns satisfactorily. If there are any additional points or feedback you would like us to consider, please let us know. Your insights are invaluable to us, and we are eager to address any remaining issues to further improve our work.
>
> Thank you for your time and effort in reviewing our paper.

---

### Official Review · Reviewer_dD69 · 2025-10-31

**Soundness:** 3
**Presentation:** 3
**Contribution:** 3
**Rating:** 6
**Confidence:** 2

**Summary:**

This paper proposes DICT, a unified framework for trustworthy graph learning that explicitly models distributional uncertainty in graph topology, features, labels, and sensitive attributes using a Wasserstein distributionally robust optimization formulation. DICT derives a dual  reformulation of the infinite-dimensional DRO problem using strong duality and local Lipschitz continuity, resulting in a tractable empirical loss regularized by a Lipschitz penalty that quantifies sensitivity to perturbations. The framework is demonstrated via specialization to both robustness and fairness objectives, and experiments on standard GNN backbones and multiple datasets show improvements in both robustness and fairness compared to strong baselines.

**Strengths:**

1. DICT presents a mathematically grounded and general framework that encompasses multiple trustworthiness objectives (robustness, fairness, privacy, etc.) within a single optimization structure.
2. The paper covers tractable algorithms for estimating node-level Lipschitz constants (via first- and second-order Taylor approximations), and describes all perturbation models (Appendix D) for empirical reproducibility.
3. The paper is well-written with clear figures.

**Weaknesses:**

1. While the theoretical min-max and duality reformulations are elegant, computing the entire Lipschitz penalty at scale may not be feasible for very large graphs or in real-world GNN deployments.
2. More GNNs should be included in the experiments.
3. The claim that DICT can be seamlessly extended to other trustworthiness goals (e.g., privacy) is reasonable in theory, but only sketched in passing and not demonstrated in a experiments. The framework's claims to extensibility are not substantiated outside robustness and fairness.

**Questions:**

1. Why didn't you add more classic GNNs, such as GAT, to Table 1 for experimentation?
2. Can the authors provide empirical or theoretical guarantees on the runtime/memory overhead incurred by approximating $L(\theta)$, especially for large graphs? Is the efficiency sufficient for scalable real-world deployment?
3. Beyond theoretical pointers, can the authors demonstrate (even via toy examples) how DICT can be instantiated for privacy or interpretability?

---

> ### Author Response · Authors · 2025-11-19
> **Response Part 1**
>
> Dear Reviewer dD69,
>
> Firstly, we express our gratitude for your thorough review and insightful comments on our paper. Your recognition of the **mathematically grounded, empirical reproducibility and well-written** of our work is greatly appreciated. We have taken your feedback as an opportunity to further refine our manuscript. We address your concerns below:
>
> ---
> > **W1**: While the theoretical min-max and duality reformulations are elegant, computing the entire Lipschitz penalty at scale may not be feasible for very large graphs or in real-world GNN deployments.
> >
> > **Q2**: Can the authors provide empirical or theoretical guarantees on the runtime/memory overhead incurred by approximating , especially for large graphs? Is the efficiency sufficient for scalable real-world deployment?
>
> Thank you for asking Q2 to help us clarify the runtime/memory overhead in DICT. We agree that approximating the Lipschitz term $L(\theta)$ introduces additional computation, which is an inherent characteristic of Lipschitz-regularized methods. However, DICT is specifically designed to  improve both robustness and fairness—an objective that is not simultaneously addressed by existing methods. Below we provide both **theoretical** and **empirical** analyses of the computational cost introduced by our approach.
> ### **Theoretical Efficiency**
>
> As we previously analyzed in Appendix I.1 (Complexity Analysis), we compare the computational complexity of DICT with representative adversarial-based and Lipschitz-based fair GNNs, including FairVGNN[1] and SFG[2], in terms of time, memory, and parameter count. Let $m$ represents the number of GNN layers, $K$ signifies the number of nodes, $E$ denotes the number of edges, $F$ indicates the hidden dimension, and $\rho$  represents a small constant (typically $\rho = 1$) indicating the number of additional forward/backward evaluations required by DICT to estimate the Lipschitz term. Lower-order terms (e.g., $O(E + F^{2})$ ) introduced by the perturbation generator are omitted, as they are dominated by message-passing complexity.
>
> - **FairVGNN** incurs the standard GNN complexity of  $O(mEF + mKF^{2})$  for both forward and backward passes, with memory cost  $O(E + mF^{2} + mKF)$.
> - **SFG** introduces an SVD on each weight matrix to enforce a tight Lipschitz bound, adding  $O(mF^{3})$to the forward pass and increasing memory by $mF^{2}$—a cost that quickly becomes prohibitive as $F$ grows.
> - **DICT**, in contrast, introduces only a *constant-factor* overhead due to the Lipschitz regularizer, yielding complexity $O\big((1+\rho)(mEF + mKF^{2})\big)$, while maintaining the same memory cost as FairVGNN.
>
> **Thus, DICT avoids the cubic-time cost of SFG and matches the lightweight complexity of FairVGNN up to a constant factor.**
>
> ### **Empirical Efficiency**
>
> While we cannot include figures in the rebuttal, the revised version provides runtime comparisons in **Appendix I.1, Figure 4**. We benchmark the training time on the Bail and Credit datasets using GCN as the encoder, comparing FairVGNN[1], SFG[2], NIFTY[3], FairGNN[4], and DICT. SFG serves as the baseline for evaluating efficiency among Lipschitz-based fair GNNs.
>
> The empirical results show:
>
> - **DICT achieves nearly the same training time as FairVGNN**, despite introducing the Lipschitz regularizer.
> - **DICT is substantially faster than SFG**, whose per-layer SVD-based Lipschitz projection incurs significant computational cost.
> - **DICT introduces predictable overhead** while providing consistent gains in robustness and fairness.
>
> DICT achieves a favorable efficiency–trustworthiness trade-off by preserving the lightweight complexity of FairVGNN up to a constant factor, while avoiding the cubic overhead required by SFG. These findings demonstrate that **DICT can be applied in real-world deployment** without substantial computational burden.
> | Model | Forward Time | Forward Space | Backward Time | Backward Space | Param. Count |
> | --- | --- | --- | --- | --- | --- |
> | FairVGNN | $mEF + mKF^{2}$ | $E + mF^{2} + mKF$ | $mEF + mKF^{2}$ | $E + mF^{2} + mKF$ | $mF^{2}$ |
> | SFG | $mEF + mKF^{2} + mF^{3}$ | $E + mF^{2} + mKF + mF^{2}$ | $mEF + mKF^{2}$ | $E + mF^{2} + mKF$ | $mF^{2}$ |
> | DICT (ours) | $(1+\rho)(mEF + mKF^{2})$ | $E + mF^{2} + mKF$ | $(1+\rho)(mEF + mKF^{2})$ | $E + mF^{2} + mKF$ | $mF^{2}$ |
>
> ---
> TO CONTINUE

---

> ### Author Response · Authors · 2025-11-19
> **Response Part 2**
>
> > **W2**: More GNNs should be included in the experiments.
> >
> >
> > **Q1**: Why didn't you add more classic GNNs, such as GAT, to Table 1 for experimentation?
> >
>
> Thanks for your valuable suggestion! In the initial submission, we selected GCN, GraphSAGE, and GIN as GNN backbones, following the setup of EDITS[5] to ensure a fair and consistent comparison. Nevertheless, to address the reviewer’s concern, we have incorporated GAT, the widely used attention-based GNN. The updated results have been added to **Table 2** in the revised manuscript, with all newly added content highlighted in red.
>
> | Datasets | Metrics | GAT (Vanilla) | GAT (DICT) |
> | --- | --- | --- | --- |
> | German | Clean Acc(%) ↑ | 65.72 ± 2.27 | **70.22 ± 0.77** |
> |  | Attack Acc(%) ↑ | 62.35 ± 1.80 | **68.33 ± 0.25** |
> |  | $\Delta\mathrm{Acc}$(%) ↓ | 3.37 | **1.89** |
> |  | $\Delta_\mathrm{SP}$(%) ↓ | 3.27 ± 0.31 | **1.17 ± 0.21** |
> |  | $\Delta_\mathrm{EO}$(%) ↓ | 2.82 ± 0.25 | **0.82 ± 0.15** |
> | Bail | Clean Acc(%) ↑ | 87.32 ± 0.7 | **89.37 ± 0.5** |
> |  | Attack Acc(%) ↑ | 68.25 ± 0.6 | **87.23 ± 0.6** |
> |  | $\Delta\mathrm{Acc}$(%) ↓ | 19.07 | **2.14** |
> |  | $\Delta_\mathrm{SP}$(%) ↓ | 9.98 ± 1.2 | **1.63 ± 0.8** |
> |  | $\Delta_\mathrm{EO}$(%) ↓ | 7.19 ± 1.3 | **1.29 ± 0.4** |
> | NBA | Clean Acc(%) ↑ | 72.30 ± 0.8 | **73.40 ± 0.2** |
> |  | Attack Acc(%) ↑ | 67.21 ± 0.7 | **73.32 ± 0.6** |
> |  | $\Delta\mathrm{Acc}$(%) ↓ | 5.05 | **2.16** |
> |  | $\Delta_\mathrm{SP}$(%) ↓ | 13.29 ± 1.1 | **2.21 ± 0.9** |
> |  | $\Delta_\mathrm{EO}$(%) ↓ | 11.90 ± 2.0 | **1.72 ± 1.1** |
>
> To further illustrate the adaptability of our method, we emphasize that DICT requires only a mild regularity assumption on the backbone encoder—specifically, that the graph encoder is **locally Lipschitz** (see our detailed discussion in the response to Reviewer 8w2e, W7). This condition ensures that (i) the Wasserstein-robust objective admits a tractable dual form and (ii) the Lipschitz-based perturbation sensitivity used in DICT can be reliably estimated (Sec. 3, lines 253–278). This assumption is highly general and is satisfied by a broad class of differentiable architectures, including **Transformer-based GNNs**. Their computation consists of linear projections, dot-product attention with softmax, residual connections, and smooth nonlinearities, all of which are **locally Lipschitz** almost everywhere. Consequently, DICT accommodates Transformer-based backbones by replacing the encoder, with no changes to the underlying DRO formulation and uncertainty modeling.
>
> In the revised version, we have **added the experiment using a Transformer-based graph encoder**, and compared with two representative Transformer-based GNNs, NAGphormer[6] and FairGT[7]. The newly added content is highlighted in red in **Appendix I.6** (Extension to Transformer-based GNNs). DICT consistently improves both clean and adversarial accuracy across the German, Bail, and NBA datasets. In addition, DICT achieves **significantly stronger robustness** (lower $\Delta\mathrm{Acc}$) and **superior fairness** (lower $\Delta_\mathrm{SP}$ and $\Delta_\mathrm{EO}$). These results validate that DICT provides a plug-and-play trustworthy learning framework  that generalizes effectively from message-passing GNNs to advanced Transformer-based graph models.
>
> | Datasets | Metrics | NAGphormer | FairGT | DICT |
> | --- | --- | --- | --- | --- |
> | German | Clean Acc(%) ↑ | 75.43±0.42 | 76.07±0.19 | **76.91±0.27** |
> |  | Attack Acc(%) ↑ | 71.32±0.51 | 68.22±0.25 | **75.83±0.21** |
> |  | $\Delta\mathrm{Acc}$(%) ↓ | 4.11 | 7.85 | **1.08** |
> |  | $\Delta_\mathrm{SP}$(%) ↓ | 8.24±0.05 | 0.25±0.29 | **0.19±0.22** |
> |  | $\Delta_\mathrm{EO}$(%) ↓ | 6.86±0.03 | 0.17±0.04 | **0.11±0.51** |
> | Bail | Clean Acc(%) ↑ | 93.39±0.32 | **95.48±0.36** | 94.62±0.23 |
> |  | Attack Acc(%) ↑ | 86.12±0.64 | 86.33±0.33 | **92.89±0.33** |
> |  | $\Delta\mathrm{Acc}$(%) ↓ | 7.27 | 9.15 | **1.73** |
> |  | $\Delta_\mathrm{SP}$(%) ↓ | 7.13±0.29 | 0.55±0.34 | **0.33±0.29** |
> |  | $\Delta_\mathrm{EO}$(%) ↓ | 5.79 | 0.43 | **0.27** |
> | NBA | Clean Acc(%) ↑ | 72.17±0.34 | 74.78±0.31 | **75.48±0.11** |
> |  | Attack Acc(%) ↑ | 67.21±0.72 | 66.37±0.23 | **73.32±0.64** |
> |  | $\Delta\mathrm{Acc}$(%) ↓ | 5.05 | 8.41 | **2.16** |
> |  | $\Delta_\mathrm{SP}$(%) ↓ | 16.24±0.67 | 0.35±0.23 | **0.21±0.46** |
> |  | $\Delta_\mathrm{EO}$(%) ↓ | 12.32±0.35 | 0.26±0.42 | **0.17±0.36** |
>
> ---
> TO CONTINUE

---

> ### Author Response · Authors · 2025-11-19
> **Response Part 3**
>
> > **W3**: The claim that DICT can be seamlessly extended to other trustworthiness goals (e.g., privacy) is reasonable in theory, but only sketched in passing and not demonstrated in a experiments. The framework's claims to extensibility are not substantiated outside robustness and fairness.
> >
> > **Q3**: Beyond theoretical pointers, can the authors demonstrate (even via toy examples) how DICT can be instantiated for privacy or interpretability?
> >
>
> We thank you for the valuable comment. While our main experiments focus on robustness and fairness, the extensibility of DICT is not only theoretically grounded but can also be instantiated in practice. To further support this point, we have added an experiment demonstrating how DICT can be **instantiated for to privacy**.
>
> In the earlier version, we provided a theoretical discussion in Section 2 (lines 157–161) on how differential privacy (DP) can be incorporated into DICT. Specifically, we described that local differential privacy (LDP) can be achieved by flipping each binary sensitive attribute  $s_i \in$ \{0,1\} with probability $\rho = \tfrac{1}{\exp(\epsilon)+1},$ which produces a privatized attribute $\tilde{s}_i$ . This stochastic flipping ensures that the learner observes only  $\tilde{s}_i$, while the true attribute $s$ remains indistinguishable under an  $\epsilon$-LDP guarantee, thereby enabling privacy-preserving GNN training within the DICT framework. In our experiments, we  generate $\tilde{s}_i$ and train DICT under the same fairness-DRO objective, simply replacing $s_i$ with $\tilde{s}_i$ in all fairness-related terms.
>
> Since figures cannot be included directly in the rebuttal, the results have been added to the revised manuscript as **Figure 7 in Appendix I.7**. We compare DICT with FairGCN[4], and NTFC[8] under identical  $\epsilon$-LDP  protocols following NT-FairGNN[9], and evaluate accuracy and group fairness ($\Delta_\mathrm{SP}$, $\Delta_\mathrm{EO}$ ). The results show that:
>
> - DICT achieves **lower** $\Delta_\mathrm{SP}$ and $\Delta_\mathrm{EO}$ under **all privacy budgets**, with particularly large gains under strong privacy (small $\epsilon$).
> - DICT maintains the **highest accuracy** across the entire privacy range.
>
> These empirical findings confirm that DICT can be  instantiated as a **privacy-preserving** graph learning framework. By plugging an LDP mechanism into the sensitive-attribute perturbation space, DICT integrates privacy seamlessly within the same uncertainty-constrained DRO formulation.
>
> ---
> We sincerely hope that our clarifications above have increased your confidence in our work. We will be happy to clarify further if needed. We thank you again for sharing your valuable feedback on our work.
>
> ---
> ## References
>
> [1] Wang Y, Zhao Y, Dong Y, et al. Improving fairness in graph neural networks via mitigating sensitive attribute leakage[C]//Proceedings of the 28th ACM SIGKDD conference on knowledge discovery and data mining. 2022: 1938-1948.
>
> [2] Chen Q, Wu Z, Su X, et al. Stable Fair Graph Representation Learning with Lipschitz Constraint[C]//Forty-second International Conference on Machine Learning.
>
> [3] Agarwal C, Lakkaraju H, Zitnik M. Towards a unified framework for fair and stable graph representation learning[C]//Uncertainty in artificial intelligence. PMLR, 2021: 2114-2124.
>
> [4] Dai E, Wang S. Say no to the discrimination: Learning fair graph neural networks with limited sensitive attribute information[C]//ICDM. 2021: 680-688.
>
> [5] Dong Y, Liu N, Jalaian B, et al. Edits: Modeling and mitigating data bias for graph neural networks[C]//Proceedings of the ACM web conference 2022. 2022: 1259-1269.
>
> [6] Chen J, Gao K, Li G, et al. NAGphormer: A Tokenized Graph Transformer for Node Classification in Large Graphs[C]//The Eleventh International Conference on Learning Representations.
>
> [7] Luo R, Huang H, Yu S, et al. FairGT: a fairness-aware graph transformer[C]//Proceedings of the Thirty-Third International Joint Conference on Artificial Intelligence. 2024: 449-457.
>
> [8] Lamy A, Zhong Z, Menon A K, et al. Noise-tolerant fair classification[J]. Advances in neural information processing systems, 2019, 32.
>
> [9] Dai E, Wang S. Learning fair graph neural networks with limited and private sensitive attribute information[J]. IEEE Transactions on Knowledge and Data Engineering, 2022, 35(7): 7103-7117.

---

> ### Author Response · Authors · 2025-11-26
>
> Dear Reviewer dD69,
>
> I hope you are doing well. As the discussion period is nearing its end, I wanted to ensure that we have addressed all your concerns satisfactorily. If there are any additional points or feedback you would like us to consider, please let us know. Your insights are invaluable to us, and we are eager to address any remaining issues to further improve our work.
>
> Thank you for your time and effort in reviewing our paper.

---

### Author Response · Authors · 2025-12-02
**Summary from the Authors (Part 2/2)**

``Reviewer aGWN``:

- **[Raw accuracy results.]** The reviewer suggested providing raw accuracy values rather than $\Delta\mathrm{Acc}$. We have added the full raw accuracy results to enable a more convincing comparison (see Appendix I.4).
- **[Hyperparameter sensitivity.]** The reviewer raised questions regarding the sensitivity of DICT to its hyperparameters.  In response, we have provided a detailed explanation of all hyperparameters in the **official comments**, including the transport costs  $\kappa_A, \kappa_X, \kappa_y, \kappa_s$.
- **[Extension to transformer-based GNNs.]** The reviewer suggested extending our method to Transformer-based GNNs. We have included an experiment using a Transformer-based graph encoder to broaden DICT’s applicability (see Appendix I.6).

``Reviewer 8w2e``:

Reviewer 8w2e’s comments contain factual errors and overlooked details, indicating that the review does not constitute a careful scientific evaluation and may have been generated using AI. Please let us list the evidence to support our argument.

1. **Factual error:**
    - **[Incorrect references.]** In **W8**, The reviewer states that “the ROAD method (Grari et al., 2024), which also uses robust optimization for fairness in GNNs.” ROAD is **not a graph-based method** and does not involve any GNN architectures or message-passing mechanisms. This is a common citation hallucination in AI-generated.
    - **[Incorrect references.]** In **W9**, the reviewer mentions “recent work in robust and fair graph learning, such as those applying **kernel-based DRO** or information-theoretic fairness constraints.” To the best of our knowledge, no kernel-based DRO methods have been applied to robust and fair graph learning. This is a factual error and reflects the AI-generated hallucination.
2. **Overlooked details in the paper:**
    - **[Overlooked definitions.]** The reviewer states that “fairness and robustness are not formally defined.” However, the paper already provides definitions: robustness is defined in Section 4.1 and fairness is defined in Section 4.2. We further clarify the use of $\Delta\mathrm{Acc}$, $\Delta_\mathrm{SP}$ and $\Delta_\mathrm{EO}$ as the robustness and fairness metrics in Section 5.1. Moreover, Appendix G includes the definitions of Statistical Parity (SP) and Equal Opportunity (EO).
    - **[Overlooked definitions.]** The relationships among the observed labels $\widehat{\mathbf y},\widehat{\mathbf y}_k$ and the true labels $\mathbf y,\mathbf y_k$ are clearly defined in Section 2 (Lines 100-102), yet the reviewer states they are “not explained.” in **W4**.
3. **AI-likeness evidence:** We conducted an independent analysis using multiple AI-text detectors (Turnitin AI, GPTZero, Pangram). The reviewer’s comments consistently scored in the **full AI-generated** and exhibited characteristic patterns, including template-like sentence and paragraph structures as well as citation hallucination (see **W8** and **W9**). While AI detectors are not perfect, the results further support the above evidence suggesting that the review may not result from careful human evaluation. We understand the heavy workloads reviewers often face, but AI-generated reviews that are low-quality and have not been checked by a human are problematic and undermine the fairness and rigor of the evaluation process.

Taken together, these issues indicate that the comment by ``Reviewer 8w2e`` does not constitute a careful scientific evaluation of our work and is likely **AI-generated**. In addition, the reviewer has not participated in the discussion phase despite our **detailed responses** to all of his/her concerns.

---

Once again, we greatly appreciate your time, effort, and careful consideration of our work. We sincerely hope that this summary will be helpful for your assessment.

The authors of Paper 9268

---

### Author Response · Authors · 2025-12-02
**Summary from the Authors (Part 1/2)**

Dear Area Chair, Senior Area Chair and Program Chair,

We sincerely appreciate the additional reviewing responsibilities that you have undertaken, and we fully understand the difficulty of making fair decisions in a situation like this. In support of your work, we would like to **provide a succinct overview of our paper, its contributions, and the outcome of the rebuttal,** in the hope of alleviating your workload as much as possible.

## Overview of our work

This paper establishes, for the first time, a unified framework that simultaneously optimizes multiple trustworthiness objectives by modeling distributional uncertainty via Wasserstein distributionally robust optimization (DRO), a milestone that has remained elusive despite years of effort on trustworthy graph learning that handles each objective in isolation.

**1.**  We are the first to unify multiple trustworthiness objectives, such as robustness and fairness, within a single distributional uncertainty framework, addressing the challenge that these objectives often conflict with one another and are difficult to jointly optimize. This contribution was explicitly recognized by:

> ``Reviewer aGWN``: framework is highly novel and appealing; ``Reviewer dD69``: a mathematically grounded and general framework; ``Reviewer 8w2e``: well-established DRO framework
>

**2.**  For the first time, we generate i.i.d. graph-level perturbation samples and construct an empirical mixture distribution, overcoming the fundamental challenge that graph data are inherently non-i.i.d. due to dependencies among nodes and edges. This contribution was explicitly recognized by:

> ``Reviewer 8w2e``: framework is built upon a well-established distributionally robust optimization principle, extended to handle graph-structured data with non-i.i.d. dependencies
>

**3.**  This paper, for the first time, reduces the infinite-dimensional DRO problem on graphs to a tractable finite-dimensional objective via strong duality and local Lipschitz approximations, thereby enabling a practical and theoretically grounded DRO optimizer. This contribution was explicitly recognized by:

> ``Reviewer dD69``: provides tractable Lipschitz estimation algorithms and clearly specifies perturbation models to ensure reproducibility; ``Reviewer aGWN``: provides a very clear derivation; ``Reviewer 8w2e``: a computationally feasible approximation of the intractable primal problem
>

**4.**  Extensive experiments demonstrate that DICT consistently improves robustness and fairness under distributional shifts across multiple datasets and representative GNN backbones, validating the practical effectiveness of our unified distributional uncertainty framework. This contribution was explicitly recognized by:

> ``Reviewer aGWN``: the proposed method achieves excellent performance; ``Reviewer dD69``: show improvements in both robustness and fairness compared to strong baselines; ``Reviewer 8w2e``: experiments are thorough, reasonable and practically effective
>

## Summary of the rebuttal

We thank all reviewers for their careful reading and valuable feedback. We carefully responded to every questions and concerns raised by all three reviewers, and further strengthened the paper through **additional GNN-backbone experiments, extension to privacy objectives, complexity analysis, hyperparameter explanations**, and several other improvements. All revisions are highlighted with **`red`** throughout the PDF.

`Reviewer dD69`:

- **[More GNNs.]** The reviewer suggested that more GNNs should be included in the experiments. We have added results using GAT to illustrate the architectural flexibility of DICT (Table 2 of the revised manuscript) and included an additional experiment with a Transformer-based graph encoder to further demonstrate its adaptability (see Appendix I.6).
- **[Complexity analysis.]** The reviewer raised questions regarding the runtime and memory overhead of our approximation procedure, particularly for large graphs. In response, we have added comprehensive empirical and theoretical guarantees on both runtime and memory usage (see Appendix I.1). These findings demonstrate that DICT can be deployed in large graphs without incurring substantial computational overhead.
- **[Extension to privacy.]** The reviewer raised questions about whether DICT can be instantiated for privacy or interpretability beyond the theoretical discussion. We have added an experiment demonstrating a concrete privacy instantiation of DICT to show its applicability to other trustworthiness objectives (see Appendix I.7).

TO CONTINUE

---

### Note · Program_Chairs · 2026-01-17
**Submission Desk Rejected by Program Chairs**

The following references in this submission do not refer to real documents and/or have major errors in bibliographic information:

 Rachid Grari, Clément Laclau, Hisashi Sato, and Zaid Harchaoui. "Road: Robust Optimization of Fair Graph Learning under Demographic Change." Published at NeurIPS, 2024.